

# A Backscatter Lidar Forward Operator for Particle-Representing Atmospheric Chemistry Models

Armin Geisinger[1], Andreas Behrendt[1], Volker Wulfmeyer[1], Jens Strohbach[1], Jochen Förstner[2], Roland Potthast[2], and Ina Mattis[3]

[1]Institute of Physics and Meteorology, University of Hohenheim, Germany
[2]Headquarter of the German Weather Service, Offenbach, Germany
[3]Observatory Hohenpeissenberg of the German Weather Service, Hohenpeissenberg, Germany

*Correspondence to:* a.geisinger@uni-hohenheim.de

**Abstract.**

State-of-the-art atmospheric chemistry models are capable of simulating the transport and evolution of particles and trace gases but there is a lack of reliable methods for model validation and data assimilation. Networks of automated ceilometer lidar systems (ACLs) could be used to fill this gap. These are already used for the detection of clouds and aerosols, providing a 3D

dataset of atmospheric backscatter profiles. However, as the aerosol number concentration cannot be obtained from the ACL data alone, one needs a backscatter-lidar forward model to simulate lidar profiles from the model variables. Such an operator allows then for a qualitative and quantitative model validation based on ACL data. In this work, we present a new backscatter-lidar operator which contains most of the microphysical properties of aerosol particles, discuss sensitivity studies and compare simulated with measured ACL profiles. A major challenge is the high sensitivity of the optical cross sections to the particle size

and shape: A slightly different particle radius may lead to quite a large change of the scattering properties. As most particle size distributions are continuous in reality, the optical cross sections are averaged over certain size-intervals which also reduces the problematic and unrealistic sensitivity significantly. To calculate the attenuated backscatter coefficient, the size-dependent particle number concentration and the scattering properties of each particle type and size have to be simulated. While the particle number concentration is a model output variable, the scattering properties have to be determined by extensive scattering

calculations. As these scattering calculations require assumptions about the particle refractive indices and shapes, sensitivity studies were performed to estimate the uncertainties related to the particle properties as represented by the model system. The strong sensitivity of the scattering characteristics to the particle radius was largely reduced by size-averaging algorithms. We focus on a case study of the eruption of the Islandic volcano Eyjafjallajökull from 20 March 2010 to 24 May 2010. The Consortium for Small-scale Modeling - Aerosols and Reactive Trace gases (COSMO-ART) model of DWD (Deutscher Wet-

terdienst) and KIT (Karlsruhe Institute of Technology) was used during this event for ash-dispersion simulation over Europe. For the forward model, the attenuated backscatter coefficient is used as lidar-independent variable, as it only relies on the laser wavelength. Finally, the forward modeled lidar profiles were compared to ACL measurements. Significant differences between ACL profiles and the output of the forward operator applied to the COSMO-ART data were found but also several identical features have been observed. Comparing the data quantitatively revealed that the model-predicted ash number concentration

is slightly too high. We identified the following key issues which are mandatory for performing quantitative comparisons be-





tween forward modeled and measured ACL profiles: First, it is suggested that the ACL systems perform automatic calibration and return the attenuated backscatter coefficient directly. Second, the particles' scattering properties have to be analyzed even more extensively as significant differences of the backscatter efficiency occur depending on the shape and refractive indices. Nevertheless, the results of this study allow for a quantitative estimation of the volcanic ash mass concentration from ACL data

as well as the related uncertainties. We consider this forward operator development as a crucial step for the future assimilation of the huge information contents of lidar backscatter data in chemical weather forecast models.

# 1 Introduction

Particles play an important role in the Earth's weather and climate (Lohmann and Feichter, 2005) as well as for our every day life in terms of air quality (Pöschl, 2005). The role of aerosols in the Earth's atmosphere is commonly separated into direct

and indirect effects (Boucher et al., 2013). Direct aerosol effects are due to the interaction with electromagnetic waves and the resulting consequences on the energy balance. The follow-up processes of direct effects are called indirect effects which are, for example, effects on cloud formation, water cycle, and plant growth. In recent years, numerical atmospheric chemistry models have become capable to simulate the evolution, transport and interaction of aerosols in the atmosphere. But observations to validate these simulations are rare.

Atmospheric chemistry models are, amongst others, COSMO-ART (Consortium for Small-scale Modeling, Aerosols and Reactive Trace gases, Vogel et al. (2009)), COSMO-MUSCAT (Multiscale Chemistry Aerosol Transport, Wolke et al. (2004)), ECMWF (European Center for Mesoscale Weather Forecast, Benedetti (2009), ENVIRO-HIRLAM (Environment - High Resolution Limited Area Model, Zakey et al. (2006)), MCCM (Multiscale Coupled Chemistry Model, Emeis et al. (2011)), MesoNH (Non-Hydrostatic Mesoscale Atmospheric Model of the French research community, Mallet et al. (2009)), WRF-

CHEM (Weather Research and Forecast Model, Chen et al. (2014)). Using these model systems, scientists have analyzed the aerosol influence on, for example, precipitation (Rieger et al., 2014), temperature (Bangert et al., 2012), radiative fluxes (Vogel et al., 2009; Chaboureau et al., 2011).

The simulation of aerosols can be separated in components: sources, sinks and evolution as well as transport and conversion processes. While the transport processes are similar for any aerosol type with only sedimentation being potentially different,

the sources and sinks have to be described individually. For example, the emission rate of mineral dust from deserts depends on the wind speed near ground (Vogel et al., 2014a). Processes leading to sinks and conversation are washout, chemical reactions, and particle aggregation. The transport processes mainly depend on the model dynamics (wind speed, wind direction) and sedimentation properties of each individual aerosol type. In consequence, the validation of aerosol transport model simulations with observations allows not only for the validation of the sources and sinks but serves also to validate the model dynamics. If

such validation can be done automatically, data assimilation is the next step to continuously converge simulation and reality. Compared to other atmospheric variables such as pressure and temperature, there are - to the best of our knowledge - no comprehensive measurement systems in use so far validating aerosol simulations operationally.





The only remote sensing technique which is capable of providing information on atmospheric particles with high temporal and spatial resolution is lidar (light detection and ranging). The most basic lidar type is the backscatter lidar which measures the backscattered signal intensity of a volume at a certain range. Comparing the data of such a backscatter lidar that is operated in the UV with simulations of an atmospheric chemistry model, allows, e.g., for the characterization of transport and optical prop-

erties of aerosol particles near sources (Behrendt et al., 2011; Álvaro M. Valdebenito B et al., 2011). Using ground-based DIAL (differential absorption lidar, Dagan (2008); Späth et al. (2016)) water-vapor can be measured, which can even be combined with backscatter measurements to derive more details of aerosol particle properties (Wulfmeyer and Feingold, 2000). Lidar techniques based on the vibrational and rotational Raman effect, like RRL (rotational Raman lidar) allow for the measurement of trace gas profiles (Whiteman et al., 1992; Turner et al., 2002; Wulfmeyer et al., 2010), as well as profiles of atmospheric tem-

perature, particle backscatter cross section, particle extinction cross section, and particle depolarization properties (Behrendt et al., 2002; Hammann et al., 2015; Radlach et al., 2008). HSRL (high spectral resolution lidar) systems furthermore allow for cloud and particle characterization (Shipley et al., 1983). Multiwavelength lidar systems offer the potential to retrieve the optical, microphysical and chemical properties of aerosols (Mamouri et al., 2012) but these systems are rare and the retrieval algorithms are very complex. Profiles of the radial wind speed can be obtained by Doppler-Lidar systems, e.g., (Banta et al.,

15  2012).

While the number of sophisticated lidar instruments that provide thermodynamic data (Wulfmeyer et al., 2015) is still low, there are already automated aerosol lidar networks in operation in Europe and Asia (Pappalardo et al., 2014; Sugimoto et al., 2008). For example the German weather service (Deutscher Wetterdienst, DWD) maintained already in 2010 a network of about 36 automated ceilometer lidar (ACL) systems (Flentje et al., 2010a). The data of such a network offers 3D particle

information with a high temporal, high vertical and a moderate horizontal resolution. The ACL systems have been used to detect cloud and boundary layer heights (Emeis et al., 2009) but the received signal delivers also information about aerosols. It is therefore worthwhile to use the ACL network measurements for the validation of particle transport model simulations.

Unfortunately, it is not possible to obtain the particle number concentration from an elastic backscatter signal alone without ancillary information and assumptions which are partly critical. The preferable way is thus to use the detailed atmospheric

description of the model to simulate lidar profiles for a model-given atmospheric state. Then, the quantitative comparison can be performed between the simulation and observation more accurately. Such a lidar simulator is called lidar forward operator.

There are already several backscatter lidar forward operators available or in development. At ECMWF (European Centre for Medium-Range Weather Forecasts), a lidar forward operator was developed and tested in an ice cloud scenario (Benedetti, 2009), based on assumptions concerning the lidar ratio $S_{\mathrm{lidar}}$. Newer implementations of the ECMWF model allow for sim-

ulating the backscatter coefficient measurement of the satellite-mounted CALIOP (Cloud-Aerosol Lidar with Orthogonal Polarization) at two wavelengths $532\,\mathrm{nm}$ and $1064\,\mathrm{nm}$ for spherical sea salt and mineral dust particles (Morcrette et al., 2009). The lidar forward model of SIČ (2014) has the capability to calculate the atmosphere optical depth (AOD) and the particle extinction coefficient from simulations of the CTM MOCAGE (Chemical-Transport Model, Modèle de Chimie Atmosphérique à Grande Echelle). But similar to the ECMWF forward operator, assumptions on the lidar ratio $S_{\mathrm{lidar}}$ are mandatory to calculate

the particle backscatter coefficient. At MetOffice, UK, a backscatter lidar forward model is being investigated (Charlton-Perez



et al., 2013) which considers clouds, one type of aerosol, and rain simulated by the Unified Model (UM). Another lidar forward model is being developed for the EURAD-IM (European Air Pollution and Dispersion – Inverse Model, Lange and Elbern (2014)), but no published results have been found in literature. Consequently, all existing lidar forward operators which we found in literature calculate only the extinction coefficient. The backscatter coefficient, however, is not calculated explicitly for

given atmospheric scatterer species. On the one hand, this method benefits from the fact that the extinction coefficient is less sensitive to the particle properties than the backscatter coefficient. On the other hand, the precision of this method is limited to the correctness of assumed lidar ratio. The method becomes furthermore unusable once there is a mixture of scatterers for which the effective lidar ratio is not known. To become independent on the assumption of a lidar ratio, we designed a forward operator which is based on the independent calculation of both the extinction and the backscatter coefficients.

Our forward operator can be adapted to any particle-representing atmospheric model system and backscatter lidar system even using multiple wavelengths. The forward operator is optimized to calculate both the attenuated backscatter coefficient and the lidar ratio from model output data with a minimum set of external information. We call this forward model the "backscatter lidar forward operator" (BaLiFOp). This forward operator is planned to be used for particle data assimilation.

Within its priority project KENDA (Kilometer-Scale Ensemble Data Assimilation) the COSMO Consortium has developed

an Ensemble Kalman Filter for data assimilation on the convective scale. It is scheduled for introduction into operational use by MeteoSwiss and DWD in 2016. An advantage of the ensemble data assimilation system is that the assimilation can be carried out based on the pure forward operator, and that it is not necessary to calculate derivatives of the forward operator or the adjoint tangential model for carrying out data assimilation. Also, it naturally introduces model increments for all variables where some dynamic covariance is observed from the underlying ensemble model runs. DWD aims to test the assimilation of

ACL data into the COSMO-ART model based on the BaLiFOp Lidar forward operator. Here, the current work is a necessary and important basis for further work to improve numerical weather prediction and particle forecasting.

In the following we explain the lidar principles and the theoretical background for the backscatter lidar forward operator (Sect. 2). This is followed by an introduction to the case study in Sect. 3. Sensitivity studies of the particles' scattering properties are presented in Sect. 4. Results of the forward operator and a comparison to ACL measurement data are shown in Sect. 5.

Finally, we discuss both the benefits and the requirements of ACL data assimilation systems (Sect. 6).

## 2   Methods

### 2.1   The Lidar Equation

The lidar principle is based on the emission of UV, visible or IR laser pulses into the atmosphere and the measurement and analysis of the backscatter signals. The received photon number per pulse $N_{\text{rec},\lambda}(z)$ from range $z$ is described by the following

equation for elastic lidars which detect the backscatter signal at the emitted wavelength

$$N_{\text{rec},\lambda}(z) = N_{\text{tr},\lambda} \frac{\tau c}{2} \eta_\lambda \, O(z) \, \frac{A_{\text{tel}}}{z^2} \, \beta_\lambda(z) \, \exp\left(-2 \int_0^z \alpha_\lambda(z') \, dz'\right). \tag{1}$$





Instrument-dependent variables of the lidar equation are the wavelength $\lambda$, the laser emitted photon number per pulse $N_{\mathrm{tr},\lambda}$, the temporal length of a laser pulse $\tau$, the speed of light $c$, the efficiency of the receiving system and detectors $\eta_\lambda$, the overlap function $O(z)$, and the net area of the receiving telescope $A_{\mathrm{tel}}$. The received signal intensity can be given either as power or in photon counts. Here, we use photon counts per laser pulse unless otherwise noted.

The range resolution is related to the temporal resolution of the data acquisition system by $\frac{\tau c}{2} = \Delta z$. Typical $\Delta z$ values for ACL systems are a few meters. The overlap function $O(z)$ is zero (no overlap) near ground and becomes 1 (full overlap) above a certain height which is typically $200\,\mathrm{m}$ to $1500\,\mathrm{m}$ above ground for ACL systems (Wiegner et al., 2014; Flentje et al., 2010b). The missing overlap limits the capability to measure in near range but has no effect on heights where full overlap has accomplished. Heights where $0 < O(z) < 1$ can be overlap-corrected if the device specific overlap function is known.

Processes in the atmosphere are described by the backscatter coefficient $\beta_\lambda(z)$ and the extinction coefficient $\alpha_\lambda(z)$. The backscatter coefficient $\beta_\lambda(z)$ describes the scattering strength into the direction of the receiving telescope and depends on the wavelength, the type, shape and size of scatterers, and their respective number-concentrations; $\beta_\lambda(z)$ is given in units of $\mathrm{m}^{-1}\,\mathrm{sr}^{-1}$. The extinction coefficient $\alpha_\lambda(z)$ is a description for the light absorption and light scattering capabilities of objects in a volume; it is given in units of $\mathrm{m}^{-1}$.

Elastic backscatter lidar systems do not allow for the separate measurement of $\beta_\lambda(z)$ and $\alpha_\lambda(z)$ as two unknowns cannot be determined with one measured variable. For calibrated backscatter lidar systems, it is thus convenient to calculate the attenuated backscatter coefficient $\gamma_\lambda(z)$ from the measured profiles

$$\gamma_\lambda(z) = \beta_\lambda(z) \, \exp\left(-2 \int\limits_0^z \alpha_\lambda(z')\, dz'\right). \tag{2}$$

It is given in units of $\mathrm{m}^{-1}\,\mathrm{sr}^{-1}$. The attenuated backscatter coefficient is independent of all instrument-specific parameters

except the wavelength. Therefore, it is the best suitable physical quantity for a comparison between ACL measurement and aerosol model using a forward operator as long as no more information concerning the aerosol microphysics is available.

## 2.2   The Backscatter Lidar Forward Operator

According to Eq. (2), the basic task of the forward operator is to calculate the extinction coefficient $\alpha_\lambda(z)$ and backscatter coefficient $\beta_\lambda(z)$ based on the atmospheric state simulated by a model. Knowing the extinction coefficient of a vertical column

allows for the solution of the two-way-transmission integral and, finally, to determine the attenuated backscatter coefficient $\gamma_\lambda(z)$. In the following, we describe our method of calculating the attenuated backscatter coefficient from a given set of atmospheric scatterer data.




### 2.2.1 Scattering of Light by Arbitrary Objects

If there are $q_s$ different types of scatterers in an illuminated volume, the total extinction coefficient $\alpha_\lambda(z)$ and the total backscatter coefficient $\beta_\lambda(z)$ are calculated with

$$\alpha_\lambda(z) = \sum_{i=1}^{q_s} \alpha_{i,\lambda}(z) = \sum_{i=1}^{q_s} \int_0^\infty n_i(R,z)\, \sigma_{\text{ext},i,\lambda}(R)\, dR, \tag{3}$$

$$\beta_\lambda(z) = \sum_{i=1}^{q_s} \beta_{i,\lambda}(z) = \sum_{i=1}^{q_s} \int_0^\infty n_i(R,z) \left(\frac{d\sigma_{\text{sca},i,\lambda}(R)}{d\Omega}\right)_\pi dR, \tag{4}$$

where $n_i(R,z)$ is the number-size distribution of scatterer type $i$ at range $z$ given in units of $\text{m}^{-3}$, $\sigma_{\text{ext},i,\lambda}$ is the extinction cross-section of scatterer type $i$ given in units of $\text{m}^2$ with radius $R$, and $\left(\frac{d\sigma_{\text{sca},i,\lambda}}{d\Omega}\right)_\pi$ is the differential backscatter cross section given in units of $\text{m}^2\,\text{sr}^{-1}$.

10     For isotropic scattering, the differential backscatter cross section is derived from the scattering cross-section $\sigma_{\text{sca},i,\lambda}(R)$ via

$$\left(\frac{d\sigma_{\text{sca},i,\lambda}(R)}{d\Omega}\right)_\pi = \frac{\sigma_{\text{sca},i,\lambda}(R)}{4\pi\,\text{sr}}. \tag{5}$$

For non-isotropic scattering, a phase function $\phi_{i,\lambda}(\theta,R)$ is used to describe the relative scattering intensity into an angle $\theta$. In the case of monostatic ACL systems, we thus get $\theta = \pi$, giving

$$\left(\frac{d\sigma_{\text{sca},i,\lambda}(R)}{d\Omega}\right)_\pi = \frac{\sigma_{\text{sca},i,\lambda}(R)}{4\pi\,\text{sr}}\, \phi_{i,\lambda}(\pi,R). \tag{6}$$

15     In the following, we treat molecule scattering and particle scattering separately, as the respective calculations depend on different physical theories and algorithms.

### 2.2.2 Scattering by Molecules

Assuming that a model distinguishes atmospheric gases such as nitrogen, oxygen, argon, and water vapor, the molecule scattering calculation can be performed for each individual gas type and molecule size using the Rayleigh theory (Young, 1981). 20 This becomes relevant if inelastic scattering or molecular absorption is involved. For ACL systems provided that a wavelength is used, which is well outside of molecular absorption lines, the individual gas contribution to the signal does not need to be distinguished.

Consequently, the molecule extinction coefficient $\alpha_{\text{mol},\lambda}(z)$ and the molecule backscatter coefficient $\beta_{\text{mol},\lambda}(z)$ can be calculated with

25   $$\alpha_{\text{mol},\lambda}(z) = N_{\text{mol},\lambda}(z)\, \sigma_{\text{sca},\text{mol},\lambda}, \tag{7}$$

$$\beta_{\text{mol},\lambda}(z) = N_{\text{mol},\lambda}(z) \left(\frac{d\sigma_{\text{sca},\text{mol},\lambda}}{d\Omega}\right)_\pi. \tag{8}$$





The molecule number density $N_{\mathrm{mol}}(z)$ is related to the ideal gas law

$$N_{\mathrm{mol}}(z) = \frac{p(z)}{k\,T(z)}, \tag{9}$$

where $p$ is the model prediction of the atmospheric pressure given in Pascal (Pa), $T$ is temperature given in Kelvin (K), and $k$ is the Boltzmann constant which has a value of about $1.381 \times 10^{-23}\,\mathrm{J\,K^{-1}}$.

To calculate the scattering cross-section $\sigma_{\mathrm{sca,mol},\lambda}$ and the scattering phase function $\phi_{a,\lambda}(\theta)$ of air, empirical equations are available. We used the formulas given by (Buchholtz, 1995) who derived empirical equations for "standard air" and wavelengths ranging from $\lambda = 200\,\mathrm{nm}$ to $\lambda = 1000\,\mathrm{nm}$. This allows us to calculate the scattering properties of air molecules $\sigma_{\mathrm{sca,mol},\lambda}$ and $\left(\frac{d\sigma_{\mathrm{sca,mol},i,\lambda}}{d\Omega}\right)_{\pi}$ directly from model output data and the ACL laser wavelength $\lambda$.

### 2.2.3 Scattering by Particles

The scattering characteristics of spheres with sizes not much smaller or larger than the wavelength are described by Mie's solution of the Maxwell equations (Mie, 1908; Wiscombe, 1980). Methods like the T-matrix (Mishchenko et al., 2002) or the discrete dipole approximation (DDA, Draine and Flatau (1994)) allow for scattering calculations of non-spherical objects; again, with sizes not much smaller or larger than the wavelength. The T-matrix algorithm is a tool for computing scattering by single and compounded particles (Mishchenko et al., 2002). It is faster than DDA but limited to rotationally symmetric objects such as ellipsoids, cylinders or Chebyshev polynomials. DDA, however, can represent arbitrarily shaped objects at the cost of high computational efforts.

As a rough estimate from our studies, the computational costs increase by about one order of magnitude when using the T-matrix instead of the Mie approach and by another two orders of magnitude when using the DDA instead of the T-matrix approach. Another increase in computational time is resulting from larger scatterers, i. e., an increase of the particle size causes an exponential increase of computing time. We therefore utilize Mie scattering algorithms to perform fast calculations although solid particles are in fact non-spherical. The effect of scattering by non-spherical particles is analyzed in a second step using the T-matrix approach for sensitivity studies. We consider these approaches as sufficient for the sensitivity studies performed in this work.

Mie-scattering related computations were performed using the IDL procedure "mie_single", provided by the Department of Atmospheric, Oceanic and Planetary Physics (AOPP), University of Oxford. Input parameters of the procedure are the real part $m$ and imaginary part $m'$ of the refractive index as well as the so-called size parameter $X_\lambda(R)$:

$$X_\lambda(R) = \frac{2\pi R}{\lambda}, \tag{10}$$

where $R$ is the radius of a single particle. The relevant output parameters are the extinction efficiency $Q_{\mathrm{ext},p,\lambda}(R)$, the scattering efficiency $Q_{\mathrm{sca},p,\lambda}(R)$, and the backscatter efficiency $Q_{\mathrm{bsc},p,\lambda}(R)$. These optical efficiencies are defined as ratio between the optical cross section and the physical cross section:

$$Q_{\mathrm{ext},p,\lambda}(R) = \frac{\sigma_{\mathrm{ext},p,\lambda}(R)}{\pi R^2}, \tag{11}$$





$$Q_{\mathrm{sca},p,\lambda}(R) = \frac{\sigma_{\mathrm{sca},p,\lambda}(R)}{\pi R^2}, \tag{12}$$

$$Q_{\mathrm{bsc},p,\lambda}(R) = \frac{\left(\frac{d\sigma_{\mathrm{sca},p,\lambda}(R)}{d\Omega}\right)_{\pi}}{\pi R^2}. \tag{13}$$

As a warning, we like to point out that the procedure changed its definition of the backscatter efficiency: The current (2012) release of mie_single returns the so-called radar backscatter efficiency which is $4\pi$ times the backscatter efficiency as we require it within the forward operator. Furthermore, the procedure expects the imaginary part of the refractive index given as negative number. If positive imaginary part values are used, the procedure runs without showing an error but returns wrong results.

### 2.2.4 Discrete Particle Number Size Distributions

A major problem of discrete size distributions is the high sensitivity of the optical cross sections to the particle size: A slightly different particle radius may lead to quite a large change of the scattering properties. We present in the following a suggestion to overcome this fundamental problem. Due to the fact that naturally occurring particle size distributions are not discrete, averaging the optical cross sections over certain size-intervals seems straightforward. We will show that this approach indeed reduces the problematic and unrealistic sensitivity significantly. If the model represents only one type of particle, i. e. with a constant refractive index but with discrete radii $R_d$, we can define the effective extinction cross section and the effective backscatter cross sections with

$$\overline{\sigma_{\mathrm{ext},R_d,m,m',\lambda}} = \frac{1}{R_{d_b} - R_{d_a}} \int_{R_{d_a}}^{R_{d_b}} Q_{\mathrm{ext}}(X_\lambda(R_d),m,m')\,\pi R_d^2\,dR_d, \tag{14}$$

$$\overline{\sigma_{\mathrm{bsc},R_d,m,m',\lambda}} = \frac{1}{R_{d_b} - R_{d_a}} \int_{R_{d_a}}^{R_{d_b}} Q_{\mathrm{bsc}}(X_\lambda(R_d),m,m')\,\pi R_d^2\,dR_d, \tag{15}$$

where $R_{d_a}$ and $R_{d_b}$ are size margins for each particle size class $d$. These integrals are then exchanged by sums in the numerical computation routines giving

$$\overline{\sigma_{\mathrm{ext},R_d,m,m',\lambda}} = \frac{1}{n_{\mathrm{samples}}} \sum_{g=1}^{n_{\mathrm{samples}}} Q_{\mathrm{ext}}(X_\lambda(R_{d_g}),m,m')\,\pi R_{d_g}^2, \tag{16}$$

$$\overline{\sigma_{\mathrm{bsc},R_d,m,m',\lambda}} = \frac{1}{n_{\mathrm{samples}}} \sum_{g=1}^{n_{\mathrm{samples}}} Q_{\mathrm{bsc}}(X_\lambda(R_{d_g}),m,m')\,\pi R_{d_g}^2, \tag{17}$$





where $n_{\text{samples}}$ is the sampling number and the sampling range $R_{d_b} - R_{d_a}$ is broken down in $g$ subsamples:

$$R_{d_g} = g \, \frac{R_{d_b} - R_{d_a}}{n_{\text{samples}}} + R_{d_a}. \tag{18}$$

This calculation of the effective values is performed for every discrete size class $d$ and - if represented by the model - also for every particle type $k$.

Consequently, the total particle extinction coefficient $\alpha_{\text{par},\lambda}(z)$ and the total particle backscatter coefficient $\beta_{\text{par},\lambda}(z)$ are calculated from:

$$\alpha_{\text{par},\lambda}(z) = \sum_k \sum_d N_{d,k}(z) \, \overline{\sigma_{\text{ext},R_d,m_k,m'_k,\lambda}}, \tag{19}$$

$$\beta_{\text{par},\lambda}(z) = \sum_k \sum_d N_{d,k}(z) \, \overline{\sigma_{\text{bsc},R_d,m_k,m'_k,\lambda}}. \tag{20}$$

Here, $N_{d,k}$ is the particle number per volume given by the model, $\overline{\sigma_{\text{ext},R_d,m_k,m'_k,\lambda}}$ and $\overline{\sigma_{\text{bsc},R_d,m_k,m'_k,\lambda}}$ are the effective optical cross sections of particle size class $d$ and particle type class $k$ with the respective real part $m_k$ and imaginary part $m'_k$ of the refractive index.

This simple solution allows for calculating $\alpha_{\text{par},\lambda}(z)$ and $\beta_{\text{par},\lambda}(z)$ by just solving a few multiplications and summations resulting in a minimal demand of computing time.

### 2.2.5 Lidar Ratio and Two-Way Transmission

The forward modeled total extinction coefficient and total backscatter coefficient are the sum of the molecule and the particle extinction and backscatter coefficients:

$$\alpha_\lambda(z) = \alpha_{\text{mol},\lambda}(z) + \alpha_{\text{par},\lambda}(z), \tag{21}$$

$$\beta_\lambda(z) = \beta_{\text{mol},\lambda}(z) + \beta_{\text{par},\lambda}(z), \tag{22}$$

equivalent to Eq. (3) and (4). The lidar ratio $S_{\text{lidar}}(z)$ is calculated by

$$S_{\text{lidar}}(z) = \frac{\alpha_{\text{par},\lambda}(z)}{\beta_{\text{par},\lambda}(z)}. \tag{23}$$

Even if the lidar ratio is not measured directly by current ACL systems, the skill of simulating the lidar ratio for given scatterer types and scatterer mixtures offers a great potential for sensitivity studies but also for applications to research lidar systems that are capable of measuring this parameter like a Raman lidar.





$T_\lambda$ describes the two-way transmission

$$T_\lambda(z) = \exp\left(-2\int_0^z \alpha_\lambda(z')\,dz'\right),\tag{24}$$

while $T_\lambda$ depends on $\alpha_\lambda$ between the instrument ($z = 0$) and $z$. Within the forward operator, the two-way transmission is discretized by:

$$T_\lambda(z) = \exp\left(-2\sum_{z=1}^{n_z(z)} \alpha_\lambda(z)\,\Delta h(z)\,\cos\Theta\right),\tag{25}$$

where the number of height levels between ground and the actual height is expressed by $n_z(z)$, the profile of the extinction coefficient is $\alpha_\lambda(z)$ and the profile of layer thicknesses is $\Delta h(z)\cos\Theta$ with $\Theta$ as zenith angle. In the case of forward modeling, a vertically pointing measurement ($\Theta = 0$), the vertical column of model grid boxes at a specific location is required. By changing the zenith angle $\Theta$, the forward operator can also simulate scanning measurements.

Conclusively, our implementation of a backscatter lidar forward operator allows for an efficient calculation of ACL profiles based on the output of atmospheric chemistry models. The biggest challenge for setting up the forward operator in a given scenario are scattering calculations, namely the calculation of the effective extinction cross section $\overline{\sigma_{\text{ext},R_d,k,\lambda}}$ and the effective differential backscatter cross section $\overline{\sigma_{\text{bsc},R_d,k,\lambda}}$ of all represented particle size and type classes. Due to its importance, we dedicate Sect 4 to sensitivity studies of the optical cross sections regarding the particle size, refractive index, and shape. Prior to that, we introduce the case study for which the forward operator was applied and compared to ACL measurements.

# 3 Case Study

## 3.1 Selection and Description

In spring 2010, an eruption of the volcano Eyjafjallajökull in Iceland lead to a closure of the European air space for several days (Sandrini et al., 2014). Due to its impact on public transport and on the transport of goods, this case was extensively analyzed by scientists from many fields of research, resulting in a substantial knowledge base (see ACP special issue "Atmospheric implications of the volcanic eruptions of Eyjafjallajökull, Iceland 2010"). Ash layers were observed from a large set of measurement instruments, allowing for tracking the volcanic ash cloud movement over Europe (Gasteiger et al., 2011a; Zakšek et al., 2013; Mona et al., 2012; Dacre et al., 2013; Waquet et al., 2014). Using images from the geostationary instrument SEVIRI (Spinning Enhanced Visible and Infrared Imager) the spatial extend of the ash cloud and its movement could be tracked and compared to the measurement of ground-based instruments (see Fig. 1). From the synergy of the two measurement systems, signals of the ACL systems could be related to clouds or volcanic ash layers, respectively. The volcanic eruption case is also beneficial for our analysis due to the fact that the ash emission takes place at one geographic location within well-known time intervals. Other cases are also interesting but may lack a precise definition of the test scenario or respective model prediction capabilities, such as the representation of background aerosol or uncertainties of the background aerosol number density.



## 3.2 The DWD ACL Network

ACL networks are the current data source of choice for an analysis of the vertical structure of aerosols. In 2010, the DWD ACL network consisted of 36 systems CHM15k ACL manufactured by Jenoptik now Lufft (see Fig. 2). A qualitative analysis of the ash cloud over Germany with this data set was made by Flentje et al. (2010b).

We used the NetCDF files with ACL raw data where one file contains the 24 hour measurement of one ACL station. The received photon number per shot is calculated from

$$N_{\text{rec}}(z,t) = \text{beta\_raw}(z,t) \cdot \text{stddev}(t) + \text{base}(t), \tag{26}$$

where $\text{beta\_raw}$ is the signal-to-noise measurement product, $\text{stddev}$ is noise, and $\text{base}$ is a daylight correction provided by the ACL software of this model. The calculation of the attenuated backscatter coefficient from ACL data follows Eq. (27) which reads

$$\gamma_\lambda^*(z) = \frac{N_{\text{rec},\lambda}(z,t)\, z^2}{N_{\text{tr},\lambda}\, \eta_\lambda\, A_{\text{tel}}\, O(z)\, \Delta z}. \tag{27}$$

The emitted photon number per shot $N_{\text{tr},\lambda}$ was calculated using:

$$N_{\text{tr},\lambda} = \frac{E_{\text{pulse},\lambda}}{E_{p,\lambda}}, \tag{28}$$

where $E_{\text{pulse},\lambda}$ is the laser pulse energy. $E_{p,\lambda}$ is the photon energy, calculated according to:

$$E_{p,\lambda} = \frac{h\,c}{\lambda}, \tag{29}$$

with $h$ as Planck's constant having a value of $6.62607 \times 10^{-34}\,\text{J}\,\text{s}$. Assuming that the pulse energy of $8\,\mu\text{J}$ given by Flentje et al. (2010c) is true and a constant, the emitted photon number per pulse is about $4.2821 \times 10^{13}$. The diameter of the receiving telescope is $100\,\text{mm}$ (Flentje et al., 2010c) which results in $A_{\text{tel}} = 78.5398\,\text{cm}^2$. The vertical resolution $\Delta z$ is $15\,\text{m}$ over the complete profile. The overlap function $O(z)$ was set to 1 which implies that ranges below about $1500\,\text{m}$ cannot be used reliably for comparisons with the forward operator.

Unfortunately, the instruments provided no calibrated measurement data at that time. As the true system efficiency $\eta_\lambda$ and the calibration coefficients are not known, we use the symbol $\eta^*$ as linear calibration factor. From a comparison with the calibrated attenuated backscatter measurements of CALIOP at $\lambda = 1064\,\text{nm}$ (Fig. 3), we determined a calibration factor of $\eta^* = 0.003$. It must be noted that the precision of this calibration attempt is rather limited due to the fact that the ACL instruments are ground based and CALIOP measures from space.

As a last step, the high-resolution ACL data was gridded to the model's vertical resolution as well as over $15\,\text{min}$ by temporal averaging. This also improved the signal-to-noise ratio of the ACL data.



### 3.3 Ash Transport Simulation of COSMO-ART

COSMO-ART was set up by the DWD in collaboration with Karlsruhe Institute of Technology (KIT) for an ash-dispersion simulation of the volcanic emissions during the eruptive phase of Eyjafjallajökull in spring 2010 (Vogel et al., 2014b). The model domain had a horizontal resolution of 7 km and 40 height layers. The height layers have a variable thickness ranging

from several meters near ground to a layer thickness of about $3\,\mathrm{km}$ in $22\,\mathrm{km}$ height above ground level. The spatial extend and model set-up is equal to the COSMO-EU domain as it is used at DWD for operational weather forecasting. A more general description of the model run is given by Vogel et al. (2014a). For this study, we used the 78-hour forecast beginning at 15 April 2010, 00:00 UTC, which includes volcanic ash emission data since 14 April 2010, 06:00 UTC.

    Volcanic ash was represented by six discrete size classes with a spherical shape and aerodynamic diameters of $1\,\mu\mathrm{m}$, $3\,\mu\mathrm{m}$,

$5\,\mu\mathrm{m}$, $10\,\mu\mathrm{m}$, $15\,\mu\mathrm{m}$ and $30\,\mu\mathrm{m}$. For each class, a number concentration was predicted by the model. Particles within a class were treated as equal, i. e., they had the same size, shape, and complex index of refraction (monodisperse classes). We therefore calculated effective optical cross sections as explained in Section 2.2.4. The lower and upper size margins $R_{d_a}$ and $R_{d_b}$ were defined as the arithmetic average between two subsequent size classes. The lower margin of the smallest size class was half its nominal diameter. The upper margin of the largest size class was 1.5 times its nominal diameter. The resulting class ranges are

shown in Fig. 4.

    A list of model variables used for the forward operator is given in Table 1. While the p number density is taken directly from model output, the molecule number density of standard air was calculated from temperature and pressure according to Eq. (9).

### 3.4 Volcanic Ash Properties

A detailed analysis of the emitted ash was performed by Schumann et al. (2010) who compared measurements made from

DLR's Falcon 20 aircraft with the data of some German research lidar systems. Ash samples were taken in-situ, analyzed using a scanning electron microscope, and assigned to matter groups. From the matter components, the complex index of refraction was calculated.

    Schumann et al. (2010) stated that the real part of the refractive index was between 1.53 and 1.60 at a wavelength of $\lambda = 630\,\mathrm{nm}$ and between $1.50$ and $1.56$ at a wavelength of $\lambda = 2000\,\mathrm{nm}$. The respective imaginary part was ranging from $-0.001\,\mathrm{i}$

to $-0.004\,\mathrm{i}$ at a wavelength of $\lambda = 630\,\mathrm{nm}$ and from $-2.0 \times 10^{-6}\,\mathrm{i}$ to $-40.0 \times 10^{-6}\,\mathrm{i}$ at a wavelength of $\lambda = 2000\,\mathrm{nm}$.

    Electron microscope images from the same study revealed that the volcanic ash particles were sharp-edged with a complex and asymmetric shape. The average asymmetry factor was 1.8 for small particles ($<0.5\,\mu\mathrm{m}$) and 2.0 of larger particles (Schumann et al., 2010). Electron microscope measurements of Rocha-Lima et al. (2014) showed that the asymmetry factor of the volcanic ash fine fraction has a value between 1.2 and 1.8.

The particle growth due to hygroscopic water coating was quantified to about 2 to 5 % at a relative humidity of 90 % (Lathem et al., 2011). A growth of 5 % does not change the scattering properties significantly in relation to the size averaging we perform.



Nevertheless, each measurement contains an error and the ash properties may change during its travel through the atmosphere. It is therefore straightforward to analyze the maximum error due to variations of the volcanic ash properties, namely the particle size, refractive index, and shape.

## 4 Sensitivity Studies

The representation of the particles by the model is clearly simplifying, so we must study the effect of these simplifications on the scattering of laser light when developing a forward operator. For a lidar forward model, sensitivities of the backscatter cross section are critical because the received signal intensity is linearly coupled to the backscatter cross section and, consequently, to the attenuated backscatter coefficient.

Prior studies already showed the complexity of non-spherical scattering calculations but there is no universal solution to the problem available. Gasteiger et al. (2011b) use Discrete Dipole Approximation (DDA) to calculate the scattering properties of complex shaped particles but the analysis is limited to size-parameters up to 20.8 (which results in a maximum particle radius of about $3.2\,\mu m$ at a wavelength of $1064\,nm$). A study of Kemppinen et al. (2015) is focused on individual ellipsoids but we neither know the ellipsoidal properties nor does COSMO-ART predict changes of the volcanic ash classes. Guessing an ellipsoidal distribution for the model predictions thus may lead to less realistic scattering calculation results than assuming spherical scatterers. Consequently, there is no sufficient scattering description for Eyjafjallajökull ash predictions of COSMO-ART available. We therefore treat the volcanic ash as spherical objects with given optical properties (see 3.4) but analyze and discuss the effect of uncertain volcanic ash properties in the following.

### 4.1 Prerequisites

Look-up tables (LUT) of Mie efficiencies and optical cross sections have been created to reduce the effort on time-consuming scattering calculations. The look up tables have three dimensions: size parameter $X_\lambda(R_p)$, real part of the refractive index $m$ and imaginary part of the refractive index $m'$.

The reasonable range of size parameters depends on the wavelength of typical lidar instruments and the radius of occurring particles $R_p$. From Eq. (10), for the ACL systems operating at $\lambda = 1064\,nm$, we find a size-parameter range of 1.18 ($R_p = 0.2\,\mu m$) and 142 ($R = 24.2\,\mu m$). We selected a radius increment of $0.024\,\mu m$.

The refractive index measurements by Schumann et al. (2010) were not performed for the wavelength of the ACL systems as explained in section 3.4. Therefore, we have to assume the refractive index we use as reference as well as an interval of uncertainty. Schumann et al. (2010) take a refractive index of 1.59 - 0.004i for their medium "M" case study and therefore, we also used this value as reference for our study. Our uncertainty intervals of real and imaginary parts were chosen according to the range of measured values at $630\,nm$ and $2000\,nm$, namely a real part range of 1.54 to 1.64 and an imaginary part range of -0.006 to -0.002. To get an estimate of the overall refractive index sensitivity for such particles, we decided to extend the range of analyzed refractive indices to real parts between 1.49 and 1.69 using increments of 0.001 and to imaginary parts





between -0.011 and -0.001 using increments of 0.00005. Consequently, the total element number of one LUT is $4.0 \times 10^7$. These look-up tables were the base for the subsequent sensitivity studies.

### 4.2 Sensitivity to the Complex Index of Refraction

Fig. 5 and Fig. 6 show $\sigma_{\text{ext}}$ and $\sigma_{\text{bsc}}$ as well as $\overline{\sigma_{\text{ext}}}$ and $\overline{\sigma_{\text{bsc}}}$ against the real and imaginary parts of the complex index of refraction. While the extinction cross section $\sigma_{\text{ext}}$ is more sensitive to the real part than to the imaginary part of the refractive index, the backscatter cross section $\sigma_{\text{bsc}}$ is strongly sensitive to both. These sensitivities are strongly reduced for the effective extinction cross section $\overline{\sigma_{\text{ext}}}$ and the effective backscatter cross section $\overline{\sigma_{\text{bsc}}}$.

An overview of the refractive index sensitivity of the effective optical cross sections is given by Fig. 7 which shows the relative errors

$$\sigma_{\text{ext,err},p}(m, m') = \frac{\overline{\sigma_{\text{ext},p}(m, m')} - \overline{\sigma_{\text{ext},p}(m^*, m'^*)}}{\overline{\sigma_{\text{ext},p}(m^*, m'^*)}} \cdot 100\%, \tag{30}$$

and

$$\sigma_{\text{bsc,err},p}(m, m') = \frac{\overline{\sigma_{\text{bsc},p}(m, m')} - \overline{\sigma_{\text{bsc},p}(m^*, m'^*)}}{\overline{\sigma_{\text{bsc},p}(m^*, m'^*)}} \cdot 100\%. \tag{31}$$

The relative error is the error of the optical cross sections if we assume that the refractive index we defined as reference ($m^*$ and $m'^*$) to be true but the real particles have a refractive index of $m$ and $m'$. We can conclude from this analysis that the maximum relative error for the given range of refractive indices is less than 10 % for the extinction cross section but ranges up to 230% for the backscatter cross section.

### 4.3 T-matrix Particle Shape Sensitivity Study

T-matrix calculations within this study are based on the FORTRAN code for randomly oriented particles written and provided by Mishchenko and Travis (1998). A detailed description of the method can be found in the work of Mishchenko et al. (2002). We modified the double-precision version of the T-matrix procedure to perform scattering calculations of multiple particle sizes automatically. In addition to that, the procedure was extended by the calculation of the backscatter cross section $\sigma_{\text{bsc}}$ according to Mishchenko et al. (2002), Eq. (9.10). Then, as a test, both mie_single and our modified T-matrix code were set up to calculate the scattering properties of the same spheres and for the same wavelength. The results of both procedures were indeed identical.

A list of T-matrix options we used for the particle shape sensitivity study is shown in Table 2. The most important particle properties are defined by the variables NP and EPS. NP defines the particle type and has a value of -1 for spheres as well as for ellipsoids. A NP value of -2 is used for cylinders. The variable EPS is an expression for the objects' diameter to length ratio. Consequently, an ellipsoid with EPS=1 is a sphere, prolate objects have an EPS<1 and oblate objects have an EPS>1.

In Figs 8, 9, 10,11, 12, and 13, the optical cross sections of spheres and several aspherical particles are plotted against the equal-volume radius. The first group of aspherical scatterers are ellipsoids with an diameter-to-length-ratio of 0.667, 0.8, 1.25





and 1.50. We found no significant differences of the extinction cross section of spheres and these ellipsoids. However, we find huge differences regarding the backscatter cross section. While the spectrum of spheres is strongly oscillating (due to constructive and destructive interference effects) even the slightly distorted ellipsoids seem to be independent of such effects. Regarding the comparison between spheres and cylinders with EPS values of 0.8, 1.0 and 1.25, the extinction cross section of the three types of cylinders tends to be higher than the extinction cross section of spheres. The backscatter cross section spectrum of cylinders and ellipsoids seems to be independent of the interference effects observed for spheres. The backscatter cross sections of cylinders are partly lower and partly higher than the backscatter cross sections of spheres.

A summary of the particle shape sensitivity study is shown in Figs 14 and 15 giving the relative differences of the effective optical cross sections for different particle shapes. The definition of the relative differences follows Eq. (30) and Eq. (31). Again, the reference is a spherical particle. Positive and negative relative differences indicate that the effective optical cross sections of spheres are underestimations and overestimations, respectively.

The effective extinction cross section of spheres is lower than the effective extinction cross section of any of the analyzed asymmetric particle. Regarding the effective backscatter cross section, however, we find relative differences of up to 286% and $-75\%$. While the small aspherical particles have a lower effective backscatter cross section compared to spheres, the effective values of the fourth size class are higher for almost all considered aspherical particles.

Averaging the relative differences of all particle shapes allows for the calculation of the mean differences. The mean differences of the effective extinction cross section reach up to 5% which means that spheres have a lower extinction than comparable aspherical objects. For the effective backscatter cross section, the mean difference is between 110% for large scatterers and $-60\,\%$ for small scatterers.

Of course, the total effect of different particle shapes for a real aerosol mixture has to be estimated with more extensive scattering calculations. The above results, however, give us valuable insight into the uncertainties of using Mie calculations for non-spherical particles. The particle shape sensitivity is reduced by size- and shape-averaging methods but the resulting effective backscatter cross section of the mixture may vary by a factor of $\pm 2$.

## 5 Results

### 5.1 Scattering Properties of Volcanic Ash Used Within the Forward Operator

A list of the effective extinction cross section and the effective backscatter cross section we determined for atmospheric gas molecules and for the six volcanic ash size classes is shown in Table 3.

### 5.2 Output Variables of the Forward Operator

Using the forward operator allows for plotting each variable of the lidar simulation for analytic purposes (see Fig. 17). These plots of forward-operator output variables are representing the major characteristics of the variables: strong extinction and strong backscattering are usually related. Time and height intervals at which only molecules exist, lead to low values of the





extinction coefficient and backscatter coefficient. Due to the decrease of the atmospheric gas number density with height, both extinction and backscatter coefficient decrease with height in an aerosol-free atmosphere. The two-way transmission decreases with height (see Eq. (24)).

The forward operator yields a lidar ratio for volcanic ash of between $15\,\mathrm{sr}$ and $80\,\mathrm{sr}$ which differs to lidar ratio values we find in literature ($40\,\mathrm{sr}$ to $100\,\mathrm{sr}$ for volcanic ash (Kokkalis et al., 2013; Ansmann et al., 2010; Mortier et al., 2013)). This difference may be a consequence of assuming spherical particles. But also for spherical particles we would expect the given lidar ratio as the extinction cross section increases exponentially with the particle size while the backscatter cross section is influenced by interference effects (see. Fig. 8 and Fig. 9). This is also represented by the effective optical cross sections which are used by the forward operator (see. Table. 3). Comparing these values with the lidar ratio findings by Gasteiger et al. (2011b), a lidar ratio of less than $20\,\mathrm{sr}^{-1}$ seems to be plausible. The authors found even for irregularly shaped objects a lidar ratio between $5\,\mathrm{sr}^{-1}$ and $20\,\mathrm{sr}^{-1}$ for size parameters between 5 and 15 (equivalent particle diameter at $\lambda = 1064\,\mathrm{nm}$ would be $1.6\,\mathrm{\mu m}$ and $4.8\,\mathrm{\mu m}$, respectively). Due to the different refractive indices and particle shapes, this comparison is no validation for our findings but shows that the result of our analysis is close to the reference calculations.

An analysis of the volcanic ash contribution to the total signal and the total mass density for each size class of COSMO-ART is shown in Fig. 16 for two cases. The cases were selected to cover two extreme situations: Case 1 is the model output from a coordinate inside the volcanic ash layer (Table 4). Case 2 is for of a coordinate where the highest values of the lidar ratio are determined by the forward operator (Table 5). Regarding case 1, the total backscatter coefficient is dominated by ash size classes 1, 2 and 3 while the signal contribution of classes 4 to 6 is less than $5\,\%$ in total. The mass contribution is dominated by classes 3 and 4 while classes 2, 5 and 6 are contributing by $10\,\%$ each to the total mass density and class 1 is nearby irrelevant for the total mass density. Regarding case 2, the total backscatter coefficient depends by about $68\,\%$ from class 4 and by $30\,\%$ from class 6. High values of the lidar ratio thus seem to be related to the number density of class 4 which was already found to be affected by destructive interference effects (as shown for example in Fig. 9). The mass contribution in case 2 is also dominated by the classes 4 and 6 but, in contrast to the backscatter coefficient, class 6 has a higher contribution to the total mass density than class 4. General conclusions from this analysis about the relationship between backscattering and mass depending on particle size and wavelength require further investigation. For our application of the forward operator in this study, however, we can conclude the following: First, the total signal inside the volcanic ash layer (case 1) is predominately dependent on classes 1, 2 and 3 whose backscatter cross sections are also overestimated by the forward operator due to the assumption of sphericity (see Fig. 15). Second, the larger particles of classes 4, 5 and 6 carry a large portion of the mass but contribute only weakly to the total signal.

## 5.3 Qualitative Comparison

A qualitative comparison allows for the identification of common and different structures between the measured and simulated lidar profiles. Different ash layer structures can hint, e.g., to errors in the model dynamics, in the source description, or in the sedimentation parametrization. If ash structures are found only in the measured profiles, either the model prediction is wrong or it misses an important aerosol type which is not present in the model. If structures are visible in the forward modeled profiles





but missing in the measured profiles, either the ACL signal is too weak because of high extinction in lower heights or the model performed a wrong ash prediction.

For a qualitative comparison between measurement and simulation, we chose the ACL station Deuselbach in West Germany and a time interval from 16 to 17 April 2010 as an example. Here, the ash layer was clearly visible in the measured profiles without being affected by low-level or high-level clouds. We calculated the attenuated backscatter coefficient from the ACL measurement according to Eq. (27), extracted the common time and height intervals and re-sampled the ACL data to the model resolution.

A comparison of ACL measurement and COSMO-ART simulation with an applied forward operator is shown in Fig. 18. Due to the inevitable instrumental noise, the automatic calibration of ACL system and subsequent background subtraction, some data points become negative which is just a statistical effect but causes missing data in the log-scale plots. Volcanic ash plumes are clearly visible on both plots. Looking at the forward operator result, the ash layer begins to cross the ACL station between 06:00 UTC and 12:00 UTC at 16 April 2010. The layer height decreases with time and partially entrains into the planetary boundary layer where it persists even at the end of 17 April 2010. As both model and forward operator only represent volcanic ash and air molecules, the ash layers can be tracked within the planetary boundary layer. This is not possible using ACL measurements alone as the volcanic ash signal is tainted by other aerosol types here. It is, however, rather difficult to determine unambiguously which ash layer structure observed by the ACL instrument can be related to the appropriate structures simulated by the model. Regarding the thin volcanic ash layer which is measured by the ACL instrument in a height between 7 and 9 km ASL at the 16th of April, around 06:00 UTC, this feature could be equivalent to the model prediction of ash in a height of 6 km ASL at 7:00 UTC. In this case, the model would have performed a rather precise prediction with only one hour time lag and a two km vertical shift. But it is also possible that the predicted ash entrainment over the ACL station is equivalent to the ash-indicating ACL signals at around 12:00 UTC. In the latter case, the model prediction would be wrong by a time lag of about 6 hours which is insufficient for time-critical applications.

The qualitative comparison is currently limited to coordinates where the major fraction of scatterers are represented by both model and forward operator. There is, however, one scatterer fraction still missing on the present model runs for a comprehensive comparison: Other aerosol types than volcanic ash like anthropogenic emissions, mineral dust, soot, pollen, etc., are not included which leads to differences especially in the planetary boundary layer. We therefore cannot predict yet whether the strong ACL signal in the planetary boundary layer is related to background aerosol or errors of the model. To overcome this problem, continuous data assimilation of aerosols and the simulation of all aerosol particle sources in the model are required.

## 5.4 Quantitative comparison

A major purpose of the backscatter lidar forward operator is the capability to perform also quantitative comparisons of measurement and model output data. Unfortunately, such comparison is of limited validity in this case study due to the unknown ACL calibration as noted in Section 3.2.

Outside the volcanic ash layer, the forward operator returns an attenuated backscatter coefficient of $1 \times 10^{-7}\,\mathrm{m^{-1}\,sr^{-1}}$ which is equal to the values of the ACL instrument after calibration. This would be expected as both temperature and pressure



are rather precisely determinable and the scattering properties of air are represented by the empirical equations which are utilized by the forward operator. We therefore assume that the selected calibration factor is valid for this scenario.

Regarding the ash layer, however, the forward operator returns stronger signals inside the ash plume as well as a lower transmission behind the cloud compared to the ACL measurement. The maximum values of the attenuated backscatter coefficient returned by the forward operator (about $6.0 \times 10^{-4}\,\mathrm{m^{-1}\,sr^{-1}}$) is 60 times higher than the maximum values reported by the ACL (about $3.0 \times 10^{-5}\,\mathrm{m^{-1}\,sr^{-1}}$). The minimum attenuated backscatter coefficients calculated by the forward operator $(1.0 \times 10^{-9}\,\mathrm{m^{-1}\,sr^{-1}})$ are about 10 times lower than observable on the ACL plots $(1.0 \times 10^{-8}\,\mathrm{m^{-1}\,sr^{-1}})$, excepting pixels which became negative due to noise. As both backscattering and extinction are higher compared to ACL observations, the ash concentration seems to be overestimated by the model. We therefore analyzed the effect on the attenuated backscatter coefficient if the model-predicted ash number densities are being reduced by factors of 10, 20 and 30. If the ash number density is reduced by factor of 20 (Fig. 19), similar maximum values of the attenuated backscatter coefficient are observed inside the ash layer for both the forward operator and the ACL $(5.0 \times 10^{-5}\,\mathrm{m^{-1}\,sr^{-1}})$. This reduction of the number density also results in less extinction, and the two-way transmission has a minimum value of $70\,\%$ (plot not shown here) which is more realistic than the minimum value observed for the original dataset ($8\,\%$, see Fig. 17).

## 6 Conclusions

A backscatter lidar model capable of calculating both the extinction and backscatter coefficients was introduced. Detailed studies concerning the scattering properties of particles and molecules were performed within this study. Instead of assuming a lidar ratio for given particles, this forward operator allows for calculating the lidar ratio even for mixtures of different particle types which is an outstanding feature with many fields of application in the future.

The forward model was applied to the COSMO-ART model but the same approach can be used for any other aerosol-chemistry-transport model. We used data of a COSMO-ART ash-dispersion simulation to run the forward operator and perform both qualitative and quantitative comparison between the output of the forward operator and measurement data of an automated ceilometer lidar (ACL) system.

The atmospheric gas mixture was simplified to a uniform mixture of "atmospheric gas". Therefore, empirically determined scattering equations were used to calculate the optical cross sections of this mixture for the given laser wavelength. From the model-predicted values of temperature and pressure, the molecule number-density and finally the molecule extinction and backscatter coefficients were calculated.

Regarding particle scattering, extensive scattering calculations were performed to create look-up tables of optical cross sections. The range of particle sizes was selected according to the volcanic ash size classes used by COSMO-ART (six monodisperse classes with diameters of $1\,\mu\mathrm{m}$, $3\,\mu\mathrm{m}$, $5\,\mu\mathrm{m}$, $10\,\mu\mathrm{m}$, $15\,\mu\mathrm{m}$ and $30\,\mu\mathrm{m}$). The range of refractive indices were adapted according to in-situ measurements of Schumann et al. (2010). As reference shape, we assumed the volcanic ash to be spherical.

Due to uncertain refractive indices of the volcanic ash and due to the fact that volcanic ash particles are not spherical, sensitivity studies sections have been performed to analyze the consequences of uncertain information for the optical cross





sections. While the extinction cross section was only weakly sensitive to variable refractive indices and particle shapes, the backscatter cross section was strongly sensitive to both the refractive index as well as the particle shape. To reduce these sensitivities, we averaged over a range of particle sizes and defined effective optical cross sections for each size class.

Using these effective optical cross sections reduced the sensitivity of optical cross sections towards the refractive index as well as the particle shape. But even after averaging, the relative uncertainty of the effective backscatter cross section exceeds 280% for uncertain refractive indices. This study also indicates the dependency of the forward operator on precise information about the particle's refractive index. Within a particle shape sensitivity study, we found a relative uncertainty of 286% for the effective backscatter cross section if some types of non-spherical ash particles are treated as spherical objects by the forward operator. Assuming that volcanic ash consists of a mixture of particle shapes, we averaged the relative differences over 7 particle shapes (4 types of ellipsoids and 3 types of cylinders). The resulting relative difference of the effective backscatter cross sections was quantified to be lower than 110%. The respective relative difference of the effective extinction cross section for mixed particle shapes is less than 6%. These sensitivities demonstrate the importance of lidar backscatter data assimilation in chemistry models because these will strongly constrain and correct the model with respect to shape and refractive indices.

A comparison between ACL measurement and the model predictions used as input for the developed forward operator was shown. Similar structures were observed but some features were referenced to different time and height locations. From our analysis at the ACL station Deuselbach, some ash layer features were predicted quite precisely by the model, for example the time of arrival of the ash plume at about 06:00 UTC with a vertical shift of about $1.5\,\mathrm{km}$. Some other features, such as the intersection with the planetary boundary layer at 17 April 2010, 03:00 UTC, was simulated about 6 hours too early to 16 April 2010, 18:00 UTC. Fine structures of the ash layer were only observable in the simulation but not in the ACL measurements due to noise.

Due to unknown calibration coefficients of the ACL system, a calibration constant $\eta^*$ was estimated by comparing the ACL data with calibrated measurements at the same wavelength. This method is sufficient for testing the forward operator but future data assimilation applications it will be dependent on automatically calibrated ACL data. For this study, however, we found a reasonable calibration constant of $\eta^* = 0.003$. Using this calibration constant, quantitative comparisons between ACL and the forward operator output were performed. We found that the molecule signal of ACL and forward operator output were of the same order of magnitude which argues that the selected calibration factor was acceptable. A comparison the volcanic ash signal led to the conclusion that the model predicted ash concentration was to be too high as the forward modeled attenuated backscatter coefficient within ash layers was 60 times higher and after attenuation 10 times lower than observed by the ACL. If the model-predicted ash concentration is manually reduced by a factor of 20, the forward modeled COSMO-ART predictions and ACL measurements became quantitatively equivalent. Such a reduction could be part of a simple particle data assimilation system helping to calibrate particle dispersion simulations before in-situ measurements are available.

Furthermore, we analyzed the contribution of each class to the total backscatter coefficient and to the total mass density (Fig. 16) for two sample cases. Regarding case 1 inside the volcanic ash layer, the classes 1, 2 and 3 were mostly responsible (94.8 %) for the calculated attenuated backscatter coefficient. The same size classes are typically overestimated by the forward operator by factor 2 to 3 due to the assumption of sphericity as shown in Section 4.3. This implies that the attenuated backscatter





coefficient inside the volcanic ash layers are by trend overestimated which is contributing furthermore to the difference between forward modeled COSMO-ART predictions and the ACL measurements.

Assuming that the ACL calibration is valid, the model-predicted ash concentration was about 10 times higher than observable. There are, however, some error sources remaining which are: First, there are only molecules and the six volcanic

ash classes represented while background aerosol is missing completely. Second, the ACL calibration is of limited precision. Third, the contribution to the attenuated backscatter coefficient of ash size classes 4, 5 and 6 is negligible even though these classes carry a large proportion of the mass. This relationship could rely on the ACL's wavelength which probably limits its sensitivity to particles larger than about $10\,\mu m$ in diameter. This result strengthens the importance of a joint use of observations and model output in combination with data assimilation in order to get the best state of the atmosphere with respect to aerosol

distributions and properties.

Conclusively, we recommend further investigation in scattering calculations of non-spherical particles to get more realistic optical cross sections for the forward operator. A decrease of uncertainties related to the forward operator can be achieved by refractive index measurements at the exact ACL wavelength. Refractive index measurements are a basic aspect of the forward operator as the optical cross sections can only be calculated if the aerosols' refractive index is known precisely. The

model - and consequently the forward operator - has to represent more aerosol types, especially background aerosols, mineral dust, sea salt and soot as missing extinction near ground cause the forward operator to overestimate the signal from layers behind. But also qualitatively, more scatterer types are required to cover all atmospheric features measured by ACL systems. In the context of international and probably intercontinental ACL networks, the creation of a scattering database for aerosols would be desirable. A central database can increase the development rate and flexibility of current lidar forward operator

implementations. The ACL networks themselves are only useful for aerosol research and data assimilation if the calibration is performed automatically and transparent. For future lidar measurement networks, the number of high spectral resolution lidar (HSRL) systems and Raman lidar systems could potentially increase and allow for the assimilation of the extinction coefficient and the backscatter coefficient directly. As many ACL devices are operating a proprietary firmware, the manufacturers have to be sensitized to data quality and reproducible measurement calibration. Otherwise, the backscatter data is nearby useless

for any quantitative comparison or aerosol data assimilation approach. The uncertainties in both modeling and measurements, however, will also require sophisticated data assimilation algorithms not only for typical atmospheric variables but also for aerosol optical properties. Also a very good first guess of model simulations with respect to aerosol particles will be necessary so that more sources, types, and sinks will be to be included.

*Acknowledgements.* The present study was part of the research project 50.0356/2012 funded by the German Federal Ministry of Transport

and Digital Infrastructure (BMVI, prior BMVBS). We furthermore acknowledge the contributors to COSMO-ART, to the ceilometer network, to the IDL procedure mie_single and to the T-matrix codes we used as a basis for our study. We are also thankful for helpful discussions with Cristina Charlton-Perez, Werner Thomas, and Frank Wagner.





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



**Table 1.** Output variables of COSMO-ART used by the forward operator for the selected case study.

| Variable | Symbol | Description | Unit |
|---|---|---|---|
| $ASH1$ | $N_1$ | Ash number density of class 1 (1 µm) | m$^{-3}$ |
| $ASH2$ | $N_2$ | Ash number density of class 2 (3 µm) | m$^{-3}$ |
| $ASH3$ | $N_3$ | Ash number density of class 3 (5 µm) | m$^{-3}$ |
| $ASH4$ | $N_4$ | Ash number density of class 4 (10 µm) | m$^{-3}$ |
| $ASH5$ | $N_5$ | Ash number density of class 5 (15 µm) | m$^{-3}$ |
| $ASH6$ | $N_6$ | Ash number density of class 6 (30 µm) | m$^{-3}$ |
| $Pmain$ | $p$ | Atmospheric pressure | hPa |
| $T$ | $T$ | Atmospheric temperature | °C |

**Table 2.** Settings of the T-matrix procedure for the particle shape sensitivity study. The parameters were kept constant during the study except the particle shape parameters (EPS and NP)

| Variable | Value | Description |
|---|---|---|
| $RAT$ | 1 | Radius is given as equal-sphere-volume radius |
| $NPNAX$ | 1 | Setting for monodisperse distributions |
| $AXMAX$ | 1 | Setting for monodisperse distributions |
| $B$ | 1D-1 | Setting for monodisperse distribution |
| $NKMAX$ | -1 | Setting for monodisperse distributions |
| $NDISTR$ | 4 | Setting for monodisperse distributions |
| $EPS$ | 0.5 ... 1.5 | Aspect ratio of the scatterer |
| $NP$ | -1 or -2 | Selects the particle type (spheres NP=-1 or cylinders NP=-2) |
| $LAM$ | 1064.e-9 | Wavelength of incoming light |
| $MRR$ | 1.59 | Real part of the refractive index |
| $MRI$ | -0.004 | Imaginary part of the refractive index |
| $NPNA$ | 19 | Number of random angles |

**Table 3.** Effective optical cross sections and lidar ratio of atmospheric gas molecules and six volcanic ash size classes used by the forward operator in the Eyjafjallajökull case study.

| Scatterer Class | $\sigma_{\text{ext}}$ (m$^2$) | $\sigma_{\text{bsc}}$ (m$^2$ sr$^{-1}$) | $S_{\text{lidar}}$ (sr$^{-1}$) |
|---|---|---|---|
| Atmospheric Gas | $3.125 \times 10^{-32}$ | $3.680 \times 10^{-33}$ | 8.49 |
| Ash 1 (1 µm) | $4.324 \times 10^{-12}$ | $0.328 \times 10^{-12}$ | 13.18 |
| Ash 2 (3 µm) | $17.821 \times 10^{-12}$ | $3.843 \times 10^{-12}$ | 4.64 |
| Ash 3 (5 µm) | $61.672 \times 10^{-12}$ | $6.200 \times 10^{-12}$ | 9.95 |
| Ash 4 (10 µm) | $177.045 \times 10^{-12}$ | $5.365 \times 10^{-12}$ | 33.00 |
| Ash 5 (15 µm) | $526.967 \times 10^{-12}$ | $20.442 \times 10^{-12}$ | 25.78 |
| Ash 6 (30 µm) | $1937.387 \times 10^{-12}$ | $23.781 \times 10^{-12}$ | 81.47 |





**Table 4.** Point-data extraction of COMSO-ART output; case 1 from 16 April 2010, 18:00 UTC, in a height of $1.9\,\mathrm{km}$ ASL. Using the number density $N_d$ of volcanic ash class $d$, we calculated the individual backscatter coefficient $\beta_{\mathrm{par},d,\lambda}$, the contribution to the total backscatter coefficient $\sum \beta_{\mathrm{par},d,\lambda}$, the individual mass density $\rho_d$, and the contribution to the total mass density $\sum \rho_d$. Ash particles were treated as spherical objects with a volumetric mass density of $2500\,\mathrm{kg\,m^{-3}}$.

| $d$ | $N_d$ | $\beta_{\mathrm{par},d,\lambda}$ | $\frac{\beta_{\mathrm{par},d,\lambda}}{\sum \beta_{\mathrm{par},d,\lambda}}$ | $\rho_d$ | $\frac{\rho_d}{\sum \rho_d}$ |
|---|---|---|---|---|---|
| - | $\mathrm{m^{-3}}$ | $\mathrm{m^{-1}\,sr^{-1}}$ | - | $\mathrm{kg\,m^{-3}}$ | - |
| 1 | 43 653 522 | $1.4 \times 10^{-5}$ | 22.3 % | $0.57 \times 10^{-7}$ | 3.3 % |
| 2 | 7 044 794 | $2.7 \times 10^{-5}$ | 41.9 % | $2.49 \times 10^{-7}$ | 14.2 % |
| 3 | 3 194 338 | $2.0 \times 10^{-6}$ | 30.7 % | $5.23 \times 10^{-7}$ | 29.8 % |
| 4 | 462 402 | $2.5 \times 10^{-6}$ | 3.8 % | $6.05 \times 10^{-7}$ | 34.5 % |
| 5 | 37 161 | $7.6 \times 10^{-7}$ | 1.2 % | $1.64 \times 10^{-7}$ | 9.3 % |
| 6 | 4474 | $1.1 \times 10^{-7}$ | 0.2 % | $1.58 \times 10^{-7}$ | 9.0 % |

**Table 5.** The same as Table 4 but for case 2 at 16 April 2010, 09:00 UTC, in a height of $1.5\,\mathrm{km}$ ASL.

| $d$ | $N_d$ | $\beta_{\mathrm{par},d,\lambda}$ | $\frac{\beta_{\mathrm{par},d,\lambda}}{\sum \beta_{\mathrm{par},d,\lambda}}$ | $\rho_d$ | $\frac{\rho_d}{\sum \rho_d}$ |
|---|---|---|---|---|---|
| - | $\mathrm{m^{-3}}$ | $\mathrm{m^{-1}\,sr^{-1}}$ | - | $\mathrm{kg\,m^{-3}}$ | - |
| 1 | 93.0 | $30.7 \times 10^{-12}$ | 0.2 % | $<0.01 \times 10^{-9}$ | <0.1 % |
| 2 | 97.0 | $372.5 \times 10^{-12}$ | 2.8 % | $<0.01 \times 10^{-9}$ | <0.1 % |
| 3 | 1.0 | $6.2 \times 10^{-12}$ | <0.1 % | $<0.01 \times 10^{-9}$ | <0.1 % |
| 4 | 1700.0 | $9129.0 \times 10^{-12}$ | 67.3 % | $2.23 \times 10^{-9}$ | 27.1 % |
| 5 | 0.5 | $10.2 \times 10^{-12}$ | <0.1 % | $<0.01 \times 10^{-9}$ | <0.1 % |
| 6 | 169.0 | $4018.8 \times 10^{-12}$ | 29.6 % | $5.97 \times 10^{-9}$ | 72.8 % |



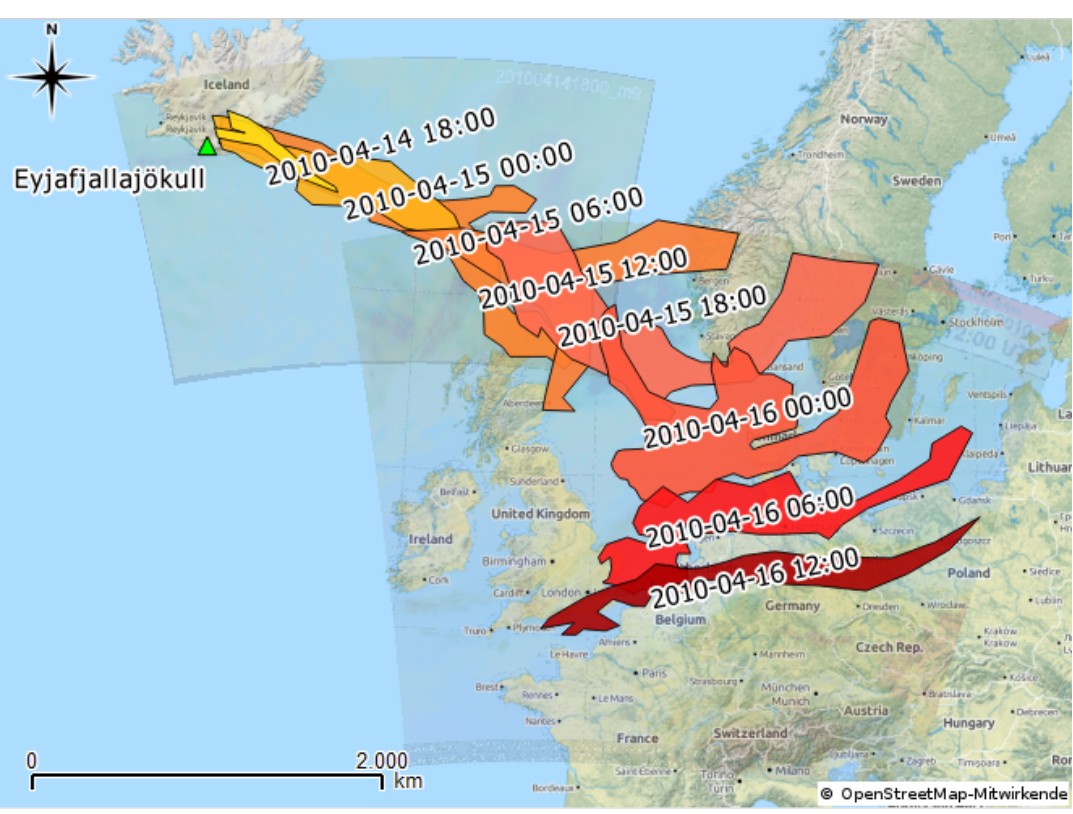

**Figure 1.** Distribution and transport of volcanic ash over northwest Europe sketched using georeferenced satellite images (Meteosat-9, Dust). After georeferencing, the ash layers were retraced as colored polygons where the color of the polygons (yellow to red) represent consecutive time steps (6 hour time steps).





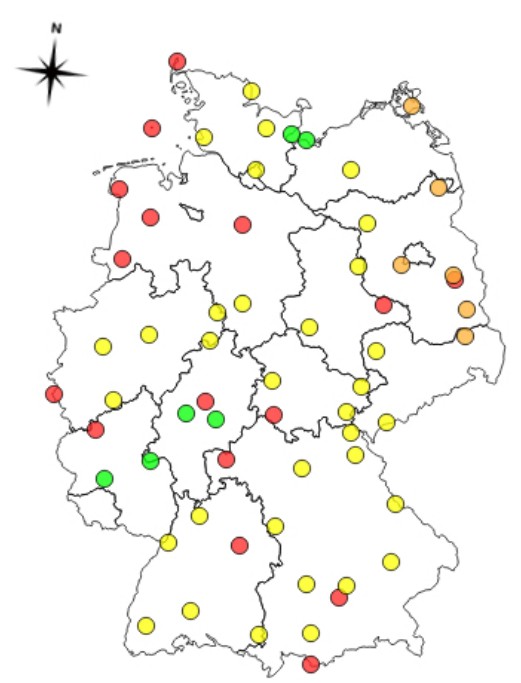

Geoinformation © Bundesamt für Kartographie und Geodäsie (www.bkg.bund.de)

**Figure 2.** The German ACL network in 2010. Each dot represents an ACL station and the dot color is an indicator for the ash layer visibility within the measurement. Red color: Near-ground fog or water clouds cover the ash cloud, orange: ash clouds are almost (yellow: partially) covered by fog or clouds, green: clean air situation with a full view on the ash layers.



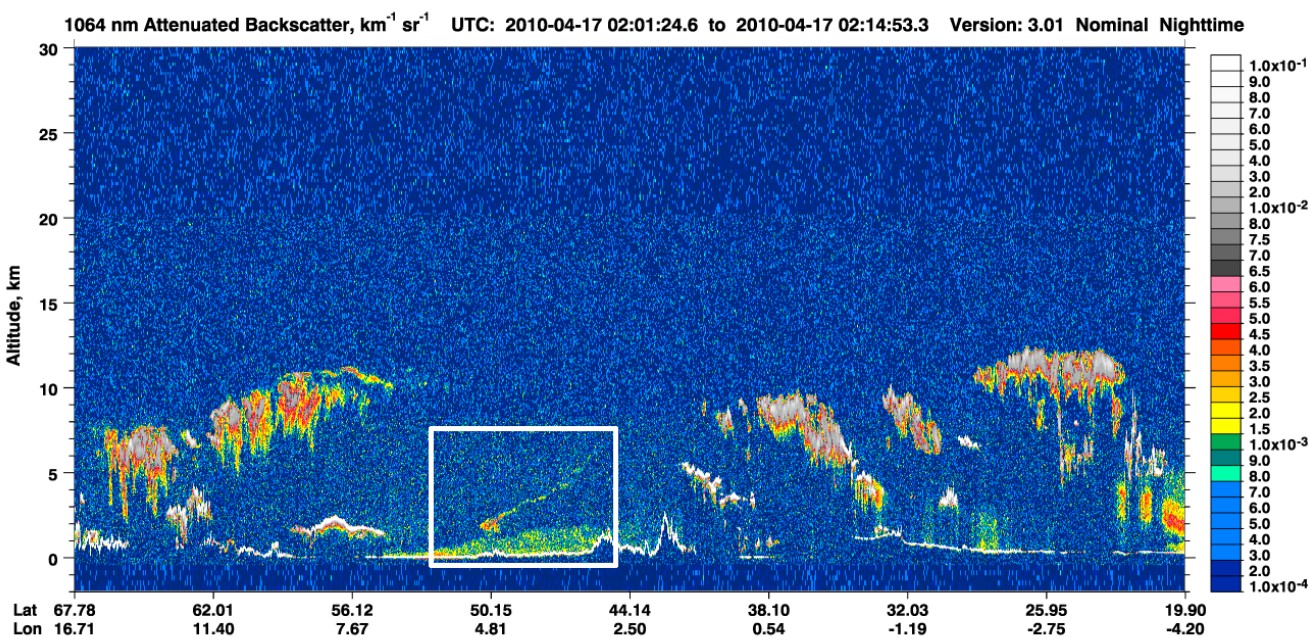

**Figure 3.** Attenuated backscatter coefficient measurement from CALIOP which was used to calibrate the ACL measurement during the Eyjafjallajökull eruption phase. The volcanic ash cloud is visible around 50.15° lat and 4.81° lon. Image obtained from http://www-calipso. larc.nasa.gov/





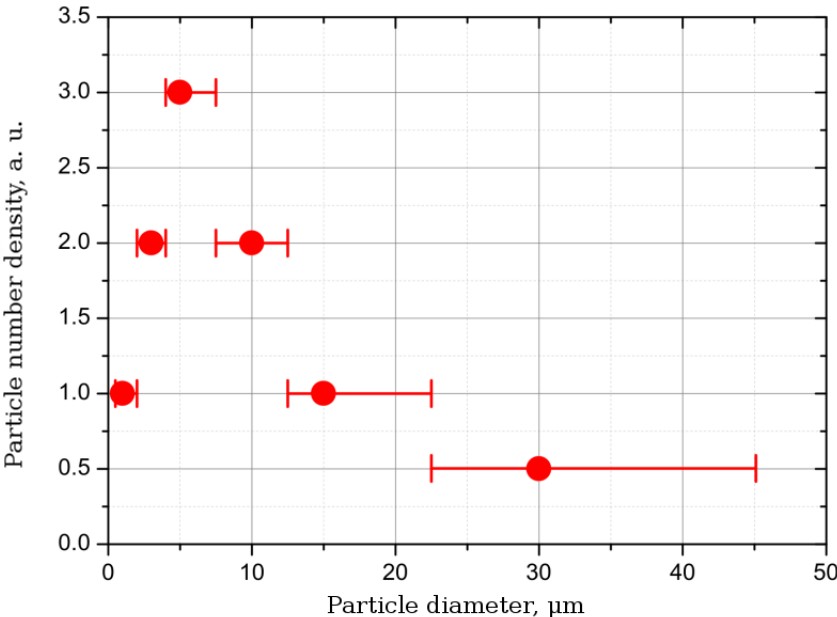

**Figure 4.** Sketch of the particle size distribution represented by COSMO-ART within the case study (red dots). The red lines with bars indicate the averaging margins we defined for the calculation of effective optical cross sections.

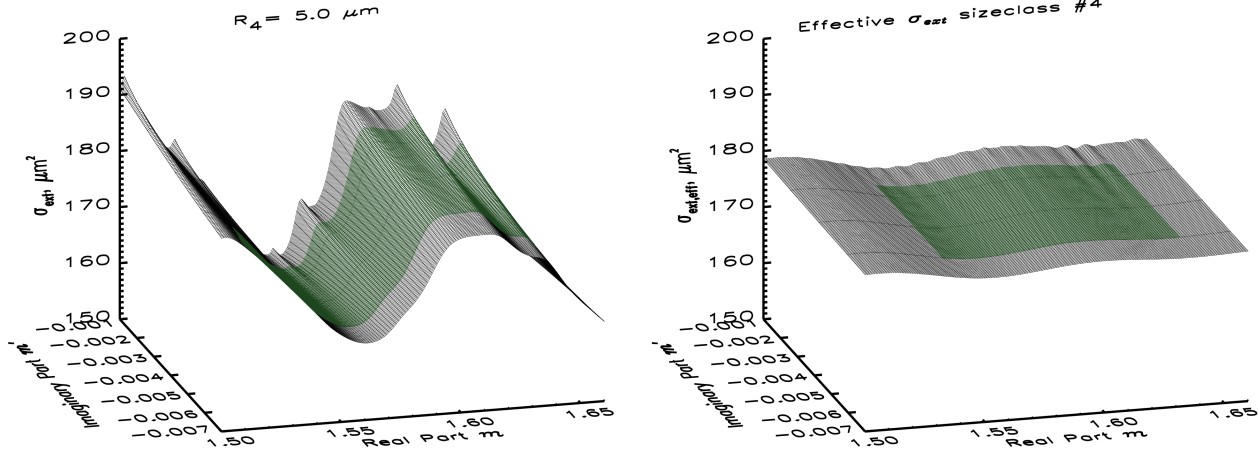

**Figure 5.** Sensitivity of $\sigma_{\mathrm{ext}}$ to the real and imaginary part of the refractive index for a single particle radius $R_p$ (left) and after calculating the effective extinction cross section $\overline{\sigma_{\mathrm{ext}}}$ (right). The green shaded area is the relevant range of real part $m$ and imaginary part $m'$ as explained in section 4.1.





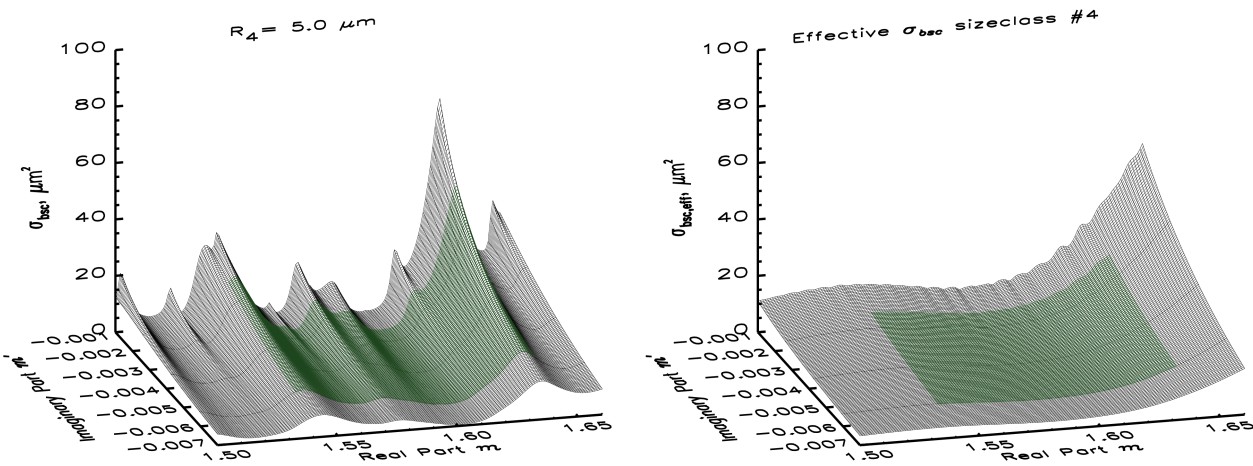

**Figure 6.** The same as Fig. 5 but for the backscatter cross section $\sigma_{\mathrm{bsc}}$ and the effective backscatter cross section $\overline{\sigma_{\mathrm{bsc}}}$.




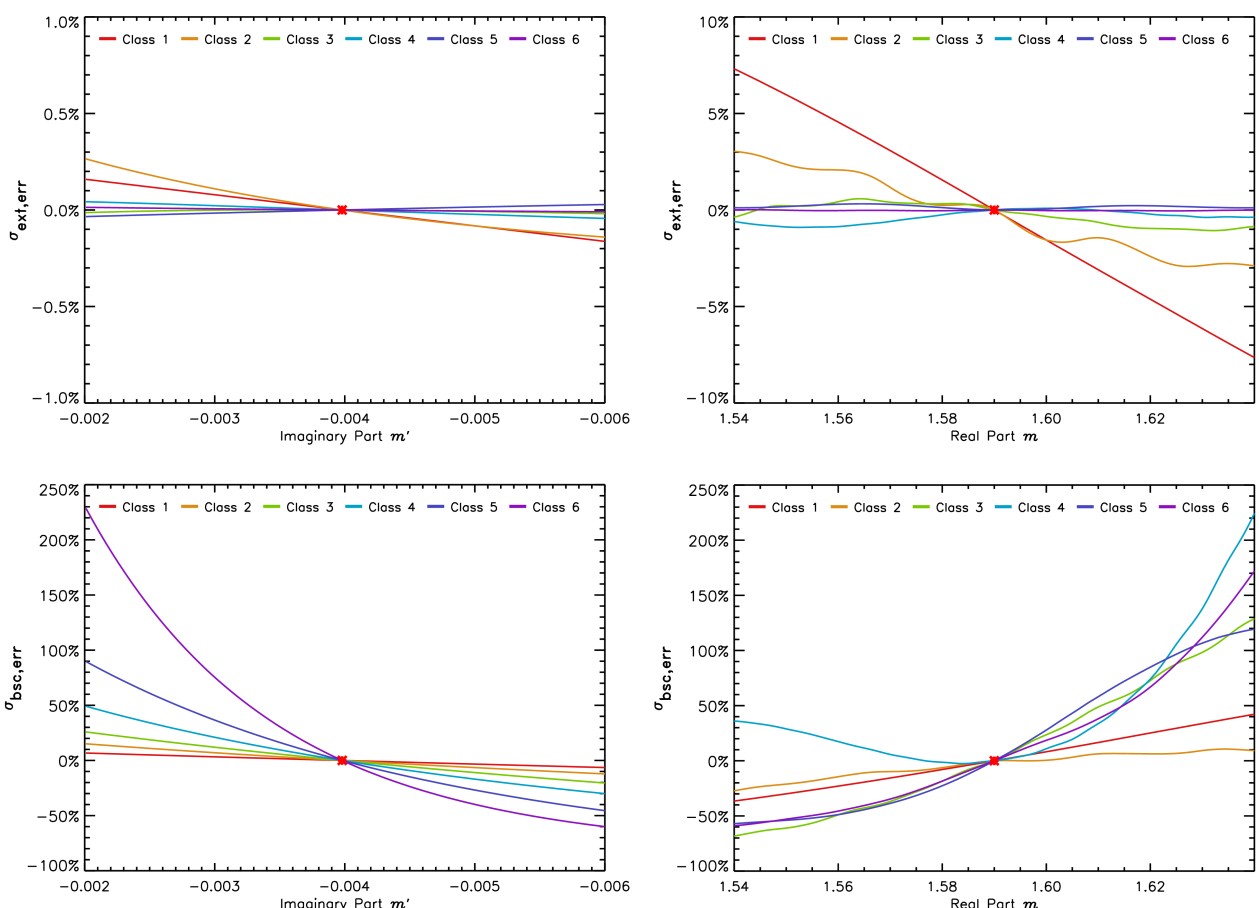

**Figure 7.** Relative errors of the effective extinction cross section (top row) and of the effective backscatter cross section (bottom row) if the assumed reference refractive index (red dot) is not equal to the true refractive index. Plots in the left row show the error for variable imaginary parts; plots in the right row for variable real parts of the refractive index. Uncertainties of the imaginary part may lead to a maximum error of 0.5% for the effective extinction cross section and of 230% for the backscatter cross section. Uncertain real parts also may lead to errors of 7% for the extinction cross section as well as of 225% for the backscatter cross section in the worst case.





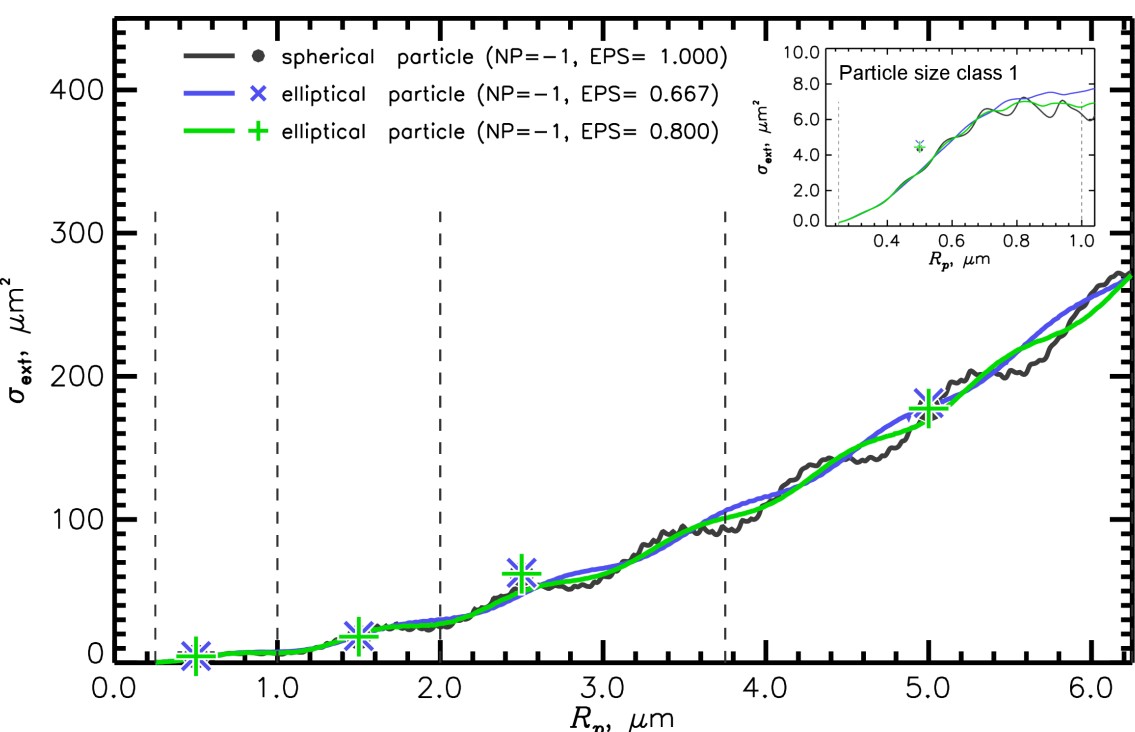

**Figure 8.** Extinction cross section spectrum for spheres (black line, EPS=1) and two prolate ellipsoids (blue and green line) given as equal-volume radius $R_p$. The dashed lines indicate the size-margins of each class and the symbols are the effective values of each size class.





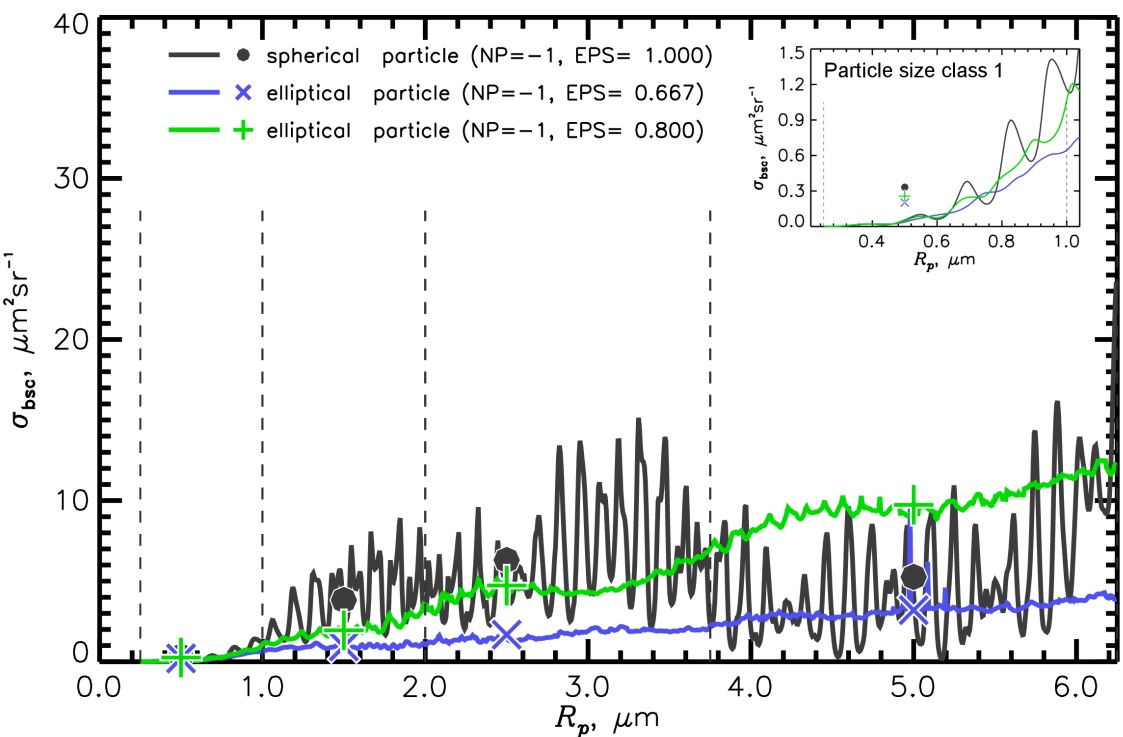

**Figure 9.** The same as Fig. 8 but for the backscatter cross section





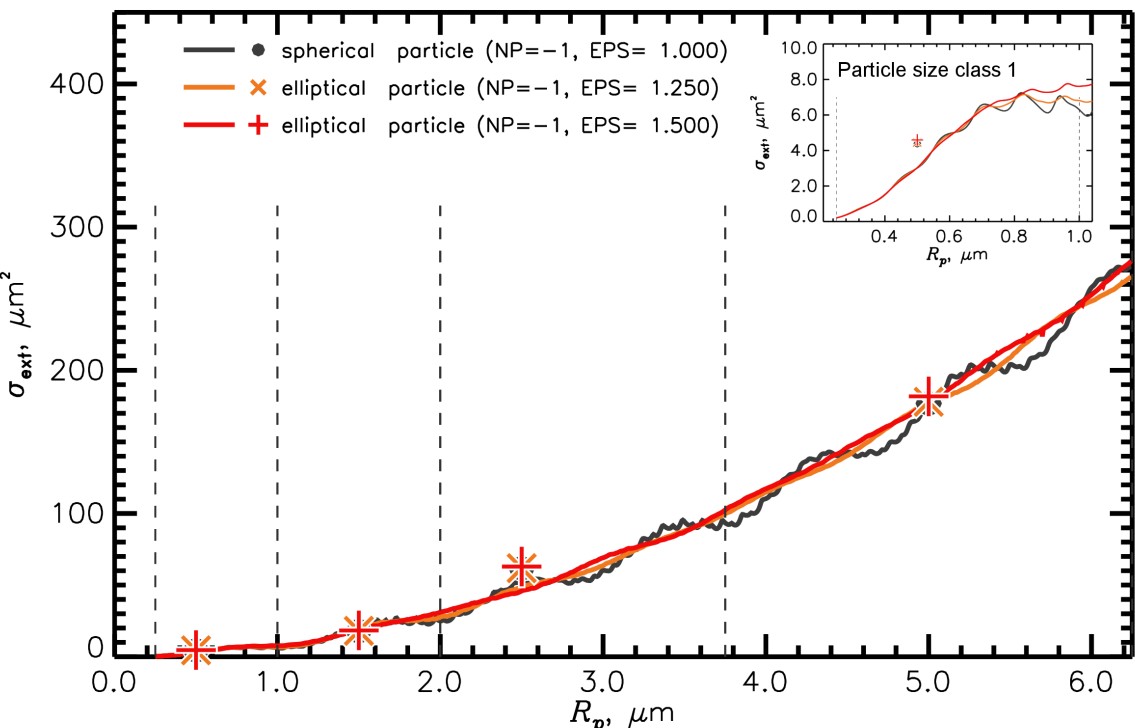

**Figure 10.** The same as Fig. 8 but for two oblate ellipsoids (orange and red lines).





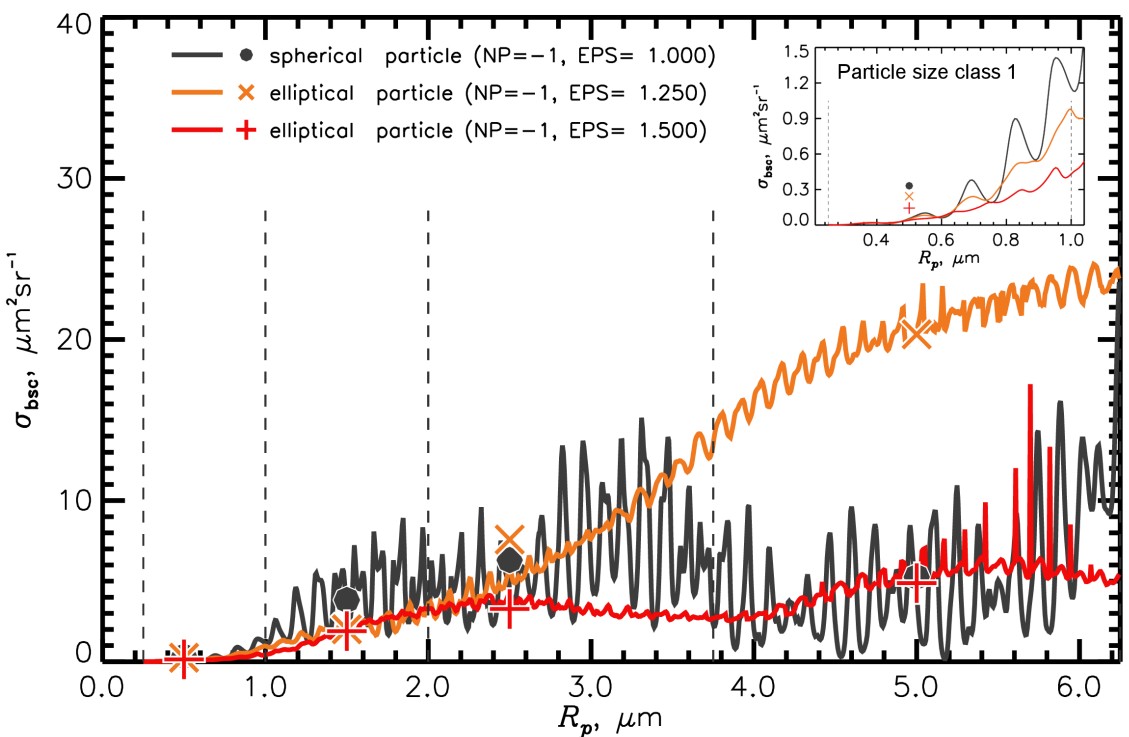

**Figure 11.** The same as Fig. 10 but for the backscatter cross sections.





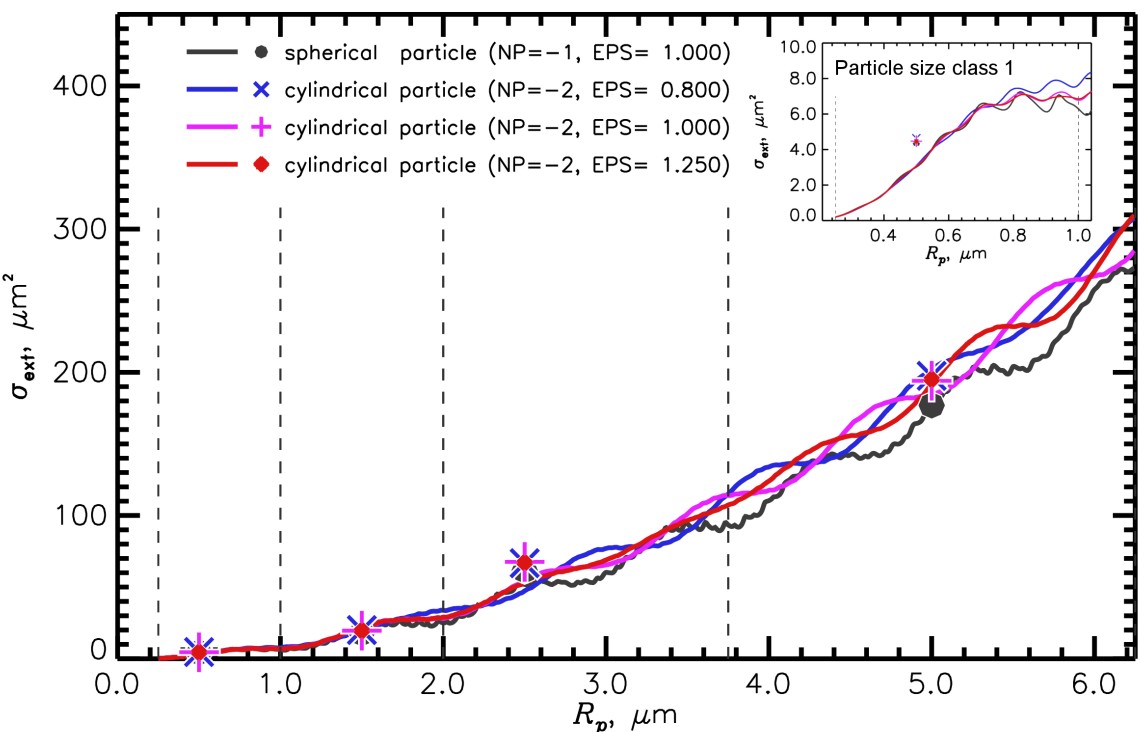

**Figure 12.** The same as Fig. 8 and Fig. 10 but for three types of cylindrical particles (blue, purple and red line).



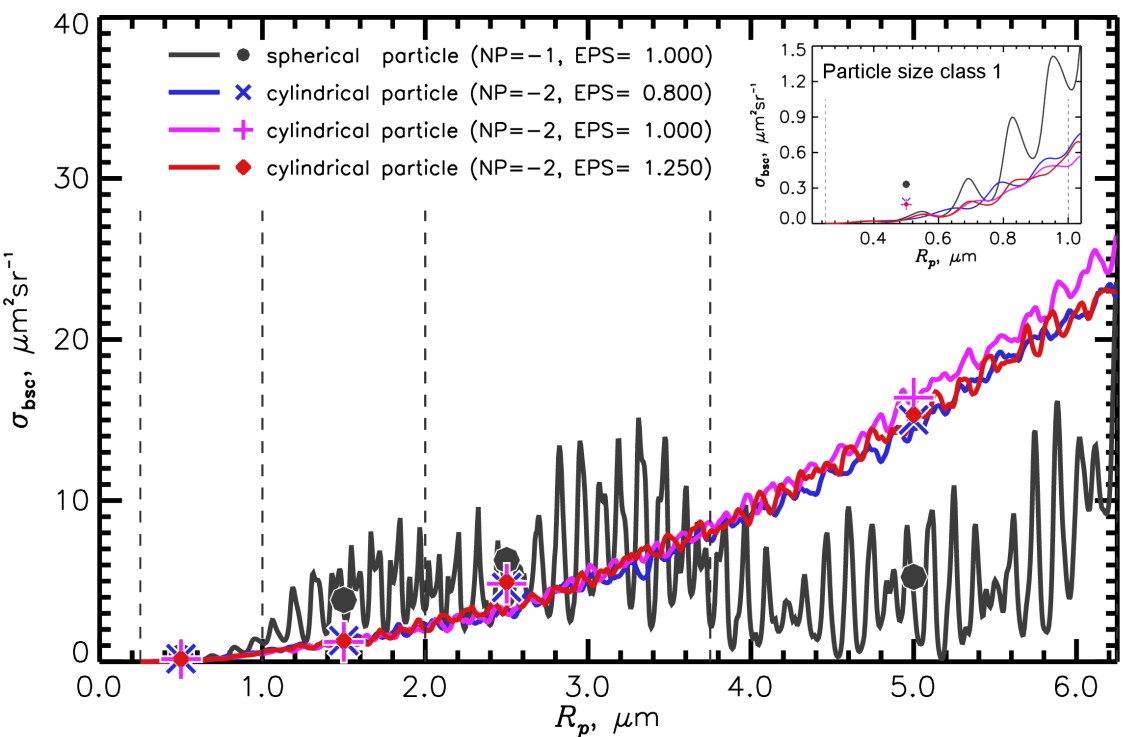

**Figure 13.** The same as Fig. 12 but for the backscatter cross sections.





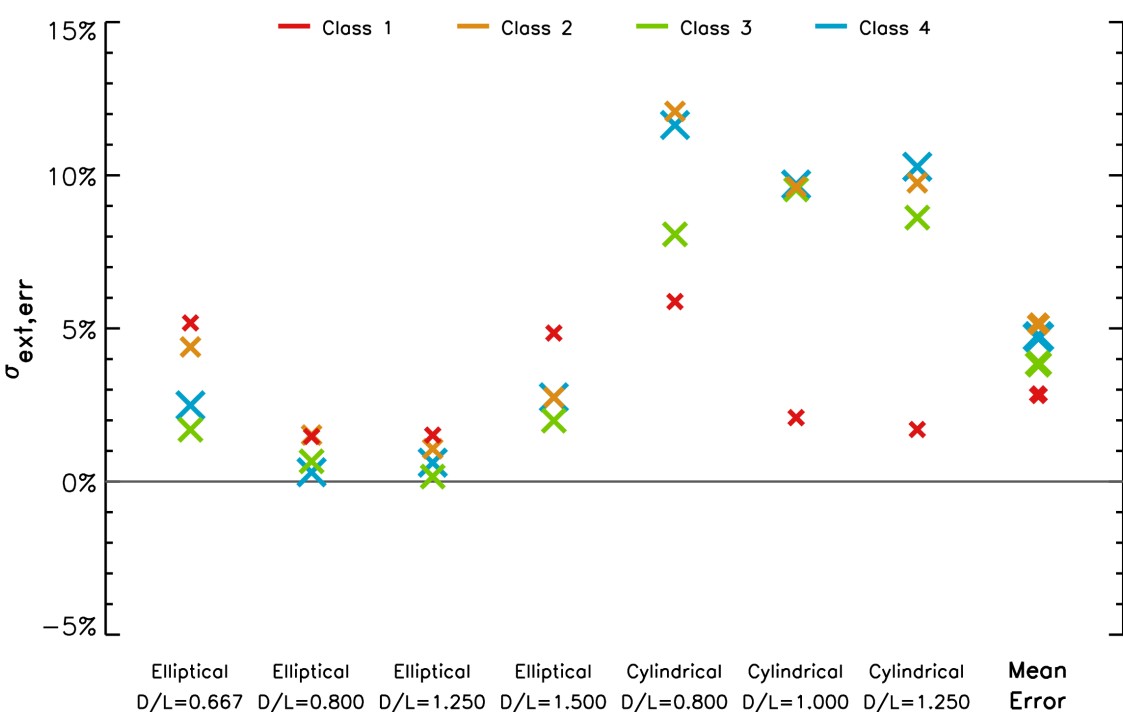

**Figure 14.** Relative errors of the effective extinction cross section if spherical particles are assumed but the real particles are of elliptical, cylindrical, or of mixed shape. The mean relative error of the analyzed particle shapes does not exceed 6% for the effective extinction cross section.





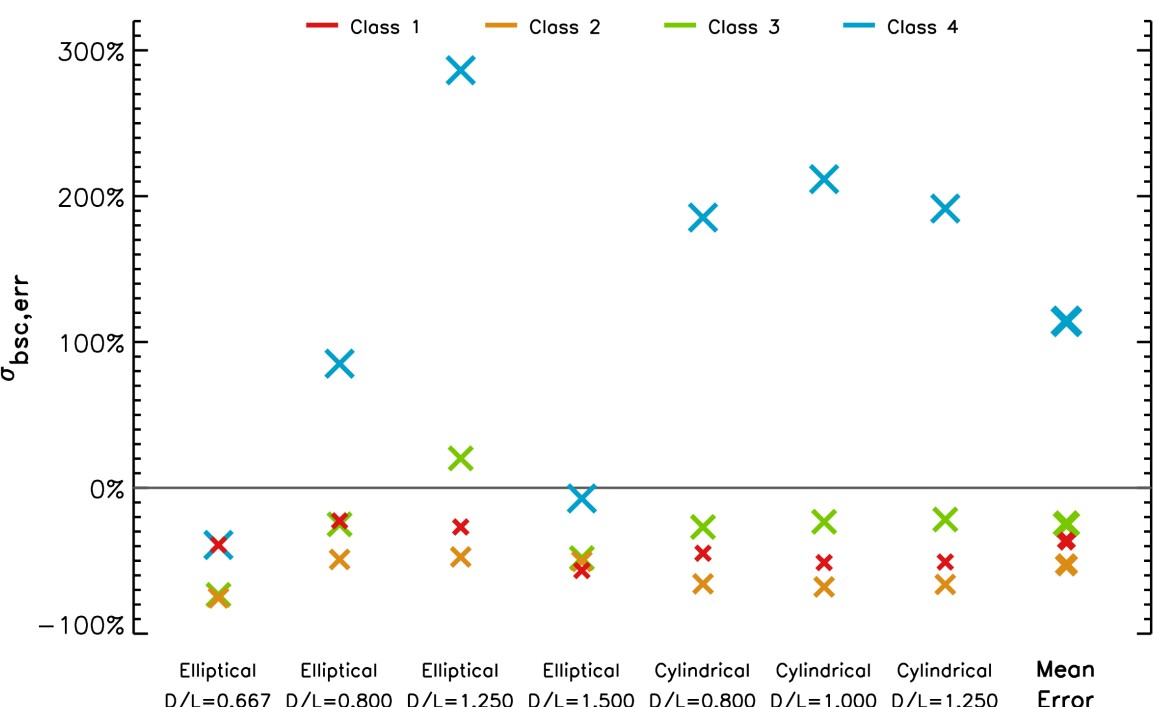

**Figure 15.** The same as Fig. 14 but for the backscatter cross sections. The mean error is the average relative error of all considered shapes on this plot.





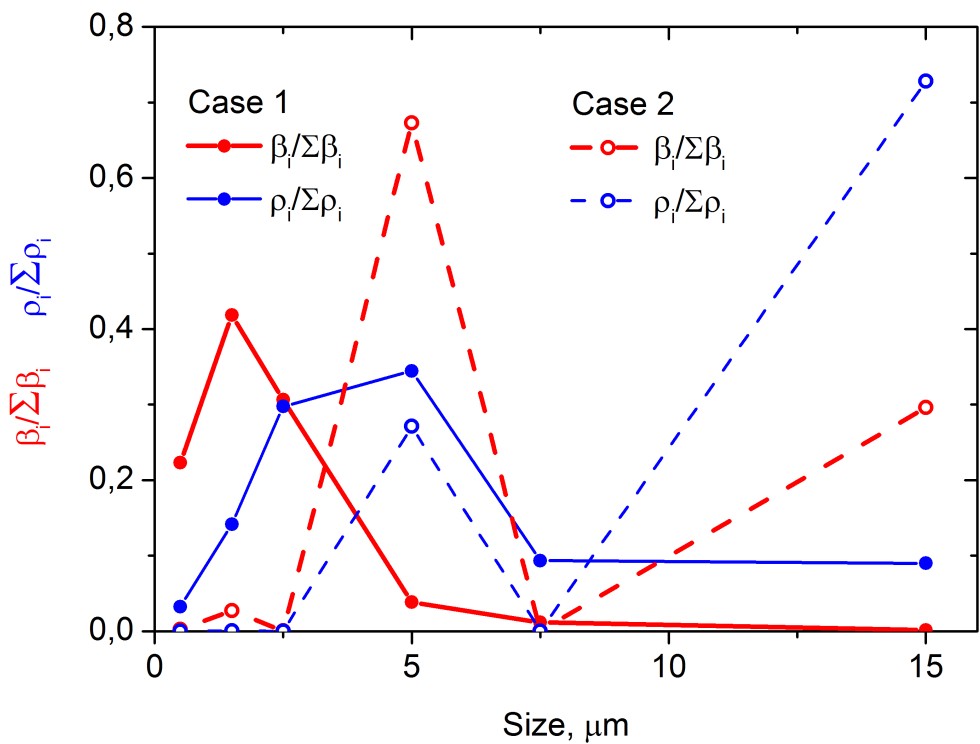

**Figure 16.** Relative contribution of the 6 volcanic ash size classes to the total backscatter coefficient (red) and to the total mass density (blue) for case 1 (solid) and case 2 (dashed lines). The data are based upon data in Table 4 and Table 5.





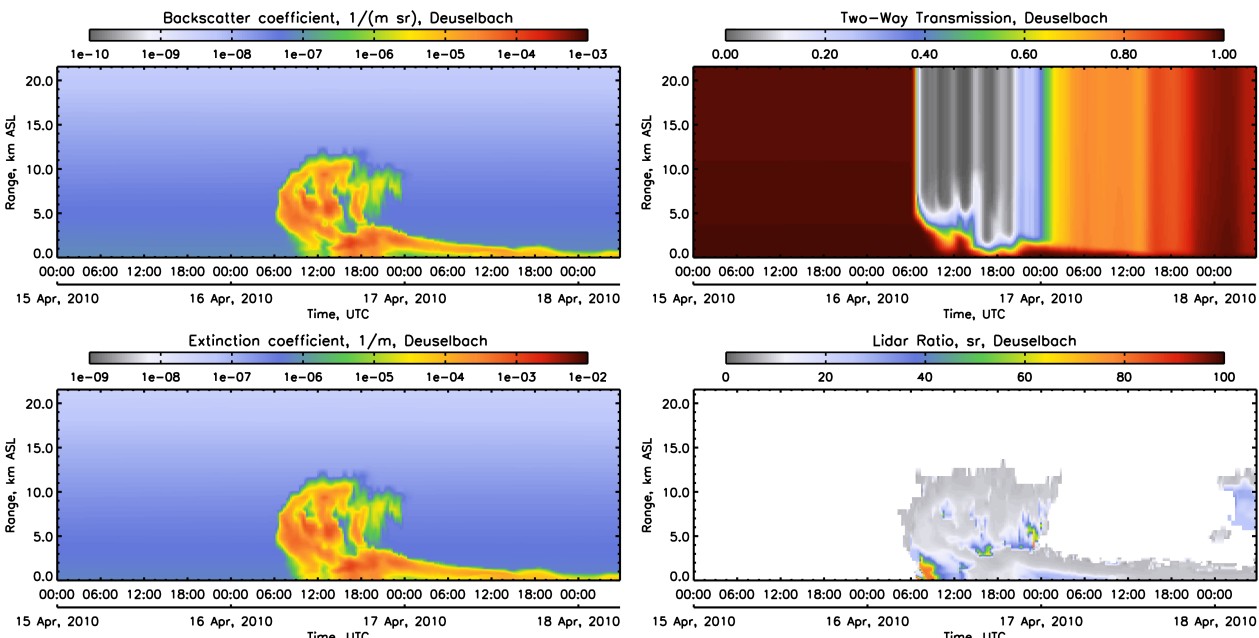

**Figure 17.** Time-height cross section of backscatter coefficient, two-way transmission, total extinction coefficient and lidar ratio, calculated by the forward model based on COSMO-ART output at the station Deuselbach (West Germany). The forward model represents clean air molecules and volcanic ash particles (no clouds, rain, fog, background aerosol or other scattering objects).



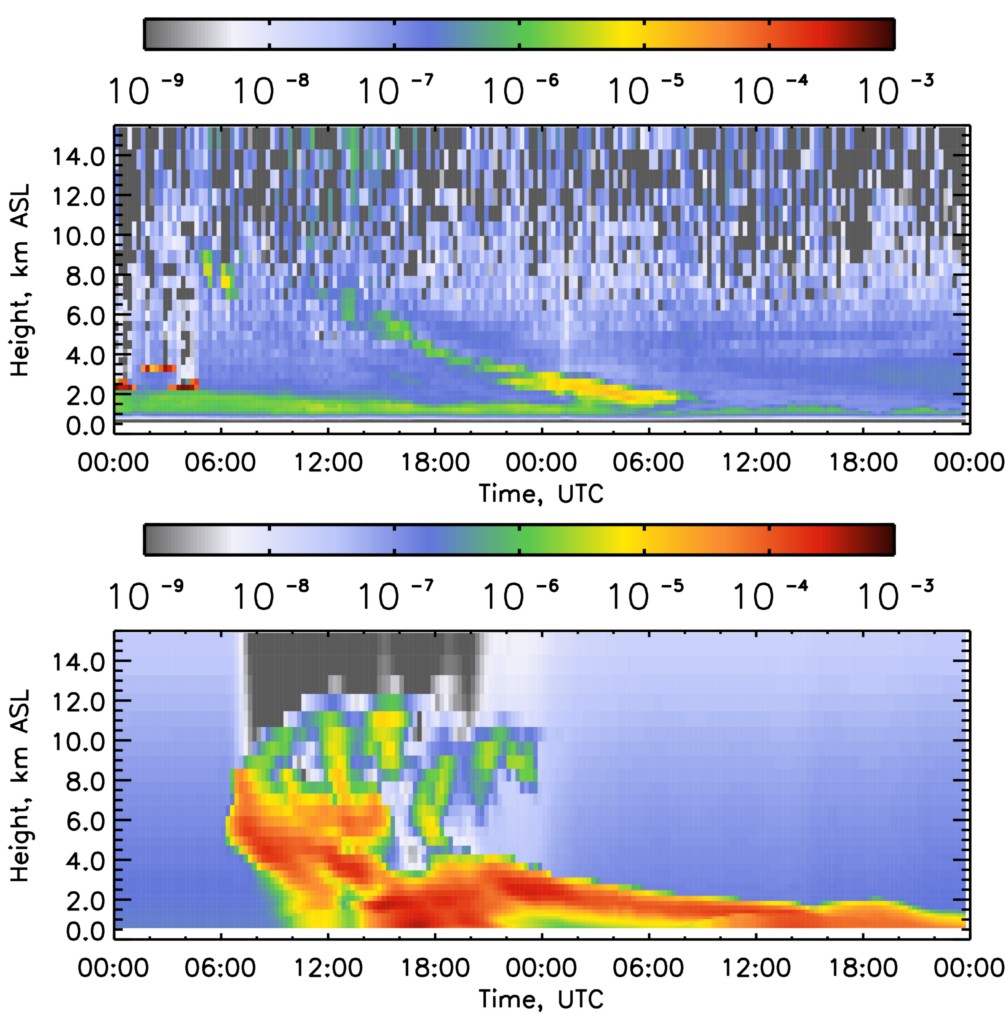

**Figure 18.** Attenuated backscatter coefficient of ceilometer (top) and forward model (bottom) at the station Deuselbach in Germany from the 16th of April 2010, 00:00 UTC to the 17th of April 2010, 24:00 UTC. The ceilometer data is calculated according to Eq. (27) and averaged according to the model resolution.





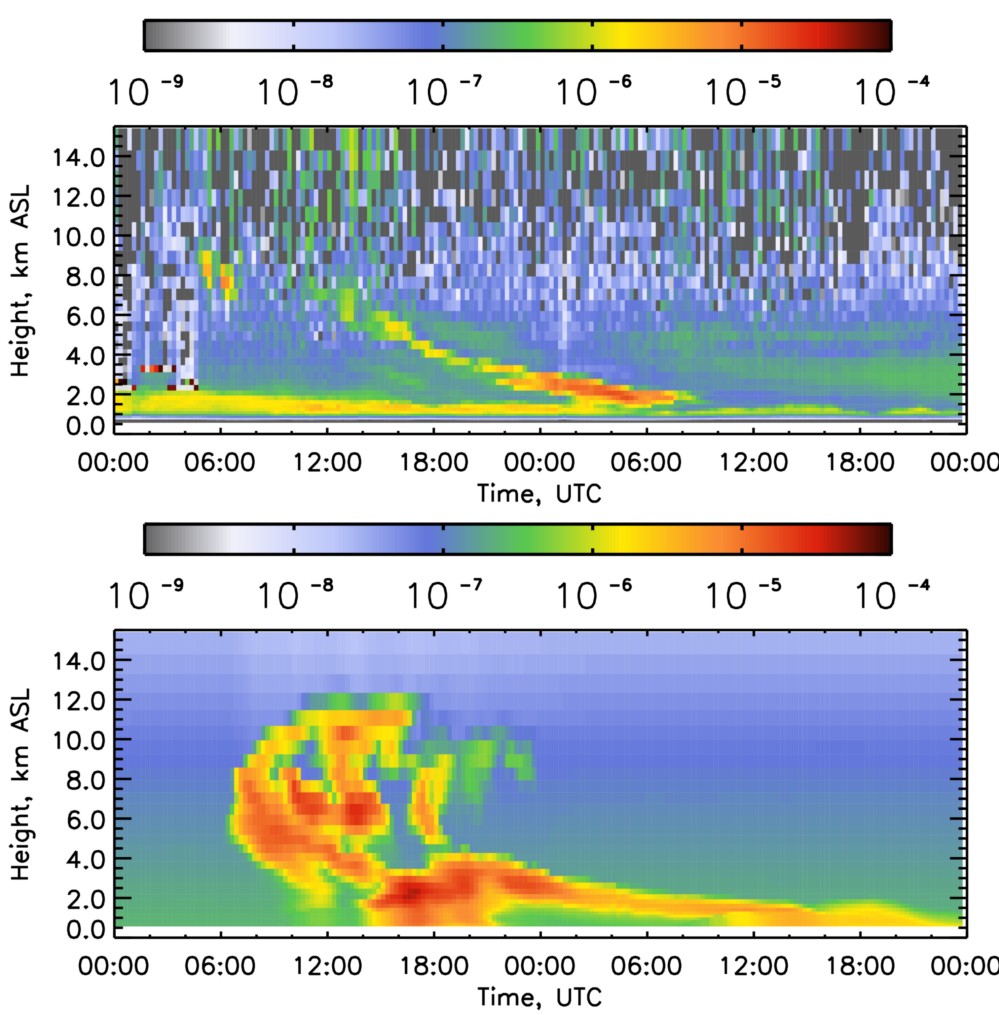

**Figure 19.** The same as Fig. 18 with a decreased volcanic ash number density. For this purpose, the ash number density predicted by COSMO-ART was reduced by factor 20 before applying the forward operator.