# Peer review of "A Backscatter Lidar Forward Operator for Particle-Representing Atmospheric Chemistry Models"

_Atmospheric Chemistry and Physics, 2016_

## Referee Comment (RC1) · Anonymous Referee #1 · 11 Aug 2016

Review of:

A Backscatter Lidar Forward Operator for Particle-Representing Atmospheric Chemistry Models by A. Geisinger et al.

**1  General comment:**

The paper describes a lidar forward operator based on basic lidar equations for calculating the backscatter and extinction coefficient profiles from a atmospheric chemistry model (ACM). This forward operator can than be used to compare aerosol concentration in the model and at a later stage assimilate lidar backscatter profiles into ACMs. Beside the method, one special case study namely the ash plume of the Eyjafjalla-jökull volcanic eruption is used in the manuscript to demonstrate the applicability of the forward operator.

I appreciate the approach of calculating backscatter and extinction coefficients directly from the particle size distributions. Thus this is an appropriate and scientifically relevant contribution to ACP in general. The manuscript is also well organized and the analysis and results are clearly communicated and contextualized. However, I recommend major revisions due my major concerns listed below, i.e. the issues should be clarified and partly revised before the manuscript can be accepted for publication.

**2  Major concerns:**

1. In the manuscript, a very prominent but also very special case is selected for the case study. Earlier studies already showed that these volcanic ash particles were very aspheric and are therefore not the optimum case to test the forward operator under "normal" atmospheric aerosol loadings. Especially with the assumption of spherical particles. Would it be better to have chosen a more simpler case with more spherical aerosol particles available ? Can you please comment on the motivation for using this case study.

2. As you mentioned on page 12 l. 27. Schumann et al. 2011 found a mean aspect ratio of about 1.8 for particles smaller 500 nm and about 2 for larger particles. This particles where sampled in-situ with the Falcon aircraft and should provide the most realistic parameters for volcanic ash particles in the atmosphere. That means that the ash particles observed where more aspheric with higher aspect ratio than it was chosen in this study. I'm sure, if you would enhance the aspect ratio up to 2 in the T-matrix calculations which is actually the mean for larger

particles than 500 nm, it produces even larger deviation in comparison to the Mie calculations. Maybe the impact is smaller than I think, but this has to be shown in a more extended sensitivity study covering the whole range of aspect ratios. The seconds questions concerning the T-matrix calculation is, why do you have chosen a smaller aspect ratio of only 1.25/0.8 for the cylindrical shape? This should also be extended to higher aspect rations of about 2 to cover the range of the observations. Maybe it would also be good to add one comparable DDA simulations to the sensitivity study to see the full range of uncertainties even though it is only possible for sizes smaller 6.2 $\mu$m. However, this particles constitute the largest contribution to the total number concentration. Maybe it is also not worth to generate the same Figure 17/18 for one type of aspherical scatterers with the T-matrix method.

3. To make a fair comparison of ACL data with the model/forward operator all major uncertainties, i.e shape, refractive index, calibration constant, have to be estimated and combined to show if the statement concerning overestimation of particle number in the model is justified. As the authors mentioned in the text, the backscatter coefficient is much more affected by the particle properties than the extinction coefficient. Thus, the low lidar ratio with most values around 10-15 in comparison to the other studies by Kokkalis et al, 2013; Ansmann et al. 2010 and Mortier et al., 2013. is a strong indication for an incorrect/overestimated backscatter coefficient and could also explain the large discrepancy between modeled and measured backscatter coefficient profile. Only if both backscatter and extinction coefficient are higher in the forward operator simulated profiles in comparison to the ACL data, than this is an indication for a too high particle number concentration in the model. But this cannot be checked with the ACL data, because of missing independent extinction data. In summary, the conclusion of a wrong number concentration cannot be drawn from the case study with so many simplifications and without considering the errors/uncertainties of the forward operator.

I recommend to provide a comprehensive estimation of the combined uncertainties with consideration of point 2 above.

4. Some statements are to strong and cannot be drawn from this case study and from the implementation of the forward operator with it's huge uncertainties regarding refractive index and particle shape. For example in the conclusion section (p. 19, l. 13-14): How do you want to constrain or correct the model regarding shape and refractive indices ? Your forward operator use the refractive index as input and larger differences can also be expected from the simplicity of assuming spheres. I don't see a chance to retrieve refractive indices or shapes with the forward operator in combination with the ACL measurements, and thus this statement is not reasonable. In addition, the forward operator needs always additional information like the refractive index. So it is hard to apply the forward operator as a general/operational tool or even for assimilation without having these information available. So that the statement of a "crucial step" written in the abstract or similar in the conclusion is little over emphasized.

**3   Specific comments/questions:**

- Why are you using the Mie code with the assumption of spherical particles? Instead you could create a similar lookup table for aspherical particles with different aspect ratios and refractive indices.

- p. 11, l. 21-25: Calipso is used for estimating the calibration constant of the ACL lidar systems. How is this estimation performed? Usually, there are larger differences in space and time between local and satellite observations beside the viewing geometry. The brings me to the question how large is the uncertainty of this method also regarding to point 3 and also point 2?

- p. 13, l. 13-15: Why should an ellipsoidal distribution lead to less realistic scattering calculation than spherical scatterers? This statement is a little bit strange, as you showed in the sensitivity study that more aspherical scatterers can better represent the observations. This statement should be removed or at least better explained.

- p. 16-17, l 34-2: The modeled structure can also be strongly overestimated by the forward operator and thus leads to visible structures in the simulated profiles. These structure could be real but just below the detection limit of the ACL measurements.

- p. 18., l. 8: It is written that both backscattering and extinction are higher compared to ACL observations. But the ACL cannot measure extinction profiles directly. So how did you compare the extinction profiles to the ACL observations ?

**4 Technical comments:**

- In the text and the especially in the figures a mixture of diameter and radius is used for the particle size. Particle size is sometimes written without mentioning either radius or diameter. I suggest to stay with one notation to avoid any confusion.

- Figures 8-11: I suggest to combine Figure 8 with 10 and Figure 9 with 11. There are not large differences in the outcome of the each figure pair and therefore the benefit of splitting the figures is low. This would reduce the number of figures which makes the manuscript more clearer. In addition, I recommend to scale the extinction cross section from 0-300 and the backscatter cross section to 0-25. The small zoom plots can be moved to the white spaces in the lower left corner

or below the legend. This would enhance the visibility of the differences between the different types.

• p.11 l 18.: How large is the measurement error of the diameter to state a number with such an accuracy ? It seems the number of the aperture of the receiving telescope has to much decimal places. I suggest to reduced the value to only two decimal places. This would account for a measurement error for less than 1mm in diameter.

• p. 8: There are a lot of basic formulas in the manuscript which makes it hard to read. For example the formulas regarding molecular scattering can be referenced and thus removed from the text. The formulas 14-17 can be combined etc. This would make the text more clearer.

• p. 14 l 30: spelling error: with a diameter

---

## Referee Comment (RC2) · Anonymous Referee #2 · 7 Sep 2016

General remarks:

The paper deals with an attractive and important research topic. The development of a backscatter lidar forward operator is very useful to support aerosol transport modeling and efforts to forecast aerosol conditions.

However, the paper is a pure (100%) technical, methodological paper and thus appropriate for AMT. I therefore recommend to move it to AMT in case that it gets finally accepted.

Unfortunately (I must say), the paper is not acceptable in its present form. But I have hope that it can survive. I was expecting a straight forward paper, a clear, compact

description of the nice method and then also a nice and easy to follow (and convincing) case study (demonstration case) which clearly corroborates the usefulness of the methodologic concept and developments. But all this is not presented! The method is introduced in a rather lengthy way, text-book knowledge is outlined in extended detail. This can easily be avoided. But! .... The demonstration case (Eyjafjalla volcanic aerosol scenario) is simply the worst case they could select! This case must be substituted by a case for which Mie scattering can be applied.

The methodology is based on the fundamental assumption that the aerosol particles can be handled as spheres (as so called Mie particles such as marine aerosol particles or urban haze or biomass burning smoke particles). This is well justified in the case of anthropogenic pollution including biomass burning smoke. However, the demonstration case, the authors selected, deals with volcanic dust particles crossing Germany quite shortly after emission so that nothing is well known, size distribution, chemical composition, shape properties. Such a situation occuring on 16-17 April 2010 has never been observed before with advanced atmospheric technology. Fortunately, several EARLINET lidars were able to pick up even the first traces of the heavy dust front which crossed Germany on 16 April 2010. The measured lidar ratios indicated irregularly shaped volcanic dust particles. The AERONET observations revealed that these fresh plumes were coarse mode dominated.... So it was by far not a normal aerosol situation. It was an extreme situation! As mentioned, nothing is well known, the size distribution of the volcanic dust particles is not known, the shape properties are not well known. Even the chemical composition could not be well characterized. Aircraft observations were started several days later, when the aerosol mixture and physico-chemical properties already had changed significantly, and spherical sulfate particles had formed from the emitted SO2, and were mixed with the voclanic ash and dust. On 16 Apri 2010, it is most likely that only dry ash and dust particles were present. And for such a case, Mie-scattering-based methods (including this backscatter lidar forward operator method) can definitely be not applied, as it is shown in this paper!

The authors are forced to discuss this case in a very speculative way so that the main question at the end was not: What do we learn from this paper? No, the main question was: Why did they choose this incredibly complicated case so that it was, at the end, impossible to demonstrate the usefulness of the new technique?

So, my conclusion is not only,.... the paper needs major revision. The conclusion is, we need an easy case that shows the applicability of the forward operator model! This paper definitely fails to show that. Without such a simple case, the paper must be rejected! To be fair, many papers of this quality get rejected at this stage, already. However, I believe that there is enough and substantial material to save it! But only if we have another, a much better, much easier demonstration case.

The conclusions based on this volcanic case the authors selected will always remain rather uncertain and thus speculative because of lack of sufficient knowledge of the relationships between the optical properties and the size/shape/composition properties of Islandic volcanic ash 2 two days after emission. There is no hope!

The paper has many other weak points. It is not well written. The reference list is far away from reflecting the aerosol lidar science field in a proper way. Even in the case of the volcanic aerosol plumes the authors leave out to mention essential papers. All this must be improved.

Point by point....:

Abstract:

The abstract must be rewritten: please provide a compact text with the goal of the article, the methods used, and the essential finds. That's it! All the motivating and explaining statements should not be given in an abstract. The right place for such information is the Introduction (section 1).

Section 1 Introduction

P 2-4: The introduction can be kept much shorter and more compact. Many details are

given that have nothing to do with the goal of the paper.

P2, l30: Please check Cuevas, MPL observations and lidar data assimilations into models (see my reference list below).

P3, l1-15: please remove the first paragraph, please focus on aerosol lidar only. The work of Japanese groups to provide lidar data for the modeling community should be mentioned in this context. They were pioneering in the 1990s and are even active presently to combine lidar and atmospheric aerosol modeling. There are also papers in which lidar data are used to validate models and verify results from the EARLINET community (check the speciall issue of EARLINET in AMT, and the papers including there reference lists). There were also efforts to combine lidar and model results in the Tellus SAMUM special issues of SAMUM (Tellus 2009, 2011). In the case of multiwavelength lidar (here the authors probably mean inversion techniques), I was surprised that the authors left out to provide the references to the fundamental Mueller, Veselovsky and Boeckmann papers! These authors are active in this field since more than 15 years. Mueller (JGR 2010, 2012) published several papers on the effects in optical closure studies when the scatterers are non spherical...and , if remember right, even not of spheroidal nature.

P3: Please add somewhere the basic work of Heese et al and Wiegner and Geiss in AMT on the characterization of ACLs (see my list below). All in all I was not very happy with the introduction section. It gave me the impression that the authors were not willing or not able to provide a well-balanced literature overview on aerosol lidar. This needs to be significantly improved and will certainly strengthen the quality of the paper as a whole. A good literature review always provides the impression, the work is done in a professional way. The other way around, ..... provides the opposite feeling.

Here are some papers, that should be cited...

Heese, B., Flentje, H., Althausen, D., Ansmann, A., and Frey, S.: Ceilometer lidar comparison: backscatter coefficient retrieval and signal-to-noise ratio determination,

Atmos. Meas. Tech., 3, 1763-1770, doi:10.5194/amt-3-1763-2010, 2010.

Wiegner, M. and Geiß, A.: Aerosol profiling with the Jenoptik ceilometer CHM15kx, Atmos. Meas. Tech., 5, 1953-1964, doi:10.5194/amt-5-1953-2012, 2012.

An introduction of a paper dealing with a ceilometer forward operator should include such 'fundamental' papers to my opinion, especially devoted to Jenoptics ceilometers.

Here, some further papers ( and one PhD work) on lidar data assimilation, and the references in these articles may help. . ... Please have a look and check!

http://www.mri-jma.go.jp/Dep/ap/ap1lab/member/tsekiyam/files/sekiyama_thesis.pdf

Wang, Y., Sartelet, K. N., Bocquet, M., Chazette, P., Sicard, M., D'Amico, G., Léon, J. F., Alados-Arboledas, L., Amodeo, A., Augustin, P., Bach, J., Belegante, L., Binietoglou, I., Bush, X., Comerón, A., Delbarre, H., García-Vízcaino, D., Guerrero-Rascado, J. L., Hervo, M., Iarlori, M., Kokkalis, P., Lange, D., Molero, F., Montoux, N., Muñoz, A., Muñoz, C., Nicolae, D., Papayannis, A., Pappalardo, G., Preissler, J., Rizi, V., Rocadenbosch, F., Sellegri, K., Wagner, F., and Dulac, F.: Assimilation of lidar signals: application to aerosol forecasting in the western Mediterranean basin, Atmos. Chem. Phys., 14, 12031-12053, doi:10.5194/acp-14-12031-2014, 2014.

Janiskova et al., Assimilation of cloud information from space-borne radar and lidar: experimental study using a 1D+4D-Var technique, QUARTERLY JOURNAL OF THE ROYAL METEOROLOGICAL SOCIETY Volume: 141 Issue: 692 Pages: 2708-2725 Part: A Published: OCT 2015

Campbell et al., CALIOP Aerosol Subset Processing for Global Aerosol Transport Model Data Assimilation, IEEE JOURNAL OF SELECTED TOPICS IN APPLIED EARTH OBSERVATIONS AND REMOTE SENSING Volume:3 Issue: 2 Pages: 203-214 Published: JUN 2010

Cuevas, E., Camino, C., Benedetti, A., Basart, S., Terradellas, E., Baldasano, J. M., Morcrette, J. J., Marticorena, B., Goloub, P., Mortier, A., Berjón, A., Hernández, Y.,

Gil-Ojeda, M., and Schulz, M.: The MACC-II 2007–2008 reanalysis: atmospheric dust evaluation and characterization over northern Africa and the Middle East, Atmos. Chem. Phys., 15, 3991-4024, doi:10.5194/acp-15-3991-2015, 2015.

I also found recently somewhere (unfortunately I do not remember the web page) a workshop report on MPL lidar data assimilation into the NMMB model. The first author was again: Cuevas. So there are presently data assimilation efforts in the dust forecast scene.

P4, l8: I sounds attractive and convincing when essential input parameters have not to be assumed. But in the case of the lidar ratio, it is frequently even better to calculate the robust extinction properties and then take a lidar ratio of 50 sr to obtain the backscatter coefficient. I would like to see if this option is mentioned too. For example, for volcanic dust (your case study) it is shown that the lidar ratio is around 50sr (Gross et al. 2012), and it would be simply much more straight forward to use such a number of 50sr than your rather unrealistic values close to 'modeled' 5-15 sr. I will come to this point later again.

To my opinion, one of the best Eyjafjalla volcanic lidar ratio papers is (please add it to the references and include the findings in the discussion) :

Groß, Silke und Freudenthaler, Volker und Wiegner, Matthias und Gasteiger, Josef und Geiß, Alexander und Schnell, Franziska (2012) Dual-wavelength linear de-polarization ratio of volcanic aerosols: Lidar measurements of the Eyjafjallajökull plume over Maisach, Germany. Atmospheric Environment, 48, Seiten 85-96. DOI: 10.1016/j.atmosenv.2011.06.017.

To my understanding, the Ansmann 2010 lidar ratios of 50-60sr (similar to the ones in Gross et al, for the lofted isolated volcanic ash layer. . .) probably correctly describe the dry volcanic aerosol lidar ratios in the beginning of the volcanic episode. Later (18-19 April and afterwards. . .) the pure volcanic dust and ash conditions were gone. Volcanic sulfate aerosols formed from the emitted volcanic SO2 so that a mixture of spherical

and irregular shaped particles were always present. I mention it only to avoid that the authors misinterpret the findings in the Gross paper. The lidar ratios were definitely in the range of 50-60sr for dry volcanic dust (in the absence of volcanic sulfate). All this is very similar to desert dust, and the desert dust lidar ratio at 1064nm seems to be equla to the lidar ratios for 532 nm or even slightly higher than the ones for 532nm, gained from 1064nm lidar/ 1020nm photometer observations.

The authors mention the Greek paper on volcanic dust of Kokkalis et al., but if they would read Papayannis et al., (Atmos Env. or Atmos Res., 2012?) they would see that it was rather complicated to identify the volcanic aerosols in the mixtures of urban haze, marine particles, Saharan dust etc over the Eastern Mediterranean„ far away from Iceland, so that information of these articles (even Mona et al., 2012) should be handled with caution.

Section 2 Methods

Section 2.1 can be shortened drastically, the lidar equation is given in so many text books, so many details are not needed. Focus on the parameters needed for Eq.2.

P4, l30: Lidars are never elastic (hopefully. . .), so please write: elastic backscatter lidar Also sections 2.2.1 and 2.2.2 contain just text book knowledge. Please keep the discussion as short as possible, use references.

P7,l7: Buchholtz's formula holds for 200 to 1000nm, but you use 1064nm (please mention the used wavelength much earlier in the paper, for example already in this section!)? Did you extrapolate? Or ignore it? Please state the wavelength immediately, and give all the Rayleigh parameters for this wavelength, including the lidar ratio. . .

Section 2.2.3, even here, please provide the wavelength of the ceilometer, and then discuss all the different size scenarios for this wavelength, give examples for which the Mie theory may holds, for what size range (Aitken mode, accumulation mode, etc. . .), the same for DDA, T-matrix etc. Which of the natural particles have almost spheroidal

shape, which look like cyclinders? Make the full discussion more realistic, more lively.

P7,l20: So you definitely use Mie scattering codes for your backscatter lidar forward operator! This is ok, and the best first step you can choose!

BUT!!! … as a consequence, please provide us with an application (demonstration case) for which Mie computation really holds (fine-mode aerosol, urban haze and smoke) . This is a mandatory condition for the final acceptance of this paper from my side!!!

Again, in conclsuion: Sections 2.2.3 and 2.2.4 are rather lengthy, not necessary, text book knowledge, sections should be as short as possible.

Section 3 Case Study

I could simply leave out the rest because I am not willing to accept a revised version with such a complicated case. But again, many figures could be left out because they are already presented in the literature, e.g., that the extinction coefficient is robust with respect to shape effects (see for example Dubovik JGR, 2006, but this is shown in many other papers too). So, one could easily skip Figs.8, 10, 12, 14.

Here is the Dubovik reference:

Dubovik, O., Sinyuk, A., Lapyonok, T., Holben, B., Mishchenko, M., Yang, P., Eck, T., Volten, H., Munoz, O., Veihelmann, B., van der Zande, W. J., Leon, J. F., Sorokin, M., and Slutsker, I.: Application of spheroid models to account for aerosol particle non-sphericity in remote sensing of desert dust, J. Geophys. Res., 111, D11208, doi:10.1029/2005JD006619, 2006.

The paper has 19 Figures, and 50% are just required to explain the complexity of this crucial volcanic ash case and the impact of size, shape and composition of volcanic particles on the related optical properties. Alone this fact corroborates that we need a better demonstration case. The main goal of the paper is: presentation of the backscatter lidar forward application plus demonstration case. And without presentation of a convincing demonstration case, the paper must be rejected because the main message and conclusion then becomes completely unclear.

P10, l23: One of the best lidar-ratio papers in this field is not mentioned in the article: Groß, Silke und Freudenthaler, Volker und Wiegner, Matthias und Gasteiger, Josef und Geiß, Alexander und Schnell, Franziska (2012) Dual-wavelength linear depolarization ratio of volcanic aerosols: Lidar measurements of the Eyjafjallajökull plume over Maisach, Germany. Atmospheric Environment, 48, Seiten 85-96. DOI: 10.1016/j.atmosenv.2011.06.017.

One needs the depolarization ratio to see in which of the layers pure or almost pure volcanic dust was observed. In the lowest layers a mixture of local pollution, sulfate particle from the volcanic emission of SO2 and volcanic ash prevailed. Nevertheless, the lidar ratio for the volcanic ash was always close to 50 sr, and in the mixtures even larger, probably because of the freshly formed quite small sulfate particles. None of the publications on the Eyjafjalla eruptions, showed lidar ratios below 40sr in the volcanic layers. And this must be the guide for all your modeling studies which to my opinion are at all rather speculative. Fact is: The reality did not provide volcanic lidar ratios of 10-20sr.

Add to the references. Pappalardo, G., Mona, L., D'Amico, G., Wandinger, U., Adam, M., Amodeo, A., Ansmann, A., Apituley, A., Alados Arboledas, L., Balis, D., Boselli, A., Bravo-Aranda, J. A., Chaikovsky, A., Comeron, A., Cuesta, J., De Tomasi, F., Freudenthaler, V., Gausa, M., Giannakaki, E., Giehl, H., Giunta, A., Grigorov, I., Groß, S., Haeffelin, M., Hiebsch, A., Iarlori, M., Lange, D., Linné, H., Madonna, F., Mattis, I., Mamouri, R.-E., McAuliffe, M. A. P., Mitev, V., Molero, F., Navas-Guzman, F., Nicolae, D., Papayannis, A., Perrone, M. R., Pietras, C., Pietruczuk, A., Pisani, G., Preißler, J., Pujadas, M., Rizi, V., Ruth, A. A., Schmidt, J., Schnell, F., Seifert, P., Serikov, I., Sicard, M., Simeonov, V., Spinelli, N., Stebel, K., Tesche, M., Trickl, T., Wang, X., Wagner, F., Wiegner, M., and Wilson, K. M.: Four-dimensional distribution of the 2010 Eyjafjalla-jökull volcanic cloud over Europe observed by EARLINET, Atmos. Chem. Phys., 13,
4429-4450, doi:10.5194/acp-13-4429-2013, 2013.

See also. . ..

Wiegner, M., Gasteiger, J., Groß, S., Schnell, F., Freudenthaler, V., Forkel, R., 2011. Characterization of the Eyjafjallajkull ash-plume:potential of lidar remote sensing. Physics and Chemistry of the Earth. doi:10.1016/j.pce.2011.01.006.

Section 3.2

I miss a Table with technical details of the ceilometer, what, for example, is the pulse rep rate? And what does it mean. . .. Assuming that the pulse energy of 8 micro Joule given by Flentje is true. Do you know the technical details or not? Are you sure that the overlap is completed in 1500m height? Do you know it? Yes or no? A calibration measurement of the ceilometer was not provided. . .. . We take one from CALIPSO for 1064 nm. All this is really not easy to accept and tolerate by a reviewer trying to teach his student day by day to do always careful calibrations and high quality observations.

Section 3.3.

P12, l9: COSMO-ART uses six size classes, and class 1 includes all particles with diameters up to 2 micrometers, so all Aitken and accumulation mode particles and the rest of all fine mode particles (up to a diameter of 1 micrometer), and then also a large fraction of the coarse mode (particles with diameters from1 to 2 micrometers) are altogether in just one single class. Is that justified from the optics point of view? I do not believe. . .

P12, l22: molecular number density of standard air was calculated. What do you mean? Do you use, for simplicity, standard atmospheric conditions? Is that justified? Schumann paper is referenced asd ACPD version, this in not acceptable, you need to take the ACP version. . . What size distributions did Schuman et al measure. . .? Should be used as a guide in all the modeling approaches.

Section 4 is really not needed if you take a simple demonstration case. . . I will not give

comments, except one: You cannot provide any uncertainty estimation (e.g. Fig 15) as long as you do not know what the truth is. We know already that neither the spherical nor the spheroidal particles shape models works in the case of irregularly shape dust particles (Gasteiger, 2011, Kemppinene, 2015).

Section 5 Results

The Schumann et al size distributions may show approximately the size distribution characteristics. Your size distribution cases 1 and 2 are just speculations. Case 1 (this COSMO modeled size distribution) appears to some extend realistic. Table 3 contains the modeled lidar ratio, and for the main classes 1,2 and 3 (case 1) the lidar ratios are 13, 4.6 and 10 sr, and on average probably close to 10sr and therefore far far away from the real-world values of 50-60sr.

As mentioned be careful when using Mediterranean Eyjafjalla dust papers, the dust maybe already widely mixed with other aerosols. And the paper of Mortier provides some column integrated lidar ratios and not values for well defined isolated lofted dust layers.

All in all, section 5.2 is so confusing and speculative, and simply ignores literature findings. There is also attempt to check the literature for observational hints that would corroborate all there speculative arguments... This part of the paper is clearly inacceptable.

Section 5.3 Qualitative Comparison

...provides no concluding remarks and any take-home messages (zero!) Again, and very clear: The shown demonstration case is a disaster for entire article and must be removed.

Section 5.4 Quantitative Comparison

Because of lack of any harmony between the model results and ceilometer observations (a factor of 60 difference!), the authors start to play around with the output (attenuated backscatter values) of the forward operator and reduced the output by a factor 10, 20, 30 etc. Why do you not check other observations (EARLINET lidar observations, even the CALIPSO overflight is available) regarding the truth! So discussion remained open. . . at the end.

Figure 4 shows particles in diameters, Figs 8,9, etc show particles in radius, Figure 16 shows particle size. . . . . ., please harmonize!

Figure 17 shows COSMO ART results ..with shape sensitive lidar ratios at all around 10sr. All colors are just grey. Nevertheless this figure corroborates best that this demonstration case is a disaster. Measured 'real world' Eyjafjalla lidar ratios are a factor of 4 higher. The discrepancy is to my opinion just caused by the shape of the volcanic particles.

Figures 18 and 19 present almost the opposite of what I would like to see, and what would be possible to show if a simple European pollution day would be selected. It is hard to believe that the ceilometer output (top panels) have something to do with the COSMO ART output (bottom panel).

---

## Author Comment (AC1) · 11 Nov 2016

*We thank the reviewers for reading the manuscript and providing detailed comments which helped to improve the manuscript substantially. We have carefully considered all comments and changed the manuscript accordingly. Please find our detailed point-to-point replies to all comments below in italic and blue. Changes made to the manuscript are shown in red.*

**Anonymous Referee #1**

**1 General comment:**

The paper describes a lidar forward operator based on basic lidar equations for calculating the backscatter and extinction coefficient profiles from a atmospheric chemistry model (ACM). This forward operator can than be used to compare aerosol concentration in the model and at a later stage assimilate lidar backscatter profiles into ACMs. Beside the method, one special case study namely the ash plume of the Eyjafjallajökull volcanic eruption is used in the manuscript to demonstrate the applicability of the forward operator.

I appreciate the approach of calculating backscatter and extinction coefficients directly from the particle size distributions. Thus this is an appropriate and scientifically relevant contribution to ACP in general. The manuscript is also well organized and the analysis and results are clearly communicated and contextualized. However, I recommend major revisions due my major concerns listed below, i.e. the issues should be clarified and partly revised before the manuscript can be accepted for publication.

**2 Major concerns:**

1. In the manuscript, a very prominent but also very special case is selected for the case study. Earlier studies already showed that these volcanic ash particles were very aspheric and are therefore not the optimum case to test the forward operator under "normal" atmospheric aerosol loadings. Especially with the assumption of spherical particles. Would it be better to have chosen a more simpler case with more spherical aerosol particles available ? Can you please comment on the motivation for using this case study.

*Our initiative for developing our forward operator is related to the dramatic eruption of the volcano Eyjafjallajökull in 2010 which had a huge impact on air traffic transportation over Europe. The UK Met Office is currently preparing to related events by setting up a small network of aerosol lidar systems. Therefore, many forecast centers are working on the improvement of simulations of these events so that our project and the operator can make major contributions in the future. Particularly, immediately during the event, the question came up of whether and how lidar instruments, especially the existing automated lidar ceilometers networks can deliver valuable information for validating the volcanic ash distribution simulations. During the event, only satellite data with very limited spatial resolution and nearly no quantitative information regarding aerosol mass concentration were available; thus all safety actions had to be based on numerical simulations with large uncertainties. Based on such model data without experimental input, the air traffic over Europe was shut down for several days causing economic costs of several billions of Euros. To become able to validate simulations of their COSMO-ART model using the existing ACL networks, DWD decided to start a dedicated project on ACL forward operators. Consequently, the aim of this study was not to find a simple case study for the forward operator but to develop a forward operator which is able to deal with such dramatic events.*

*Furthermore, it is important to point that other case studies, e.g., also not these suggested by the reviewers are "simpler". It may be the case that the aerosol microphysics of volcanic outbreak studies are more complex; however, it is much better possible to constrain here the emission of the point source, which was subject of many E15 studies, e.g., in Pappalardo et al (2013). Other so-called simpler cases such as a Saharian dust outbreak require excellent knowledge of the spatially distributed aerosol sources, which are – unfortunately largely unknown. Consequently, the better knowledge of aerosol microphysics is replaced by poor knowledge of the initial aerosol distribution.*

*Therefore, we do not agree that the interpretation of other cases is simpler. Last but not least, it was not subject of this project to simulate other cases due to lack of computing ressources as well as lack of representation of aerosol sources and microphysics. Thus, we decided to focus on the E15 case, which is fundamental for the understanding and tests of our operator.*

➔ *We have clarified the importance and the reason for the case selection as beginning of the introduction (p. 2, l 9-14),*

2. As you mentioned on page 12 l. 27. Schumann et al. 2011 found a mean aspect ratio of about 1.8 for particles smaller 500 nm and about 2 for larger particles. This particles where sampled in-situ with the Falcon aircraft and should provide the most realistic parameters for volcanic ash particles in the atmosphere. That means that the ash particles observed where more aspheric with higher aspect ratio than it was chosen in this study. I'm sure, if you would enhance the aspect ratio up to 2 in the T-matrix calculations which is actually the mean for larger particles than 500 nm, it produces even larger deviation in comparison to the Mie calculations. Maybe the impact is smaller than I think, but this has to be shown in a more extended sensitivity study covering the whole range of aspect ratios. The seconds questions concerning the T-matrix calculation is, why do you have chosen a smaller aspect ratio of only 1.25/0.8 for the cylindrical shape? This should also be extended to higher aspect rations of about 2 to cover the range of the observations. Maybe it would also be good to add one comparable DDA simulations to the sensitivity study to see the full range of uncertainties even though it is only possible for sizes smaller 6.2 µm. However, this particles constitute the largest contribution to the total number concentration. Maybe it is also not worth to generate the same Figure 17/18 for one type of aspherical scatterers with the T-matrix method.

*The major problem for representing non-spherical scatterers is to find an appropriate representation. We could follow the findings of Schumann et al. 2011 and represent the volcanic ash by particles with an aspect ratio of 1.8 or 2.0 – but for which particle shape? Observed ash particles were complex shaped and different to each other; some were more cylindrical, others ellipsoidal, cubic, fractal, spherical, or had even more complex shapes. This problem cannot be solved by finding "one ideal representative shape". Thus we decided to use a certain particle shape and analyzing the uncertainties due to this assumption. Please note that we provide in this manuscript the first uncertainty analysis of this type for the Eyjafjallajökull eruption in 2010. Certainly, this analysis can be further refined in future studies based on our present work.*

*Concerning the second part of the question, please find the updated section 4 of our manuscript where we extended the particle shape sensitivity study with ellipsoids and cylinders having an aspect ratio ranging from 0.5 to 2 as concerned by the reviewer. It seems like the extremely asymmetric particle shapes are not in general responsible for the extreme values of the relative difference of the effective backscatter cross section related to spheres.*

*Finally, we would encourage anyone to perform detailed scattering analysis using DDA for the relevant scatterer shapes and sizes. Calculating the orientationally-averaged backscatter efficiency of a cube with a width of 2 times the wavelength (1064 nm) for one real and imaginary part and one shape took us several hours in 2013. If we repeat this calculation ideally for more than 20 real parts, 20 imaginary parts, 200 radii, and 10-20 shapes, the look-up table creation would take several years. Reducing the computation time by two orders of magnitude would even not be sufficient. The t-matrix solution - in comparison - takes about 0.02 to 15 seconds per particle shape depending on the particle-size-to-wavelength-ratio and the aspect ratio of the particle.*

*In conclusion, our simulations are now in agreement with the experimental results and provide for the first time an approach to implement these in a forward operator including a detailed uncertainty analysis. We are convinced that this study provide a significant contribution to the operator development for the aerosol and atmospheric sciences communities. Reworked the particle shape sensitivity study*

➔ *We have increased the extent of considered particle shapes for the t-matrix sensitivity study as proposed by the reviewer.*

➔ *We have combined Figures 8,10, and 14 as well as Figs. 9,11, and 12*

3. To make a fair comparison of ACL data with the model/forward operator all major uncertainties, i.e shape, refractive index, calibration constant, have to be estimated and combined to show if the statement concerning overestimation of particle number in the model is justified. As the authors mentioned in the text, the backscatter coefficient is much more affected by the particle properties than the extinction coefficient. Thus, the low lidar ratio with most values around 10-15 in comparison to the other studies by Kokkalis et al, 2013; Ansmann et al. 2010 and Mortier et al., 2013. is a strong indication for an incorrect/overestimated backscatter coefficient and could also explain the large discrepancy between modeled and measured backscatter coefficient profile.

*Thanks for pointing out this very interesting question.*

*We agree with the first part of this statement which is exactly our approach: We calculated the attenuated backscatter coefficient profile and estimated the expected errors of our method. If the difference between forward modeled and measured lidar profiles exceeds the expected error, the particle number concentration may be over- or underestimated by the model.*

*The considerations of the second part was also noted by anonymous reviewer #2 so we decided to double-check the results as we agree that these low values of the average lidar ratio seem to be even beyond the scope of uncertainties related to the particle shape effect. We (in combination with further suggestions from reviewers #1 and #2) reworked the sensitivity studies and section 4 of our manuscript.*

*After calculating the look-up tables using t-matrix (instead of the IDL code mie_single) and writing a new calculation routine for the effective scattering cross section calculation using Rscript (instead of IDL), we got nearly the same results as before. Surprisingly, the newly created plots of lidar ratio versus equal-volume radius showed for the whole first size class higher lidar ratio values than we found as calculated average lidar ratio.*

*The error was simple due to the fact that we calculated the average lidar ratio from the effective values of the extinction and backscatter cross section listed in Table 4 instead of calculating the average lidar ratio of each size class depending on the lidar ratio spectrum of the size class as shown by Fig. 10 in our manuscript.*

*Consequently, the new scattering calculations are now in close agreement with obervations. The correctly calculated average lidar ratio values have been updated in Table 4 of our manuscript. The average lidar ratio of all classes is 61 sr and fits very well to the observations of other groups.*

*Regarding the lower values of the lidar ratio, we would like to mention that Gasteiger et al. (2011) "Modelling lidar-relevant optical properties of complex shaped mineral dust aerosols" found lidar ratio values lower than 8 sr$^{-1}$. This also shows that for a model which differentiates size classes, the assumption of a fixed lidar ratio may lead to unpredictable errors and uncertainties. We hereby emphasize again the need for lidar forward operators which are not based on the assumption of a fixed lidar ratio.*

- ➔ *Double-checked the scattering calculations*
- ➔ *Double-checked the calculation of the average lidar ratio values shown in table 4*
- ➔ *Updated the average lidar ratio values in Table 4 by validated values*
- ➔ *Updated Section 4*
- ➔ *Added Fig. 9 which shows the lidar ratio spectrum over size for spherical and several non spherical particle types*
- ➔ *Added this discussion to the manuscript, namely Section 4.3 as well as P19, l 32 to P20, l 4*

Only if both backscatter and extinction coefficient are higher in the forward operator simulated profiles in comparison to the ACL data, than this is an indication for a too high particle number concentration in the model. But this cannot be checked with the ACL data, because of missing independent extinction data. In summary, the conclusion of a wrong number concentration cannot be drawn from the case study with so many simplifications and without considering the errors/uncertainties of the forward operator.

I recommend to provide a comprehensive estimation of the combined uncertainties with consideration of point 2 above.

*This is correct but there is still huge information content in the aerosol lidar data. Deviations need to be characterized more in detail in the future; however, we are already convinced that large deviations in the backscatter profiles are mainly due to errors in atmospheric dynamics. This deviation can easily be detected in model-observation (M-O) statistics and will be corrected, if just ceilometer data are assimilated in the future. Otherwise, currently, there are only very few extinction profiles available. Please note that there are worldwide only a very few automated Raman lidar systems yet (we know of only 4). Their information can easily be analyzed without forward operator as well because it separates between extinction and backscatter profiles. This is a great advantage and future potential. Currently, it is important to study which information can be obtained from the existing networks of automated lidar ceilometers (alone in Europe more than 100 of these instruments are in operation, see http://www.dwd.de/ceilomap), which can be realized with our operator by combining the backscatter and extinction profiles*

*Deploying dense networks of advanced lidar systems which measure these profiles directly is desirable and would allow for using (maybe this) lidar forward operator based on backscatter and extinction profiles with lower uncertainties.*

*Simplifications are, however, required tools for representing real-world problems in a numerical description efficiently. This allows for stating that a fictive true value is within an uncertainty interval. If the remaining difference is greater than the forward modeled value plus uncertainty interval, this difference is likely to be due to the reference dataset, i.e. the modeled number concentrations.*

*We agree that the case study suffers from wide uncertainty intervals (consider for example the ACL data calibration) but these current limitations have to be published to give insight into the requirements for subsequent forward operators designed to perform ACL data assimilation in the future.*

➔ *We have added this note on future ACL networks in the quantitative comparison (P17, l8-10)*

➔ *The discussion of the extinction and backscatter profiles is already included in the manuscript, see P18, l19-32*

4. Some statements are to strong and cannot be drawn from this case study and from the implementation of the forward operator with it's huge uncertainties regarding refractive index and particle shape. For example in the conclusion section (p. 19, l. 13-14): How do you want to constrain or correct the model regarding shape and refractive indices ? Your forward operator use the refractive index as input and larger differences can also be expected from the simplicity of assuming spheres. I don't see a chance to retrieve refractive indices or shapes with the forward operator in combination with the ACL measurements, and thus this statement is not reasonable. In addition, the forward operator needs always additional information like the refractive index. So it is hard to apply the forward operator as a general/operational tool or even for assimilation without having these information available. So that the statement of a "crucial step" written in the abstract or similar in the conclusion is little over emphasized.

*We agree that this statement is overemphasized as the forward operator currently aims on assimilating the particle number concentration. However, there is potential to realize this by application of our forward operator to multi-wavelength lidar systems. The operator is designed accordingly.*

➔ *We have removed the sentence at page 19, line 12 and 13 in the manuscript.*

**3 Specific comments/questions:**

• Why are you using the Mie code with the assumption of spherical particles? Instead you could create a similar lookup table for aspherical particles with different aspect ratios and refractive indices.

*Our simulations confirm that the results obtained for spherical scatters do not differ drastically from aspherical scatterers when averages over the size bins are taken. The differences obtained from the simulations are taken as estimate for the uncertainties. So we are in fact using the results. But it should be noted that ellipsoids are not necessarily better than spheres as the real shapes are anyhow much more complex and cannot be taken for the simulation so far. Here further progress in scattering theory is required.*

• p. 11, l. 21-25: Calipso is used for estimating the calibration constant of the ACL lidar systems. How is this estimation performed? Usually, there are larger differences in space and time between local and satellite observations beside the viewing geometry. The brings me to the question how large is the uncertainty of this method also regarding to point 3 and also point 2?

*The ACL calibration was done by two criteria. First, we used the CALIOP value of the 1064 nm calibrated attenuated backscatter coefficient at 50.15° lat / 4.81° lon in a height of 2 km ASL which is about $5 \cdot 10^{-6}$ $m^{-1}$ $sr^{-1}$ (Fig. 3 in the manuscript) to determine the maximum attenuated backscatter coefficient which should be measured by the ACL system inside the volcanic ash cloud. The second calibration criterion was the forward modeled attenuated backscatter coefficient outside the volcanic ash layer in 3.0 km ASL which is dominated by molecule scattering and has a value of about $1 \cdot 10^{-7}$ $m^{-1}$ $sr^{-1}$ (Fig. 18 in the manuscript).*

*We agree that this pragmatic approach is of very limited precision. However, with this calibration, both the volcanic ash and the molecular background attenuated backscatter values correspond in magnitude which (roughly) validates this approach. We estimate that the calibration at least is accurate to an order of magnitude.*

➔ *We have added the explanation of the calibration to section 3.2. (P11, l12-21).*

• p. 13, l. 13-15: Why should an ellipsoidal distribution lead to less realistic scattering calculation than spherical scatterers? This statement is a little bit strange, as you showed in the sensitivity study that more aspherical scatterers can better represent the observations. This statement should be removed or at least better explained.

*Both statements are valid. Yes, more aspherical scatterers may represent the observation better. But which type of aspherical scatterer leads to the best results cannot be decided. Cylinders and ellipsoids with the same aspect ratio have very different scattering properties so assuming either one would result in higher errors than using spheres. Thus, we prefer at the current stage sphere as best compromise and use the uncertainties in the analysis as discussed. Please note that this approach is not unusual but that all other studies sofar – to best best of our knowledge – do the same. We have even refined the approach by averaging over the size classes of te model and discussing in detail the uncertainties.*

➔ *We added this consideration to section 4 at P16, l3-10*

• p. 16-17, l 34-2: The modeled structure can also be strongly overestimated by the forward operator and thus leads to visible structures in the simulated profiles. These structure could be real but just below the detection limit of the ACL measurements.

*Thanks for this clarification!*

• p. 18., l. 8: It is written that both backscattering and extinction are higher compared to ACL observations. But the ACL cannot measure extinction profiles directly. So how did you compare the extinction profiles to the ACL observations ?

*Due to the definition of the two-way transmission (see eq. 24 in our manuscript), strong extinction results in a strongly reduced backscatter signal intensity behind. In Fig. 18 this "shadowing" is represented by a dark grey shaded area above the volcanic ash layer. The attenuated backscatter coefficient above the volcanic ash layer is about 100 times lower than the attenuated backscatter coefficient in the same height but without volcanic ash particles. This is equivalent to a two-way transmission of about 1% which would be expected for light fog but not for thin volcanic ash layers.*

➔ *The discussion of the extinction and backscatter profiles is already included in the manuscript, see P18, l19-32*

4 Technical comments:

• In the text and the especially in the figures a mixture of diameter and radius is used for the particle size. Particle size is sometimes written without mentioning either radius or diameter. I suggest to stay with one notation to avoid any confusion.

*Figure 4 is a visualization of the model set-up which is defined by the particle diameter while the other figures are related to scattering calculations given in radius. We believe due to this fact that it is better to leave these figures as they are. Due to redundant information of Fig. 16 and Tables 4 and 5, we removed Fig. 16 however.*

➔ *We removed Fig. 16 as it contains redundant information.*

• Figures 8-11: I suggest to combine Figure 8 with 10 and Figure 9 with 11. There are not large differences in the outcome of the each figure pair and therefore the benefit of splitting the figures is low. This would reduce the number of figures which makes the manuscript more clearer. In addition, I recommend to scale the extinction cross section from 0-300 and the backscatter cross section to 0-25.

The small zoom plots can be moved to the white spaces in the lower left corner or below the legend. This would enhance the visibility of the differences between the different types.

*We agree that these plots have been sub-optimal. We have created better ones instead with a clearer interface and also created a full plot for the results of the first size class as part of the figures.*

➔ *Figs 8, 10, and 14 have been combined in one figure (as well as Figs. 9,11, and 12).*
➔ *To improve the visibility of differences for the first size class, the results of the first class have been added as separate plot to the respective figures.*

• p.11 l 18.: How large is the measurement error of the diameter to state a number with such an accuracy ? It seems the number of the aperture of the receiving telescope has to much decimal places. I suggest to reduced the value to only two decimal places. This would account for a measurement error for less than 1mm in diameter.

➔ *We reduced the number of decimal places as suggested.*

• p. 8: There are a lot of basic formulas in the manuscript which makes it hard to read. For example the formulas regarding molecular scattering can be referenced and thus removed from the text. The formulas 14-17 can be combined etc. This would make the text more clearer.

➔ *We replaced eq. 7 and 8 by a reference to eq. 5 and 6.*

• p. 14 l 30: spelling error: with a diameter

*Thanks..*

➔ *Corrected*

*Anonymous Referee #2*

**1 General remarks:**

The paper deals with an attractive and important research topic. The development of a backscatter lidar forward operator is very useful to support aerosol transport modeling and efforts to forecast aerosol conditions.

*We thank the reviewer for these positive words.*

However, the paper is a pure (100%) technical, methodological paper and thus appropriate for AMT. I therefore recommend to move it to AMT in case that it gets finally accepted.

*We do not agree with this statement. This is clearly not a pure technical paper because we also provide comparisons of model output with observations. This is not subject of AMT but a topic which fits very well to ACP.*

Unfortunately (I must say), the paper is not acceptable in its present form. But I have hope that it can survive. I was expecting a straight forward paper, a clear, compact description of the nice method and then also a nice and easy to follow (and convincing) case study (demonstration case) which clearly corroborates the usefulness of the methodologic concept and developments. But all this is not presented! The method is introduced in a rather lengthy way, text-book knowledge is outlined in extended detail. This can easily be avoided. But! .... The demonstration case (Eyjafjalla volcanic aerosol scenario) is simply the worst case they could select! This case must be substituted by a case for which Mie scattering can be applied.

*It is difficult for us to react to these rather vague and confusing comments. We are convinced that the set up and the contents of our work is very concise and cannot derive from the reviewers comments anything helpful, if just statements like "all this is not presented!" are made. Where is missing something and why? We expect from a high-quality review that simple and general statements are clearly substantiated by scientific arguments. Unfortunately, these are largely missing. Another example is the statement that the E15 case is "simply the worst case". What case is worse and what is simpler and why? Again, this statement is also not substantiated. In contrast, we have many scientific arguments, which disprove this statement of the reviewer (see below on page 10, l 11-23). In the literature, there is unfortunately quite some confusion about the nomenclature of scattering theory and related lidar equations (e.g., factor 4 pi in the backscatter cross section or not) so that we believe that the equations are beneficial for a detailed understanding.*

*The Eyjafjallajökull case – as we are stressing now even more clearly in the revised manuscript – was the most dramatic event in recent years regarding the impact of aerosols on society and economic loss. For this event, only ACL data are available with sufficient spatial and temporal coverage to investigate details of the dynamical evolution. E.g., EARLINET data as consisting of non-permanent operating instruments which have to be started manually sampled unfortunately only part of the event at some distant stations in Europe. ACL data, however, are automated and thus continuously available, and available from more than one hundred of stations in Europe (see www.dwd.de/ceilomap).Thus, it this is important to analyze also the use of these continuously operating ACL networks. Currently, the UK Met Office is setting up a small network of lidar systems, which is entirely dedicated to observe corresponding events.*

*We hope that this is convincing to accept that our study is a significant contribution to further advance the research on atmospheric aerosols with the help of a forward operator.*

➔ *We have clarified the importance and the reason for the case selection as beginning of the introduction (p. 2, l 9-14),*

➔ *We have explained the benefits of selecting a case study of a volcanic eruption (p. 10, l 11-23),*

➔ *We have clarified the benefits of the forward operator further in the abstract (p. 1, l 25-29)*

The methodology is based on the fundamental assumption that the aerosol particles can be handled as spheres (as so called Mie particles such as marine aerosol particles or urban haze or biomass burning smoke particles). This is well justified in the case of anthropogenic pollution including biomass burning smoke. However, the demonstration case, the authors selected, deals with volcanic dust particles crossing Germany quite shortly after emission so that nothing is well known, size distribution, chemical composition, shape properties. Such a situation occuring on 16-17 April 2010 has never been observed before with advanced atmospheric technology. Fortunately, several EARLINET lidars were able to pick up even the first traces of the heavy dust front which crossed Germany on 16 April 2010. The measured lidar ratios indicated irregularly shaped volcanic dust particles. The AERONET observations revealed that these fresh plumes were coarse mode dominated. So it was by far not a normal aerosol situation. It was an extreme situation! As mentioned, nothing is well known, the size distribution of the volcanic dust particles is not known, the shape properties are not well known. Even the chemical composition could not be well characterized. Aircraft observations were started several days later, when the aerosol mixture and physicochemical properties already had changed significantly, and spherical sulfate particles had formed from the emitted SO2, and were mixed with the voclanic ash and dust. On 16 Apri 2010, it is most likely that only dry ash and dust particles were present. And for such a case, Mie-scattering-based methods (including this backscatter lidar forward operator method) can definitely be not applied, as it is shown in this paper!

*The first comment of the reviewer is incorrect and incomprehensible to us. Particularly in this work, we are performing new calculations to handle also non-spherical particles. Consequently, we are fully aware of these uncertainties and already dedicated a full section (section 4) to extensive particle shape and refractive index sensitivity studies where the uncertainties of the forward operator related to uncertain volcanic ash properties are estimated. **It is seems so that the reviewer oversaw this section.***

*This method is fundamental for any realistic case study and requires the definition of a reference particle with fixed properties. Discussing whether this reference particle should be spherical, ellipsoidal, cylindrical, or of any other shape is not decisive– it is straightforward to use spherical particles as a reference(just as reference!) for calculating the maximum difference toscatterers with other properties than thisreference scatterer. Please note that the same is done by many other studies. Changing the reference scatterers shape for example to ellipsoidal, we would also have to perform the same tasks which includes to measure what happens if this assumed reference scattererwould be spherical instead.But even then, the real particle shape is much more complex and it is at-date fully unclear whether the results with ellipsoidal scatterers are closer or even further away from reality than the results obtained with spherical scatterers. Please note that the forward operator is not responsible for defining number-size distributions or applying chemical interactions – that is part of the aerosol models.*

➔ *We have revised section 4.3 of our manuscript,*
➔ *We have noted that such sensitivity studies are mandatory except for aerosol which are perfectly spherical (p. 14, l 13-14),*
➔ *We have clarified this particle shape issue (p. 17, l19-25).*

The authors are forced to discuss this case in a very speculative way so that the main question at the end was not: What do we learn from this paper? No, the main question was: Why did they choose this incredibly complicated case so that it was, at the end, impossible to demonstrate the usefulness of the new technique?

*Where is our work speculative and why "it is impossible the demonstrate …"? We also do not agree to characterize this case as "incredibly complicated", which is another statement neither substantiated by the reviewer nor considering that other cases are also suffering from large uncertainties. The new findings of our study are clearly described in the conclusions. A forward operator in combination with an uncertainty study is an essential tool for studying and understanding atmospheric processes. It is fundamental to provide the basis for future data assimilation studies. Such a forward operator is currently lacking not only for volcanic ash studies but also a large number of other aerosol types. As we found out, a spherical model is not too far off from non-spherical models if averages over certain size distributions are taken. This new approach, described in our manuscript to the best of our knowledge for the first time (!), reduces sensitivities to uncertainties regarding the shape and refractive*

*index significantly. Thus, we certainly demonstrate the "usefulness of the new technique" with important case. It should be noted, that we cannot expect agreement between measured and forward-operator-based simulated lidar profiles because the model number densities cannot be expected to be accurate.*

*The reviewer is wrong in assuming that only the forward model is validated by the comparisons. On the contrary, the differences between measurements and simulation reveal also the shortcomings of the atmospheric model. Because we have estimated the uncertainties of the forward model, we can conclude on remaining differences which can be attributed to deficiencies in the modeled number density of the volcanic ash particles.*

So, my conclusion is not only,...the paper needs major revision. The conclusion is, we need an easy case that shows the applicability of the forward operator model! This paper definitely fails to show that. Without such a simple case, the paper must be rejected! To be fair, many papers of this quality get rejected at this stage, already. However, I believe that there is enough and substantial material to save it! But only if we have another, a much better, much easier demonstration case.

*We disagree with these – again quite confusing - statements of the reviewer. Does the manuscript need major revisions or must it be rejected or can it be "saved" from the point of view of the reviewer? We are not sure what the reviewer means here.*

*Regardless from this interpretation, we again emphasize that selecting a pseudo-simple case study just to avoid problems is not possible. Also, the reviewer is lacking any information what a "simple case" is. We are not aware that any other cases are simpler; let it be a Saharian dust outbreak with largely unknown aerosol sources, which must be parameterized in the model, or a pollution source with even more unknown sources and complex chemical conversions. Obviously, taking these points into account, the E15 is clearly comparable in complexity due to the much better knowledge of the point source studied, e.g., in Pappalardo et al (2013). As already explained above, the selected case has the advantage that the location of the source is well known and its strength can be well estimated; much better than for any other type of distributed aerosol sources. Furthermore, as the aerosol particles are advected in the middle troposphere, these can be clearly identified in the observations and much better differentiated from the aerosol background than other sources, e.g., pollution, which is already mixed with other sources in the atmospheric boundary layer. Consequently, thinking more generally and deeply about all uncertainties, the E15 case may be more complex with respect to the representation of aerosol microphysics but it is much simpler with respect to the assumptions and the simulations with respect to the source. In contrast, huge deviations in the simulation of size distributions and total number densities will still occur with respect to the observations, if other cases (Saharian dust outbreaks, pollution cases, etc.) are considered.*

The conclusions based on this volcanic case the authors selected will always remain rather uncertain and thus speculative because of lack of sufficient knowledge of the relationships between the optical properties and the size/shape/composition properties of Islandic volcanic ash 2 two days after emission. There is no hope!

*The scientific topic is the analysis of the quantitative results of our forward operator and its uncertainties. Consequently, our approach is the opposite of being "speculative" as it allows for quantifying the errors due to uncertain information of each particle class provided by the atmospheric chemistry model. The range of uncertain refractive indices and shapes was chosen such that even huge changes of the ash optical properties would also be covered by the sensitivity / error analysis (see section 4 in our manuscript). By applying the size-averaging algorithms we demonstrated in section 4.2, the maximum error due to uncertain refractive indices could be reduced to a factor of less than 3. We even analyze the effect of uncertain particle shapes and perform an error analysis of the spherical in comparison to 11 asymmetric particle shapes. This is explained extensively in section 4.3 of our manuscript and allows for quantifying the uncertainties which is in contrary to being speculative!*

*Furthermore, we believe that any scientific progress is only possible by analyzing challenging and important cases. As explained above, the Icelandic volcanic ash is a corresponding case. There is no reason to dismiss its scientific analysis.*

The paper has many other weak points. It is not well written. The reference list is far away from reflecting the aerosol lidar science field in a proper way. Even in the case of the volcanic aerosol plumes the authors leave out to mention essential papers. All this must be improved.

*Again, the reviewer is making unsubstantiated comments, which makes it very difficult for us to respond to it. We think it is just not sufficient to make an easy state that a paper "is not well written" but we expect that the criticism is founded by arguments. Furthermore, we had appreciated very much, if the reviewer had substantiated the statements about missing work by examples. However, we are very happy to add references, like some of the references which were proposed by the reviewer.*

➜ *We have added the following references:*

➜ *Cuevas, E., Camino, C., Benedetti, A., Basart, S., Terradellas, E., Baldasano, J. M., Morcrette, J. J., Marticorena, B., Goloub, P., Mortier, A., Berjón, A., Hernández, Y., Gil-Ojeda, M., and Schulz, M.: The MACC-II 2007–2008 reanalysis: atmospheric dust evaluation and characterization over northern Africa and the Middle East, Atmos. Chem. Phys., 15, 3991-4024, doi:10.5194/acp-15-3991-2015, 2015.*

➜ *Groß, Silke und Freudenthaler, Volker und Wiegner, Matthias und Gasteiger,Josef und Geiß, Alexander und Schnell, Franziska (2012) Dual-wavelength lineardepolarization ratio of volcanic aerosols: Lidar measurements of the Eyjafjallajökullplume over Maisach, Germany. Atmospheric Environment, 48, Seiten 85-96. DOI:10.1016/j.atmosenv.2011.06.017.*

**2 Point by point....:**

Abstract:

The abstract must be rewritten: please provide a compact text with the goal of the article, the methods used, and the essential finds. That's it! All the motivating and explaining statements should not be given in an abstract. The right place for such information is the Introduction (section 1).

➔ *We have followed this comment and shortened the abstract to improve its quality.*

**Section 1 Introduction**

P 2-4: The introduction can be kept much shorter and more compact. Many details are given that have nothing to do with the goal of the paper.

➔ *We have followed this comment and shortened some points. However, we added a few more lines in order to address the comments of anonymous reviewer #1.*

P2, l30: Please check Cuevas, MPL observations and lidar data assimilations into models (see my reference list below).

➔ *We have added this reference to the manuscript. However, this paper does not describe our new scientific approach.*

P3, l1-15: please remove the first paragraph, please focus on aerosol lidar only. The work of Japanese groups to provide lidar data for the modeling community should be mentioned in this context. They were pioneering in the 1990s and are even active presently to combine lidar and atmospheric aerosol modeling. There are also papers in which lidar data are used to validate models and verify results from the EARLINET community (check the speciall issue of EARLINET in AMT, and the papers including there reference lists). There were also efforts to combine lidar and model results in the Tellus SAMUM special issues of SAMUM (Tellus 2009, 2011). In the case of multiwavelength lidar (here the authors probably mean inversion techniques), I was surprised that the authors left out to provide the references to the fundamental Mueller, Veselovsky and Boeckmann papers! These authors are active in this field since more than 15 years. Mueller (JGR 2010, 2012) published several papers on the effects in optical closure studies when the scatterers are non spherical. . .and , if remember right, even not of spheroidal nature.

*Of course, there are many papers on aerosol research with lidar. We are fully aware of this. But complicated multi-wavelength retrievals are not the focus of the present manuscript; same as depolarization lidar measurements etc. This may explain the surprise of the reviewer, which is, however, not decisive for the choice of the references here. As clearly explained in our introduction, our focus is to study the use of automated lidar ceilometers = simple one-wavelength backscatter lidar systems. And please note, that respective forward operators for these systems (multi-wavelength Raman lidars) do not exist either! Of course, it is reasonable to start with simple backscatter lidar systems as we do before coming to very complex research lidar systems which are anyhow far from being continuously operational and forming dense networks.*

➔ *We have changed the retrieval =>  inversion issue*
➔ *As our manuscript aims on quantitative comparisons using a forward operator we did not add references related to quantitative model-observation comparisons or retrieval papers.*

P3: Please add somewhere the basic work of Heese et al and Wiegner and Geiss in AMT on the characterization of ACLs (see my list below). All in all I was not very happy with the introduction section. It gave me the impression that the authors were not willing or not able to provide a well-balanced literature overview on aerosol lidar. This needs to be significantly improved and will certainly strengthen the quality of the paper as a whole. A good literature review always provides the impression, the work is done in a professional way. The other way around, . . ..provides the opposite feeling. Here are some papers, that should be cited. . .

*We are not happy about this unfair statement of the reviewer. The subjective "impression" of the reviewer is not correct. Some authors of this manuscript have 20 and more years of experience in lidar research. Suggesting that the work would not be made in a professional way is thus simply not acceptable. Of course, we are very well aware of the previous work in aerosol lidar research but it makes no sense to cite all hundreds existing publications on ACL. The references suggested by the reviewer may reflect his/her personal interest/work but these are mostly not relevant to the topic of the manuscript. The reviewer should note that this manuscript is not focusing on multi-wavelength Raman lidar retrieval algorithm techniques. Thus we decided not to include most ofthese references of the reviewer.*

Heese, B., Flentje, H., Althausen, D., Ansmann, A., and Frey, S.: Ceilometer lidar comparison: backscatter coefficient retrieval and signal-to-noise ratio determination, Atmos. Meas. Tech., 3, 1763-1770, doi:10.5194/amt-3-1763-2010, 2010.

*A paper on retrieval algorithms (invers approach of forward modeling)*

➔ *We have not added this reference to our manuscript*

Wiegner, M. and Geiß, A.: Aerosol profiling with the Jenoptik ceilometer CHM15kx, Atmos. Meas. Tech., 5, 1953-1964, doi:10.5194/amt-5-1953-2012, 2012.

*We already have included the reference to Flentje et al. (2010)which contains a very detailed description and discussion for measurements of the ACL network during the volcanic eruption.*

➔ *We have not added this reference to our manuscript*

An introduction of a paper dealing with a ceilometer forward operator should include such 'fundamental' papers to my opinion, especially devoted to Jenoptics ceilometers. Here, some further papers ( and one PhD work) on lidar data assimilation, and the references in these articles may help.

Please have a look and check!

http://www.mri-jma.go.jp/Dep/ap/ap1lab/member/tsekiyam/files/sekiyama_thesis.pdf

*Again a work related to forward operators which are based on fixed lidar ratio assumptions. We already included several references related to this method.*

➔ *We have not added this reference to our manuscript*

Wang, Y., Sartelet, K. N., Bocquet, M., Chazette, P., Sicard, M., D'Amico, G., Léon, J. F., Alados-Arboledas, L., Amodeo, A., Augustin, P., Bach, J., Belegante, L., Binietoglou, I., Bush, X., Comerón, A., Delbarre, H., García-Vízcaino, D., Guerrero-Rascado, J. L., Hervo, M., Iarlori, M., Kokkalis, P., Lange, D., Molero, F., Montoux, N., Muñoz, A., Muñoz, C., Nicolae, D., Papayannis, A., Pappalardo, G., Preissler, J., Rizi, V., Rocadenbosch, F., Sellegri, K., Wagner, F., and Dulac, F.: Assimilation of lidar signals: application to aerosol forecasting in the western Mediterranean basin, Atmos. Chem. Phys., 14, 12031-12053, doi:10.5194/acp-14-12031-2014, 2014.

Janiskova et al., Assimilation of cloud information from space-borne radar and lidar: experimental study using a 1D+4D-Var technique, QUARTERLY JOURNAL OF THE ROYAL METEOROLOGICAL SOCIETY Volume: 141 Issue: 692 Pages: 2708-2725 Part: A Published: OCT 2015 Campbell et al., CALIOP Aerosol Subset Processing for Global Aerosol Transport Model Data Assimilation, IEEE JOURNAL OF SELECTED TOPICS IN APPLIED EARTH OBSERVATIONS AND REMOTE SENSING Volume:3 Issue: 2 Pages: 203-214 Published: JUN 2010

*Works related to inversion approaches, not forward operators*

➔ *We have not added these reference to our manuscript*

Cuevas, E., Camino, C., Benedetti, A., Basart, S., Terradellas, E., Baldasano, J. M., Morcrette, J. J., Marticorena, B., Goloub, P., Mortier, A., Berjón, A., Hernández, Y., Gil-Ojeda, M., and Schulz, M.: The MACC-II 2007–2008 reanalysis: atmospheric dust evaluation and characterization over northern Africa and the Middle East, Atmos. Chem. Phys., 15, 3991-4024, doi:10.5194/acp-15-3991-2015, 2015.

*This paper fits very well so we have added it to the manuscript*

➔ *We have added this reference to our manuscript*

I also found recently somewhere (unfortunately I do not remember the web page) a workshop report on MPL lidar data assimilation into the NMMB model. The first author was again: Cuevas. So there are presently data assimilation efforts in the dust forecast scene.

*Indeed, there are lots of activities related to data assimilation of aerosols and we agree that this is a very important field of research.*

P4, l8: I sounds attractive and convincing when essential input parameters have not to be assumed. But in the case of the lidar ratio, it is frequently even better to calculate the robust extinction properties and then take a lidar ratio of 50 sr to obtain the backscatter coefficient. I would like to see if this option is mentioned too. For example, for volcanic dust (your case study) it is shown that the lidar ratio is around 50sr (Gross et al. 2012), and it would be simply much more straight forward to use such a number of 50sr than your rather unrealistic values close to 'modeled' 5-15 sr. I will come to this point later again.

To my opinion, one of the best Eyjafjalla volcanic lidar ratio papers is (please add it to the references and include the findings in the discussion):

Groß, Silke und Freudenthaler, Volker und Wiegner, Matthias und Gasteiger, Josef und Geiß, Alexander und Schnell, Franziska (2012) Dual-wavelength linear depolarization ratio of volcanic aerosols: Lidar measurements of the Eyjafjallajökull plume over Maisach, Germany. Atmospheric Environment, 48, Seiten 85-96. DOI: 10.1016/j.atmosenv.2011.06.017.

To my understanding, the Ansmann 2010 lidar ratios of 50-60sr (similar to the ones in Gross et al, for the lofted isolated volcanic ash layer. . .) probably correctly describe the dry volcanic aerosol lidar ratios in the beginning of the volcanic episode. Later (18-19 April and afterwards. . .) the pure volcanic dust and ash conditions were gone. Volcanic sulfate aerosols formed from the emitted volcanic SO2 so that a mixture of spherical and irregular shaped particles were always present. I mention it only to avoid that theauthors misinterpret the findings in the Gross paper. The lidar ratios were definitely inthe range of 50-60sr for dry volcanic dust (in the absence of volcanic sulfate). All this is very similar to desert dust, and the desert dust lidar ratio at 1064nm seems to be equal to the lidar ratios for 532 nm or even slightly higher than the ones for 532nm, gainedfrom 1064nm lidar/1020nm photometer observations.

*We agree that these low values of the average lidar ratio seem to be even beyond the scope of uncertainties related to the particle shape effect. We (in combination with further suggestions from reviewers #1 and #2) reworked the sensitivity studies (Section 4) of our manuscript. After calculating the look-up tables using t-matrix (instead of the IDL code mie_single) and writing a brand new calculation routine for the effective scattering cross section calculation using Rscript (instead of IDL), we got nearby the same results as before. Surprisingly, the newly created plots of lidar ratio versus equal-volume radius showed for the whole first size class higher lidar ratio values than we found as calculated average lidar ratio. This deviation was simply due to the fact that we calculated the average lidar ratio from the effective values of the extinction and backscatter cross section listed in Table 4 instead of calculating the average lidar ratio of each size class depending on the lidar ratio spectrum of the size class as shown by Fig. 10 in our manuscript.*

*Consequently, we have confirmed that our scattering calculations are in a good shape. The correctly calculated average lidar ratio values have been updated in Table 4 of our manuscript. The average lidar ratio of all classes is 61 sr and perfectly fits to the observations of other groups. Regarding the lower values of the lidar ratio, we would like to mention that Gasteiger et al. (2011) "Modelling lidar-relevant optical properties of complex shaped mineral dust aerosols" found lidar ratio values lower than 8sr$^{-1}$. This also shows that for a model which differentiates size classes, the assumption of a fixed lidar ratio may lead to unpredictable errors and uncertainties. We hereby like to emphasize again the need for lidar forward operators which are not based on the – from our point of view – insufficiently criticized assumption of a fixed lidar ratio.*

➔ *Double-checked the scattering calculations*
➔ *Double-checked the calculation of the average lidar ratio values shown in table 4*
➔ *Updated the average lidar ratio values in Table 4 by validated values*
➔ *Updated Section 4*
➔ *Added Fig. 9 which shows the lidar ratio spectrum over size for spherical and several non spherical particle types*
➔ *Added this discussion to the manuscript, namely Section 4.3 as well as P19, l 32 to P20, l 4*

The authors mention the Greek paper on volcanic dust of Kokkalis et al., but if they would read Papayannis et al., (Atmos Env. or Atmos Res., 2012?) they would seethat it was rather complicated to identify the volcanic aerosols in the mixtures of urban haze, marine particles, Saharan dust etc over the Eastern Mediterranean„ far awayfrom Iceland, so that information of these articles (even Mona et al., 2012) should behandled with caution.

*We are aware of this issue.*

**Section 2 Methods**

Section 2.1 can be shortened drastically, the lidar equation is given in so many textbooks, so many details are not needed. Focus on the parameters needed for Eq.2.

*We think that a manuscript which introduces a backscatter lidar forward operator should contain the lidar equation as the readers are not only experienced lidar researchers.*

P4, l30: Lidars are never elastic (hopefully. . .), so please write: elastic backscatterlidar Also sections 2.2.1 and 2.2.2 contain just text book knowledge. Please keep thediscussion as short as possible, use references.

➔ *We changed this wording issue.*

P7,l7: Buchholtz's formula holds for 200 to 1000nm, but you use 1064nm (please mention the used wavelength much earlier in the paper, for example already in this section!)? Did you extrapolate? Or ignore it? Please state the wavelength immediately,and give all the Rayleigh parameters for this wavelength, including the lidar ratio. . .

*Yes, we extrapolated as it is near 1000 nm and the values are nearby constant for these values.*

➔ *We have noted this in the manuscript, see P6, l25.*

Section 2.2.3, even here, please provide the wavelength of the ceilometer, and then discuss all the different size scenarios for this wavelength, give examples for which the Mie theory may holds, for what size range (Aitken mode, accumulation mode, etc. . .), the same for DDA, T-matrix etc. Which of the natural particles have almost spheroidal shape, which look like cyclinders? Make the full discussion more realistic, more lively.

*There is an infinite variety of particle shapes, particle types, matter compositions, lidar applications, etc. We believe that going into the details of any related topic would reduce the readability of the paper. All the proposed aspects are complicated fields of research – it would be better to initiate dedicated studies to the related topics than just scraping the surface.*

*Regarding the wavelength, however, the exact value of ACL systems in the case was already written in Section 2.2.2.*

P7,l20: So you definitely use Mie scattering codes for your backscatter lidar forwardoperator! This is ok, and the best first step you can choose! BUT!!! … as a consequence, please provide us with an application (demonstration case) for which Mie computation really holds (fine-mode aerosol, urban haze and smoke) . This is a mandatory condition for the final acceptance of this paper from my side!!!

*First, we do not agree that "Mie computation really holds" for the given particle classes: These particles are not 100% spherical and assuming so without performing an uncertainty analysis would simply degrade the validity of such a case study. Second, we have demonstrated that the forward operator is compatible with any scattering calculation method which outputs both the extinction and the backscatter cross section. This can be either Mie scattering code for spherical particles, t-matrix scattering code for rotationally-symmetric non-spherical particles, DDA for arbitrarily-shaped particles or whatever will come in the future. So improvements related to the aerosols' scattering properties are only a question of scattering calculation – but not a deficit of the forward operator! Third, we are already using t-matrix codes for the error estimation. The reviewer seems to have missed that fact – even if we have documented the use of t-matrix code several times and also revealingly during the complete section 4.3.*

Again, in conclsuion: Sections 2.2.3 and 2.2.4 are rather lengthy, not necessary, textbook knowledge, sections should be as short as possible.

*Sections 2.2.3 and 2.2.4 contain the explanation on how to set up the forward operator for multiple scatterer types which is the basis for adaption.*

➔ *We did not remove these sections from our manuscript.*

**Section 3 Case Study**

I could simply leave out the rest because I am not willing to accept a revised versionwith such a complicated case. But again, many figures could be left out because theyare already presented in the literature, e.g., that the extinction coefficient is robust withrespect to shape effects (see for example Dubovik JGR, 2006, but this is shown in many other papers too). So, one could easily skip Figs.8, 10, 12, 14.

*There is no reason why we should omit one of the two most important output quantities of the forward operator. We think that a reader should be allowed to compare the resulting extinction cross section and backscatter cross section curves in detail as it is also the basis for the calculation of the lidar ratio plots.*

*But we agree that this valuable information could be condensed.*

➔ *To reduce the number of figures and improve the comparability of the numerous scatterer types, we combined Figures 8,10, and 14 as well as Figs. 9,11, and 12*

Here is the Dubovik reference:

Dubovik, O., Sinyuk, A., Lapyonok, T., Holben, B., Mishchenko, M., Yang, P., Eck, T.,Volten, H., Munoz, O., Veihelmann, B., van der Zande, W. J., Leon, J. F., Sorokin,M., and Slutsker, I.: Application of spheroid models to account for aerosol particlenon-sphericity in remote sensing of desert dust, J. Geophys. Res., 111, D11208,doi:10.1029/2005JD006619, 2006.

The paper has 19 Figures, and 50% are just required to explain the complexity ofthis crucial volcanic ash case and the impact of size, shape and composition of volcanic particles on the related optical properties. Alone this fact corroborates that weneed a better demonstration case. The main goal of the paper is: presentation of thebackscatter lidar forward application plus demonstration case. And without presentation of a convincing demonstration case, the paper must be rejected because the mainmessage and conclusion then becomes completely unclear.

*This statement does not make sense. There exists not a "simple case", as demonstrated above, but the uncertainties are just transferred from aerosol microphysics to the sources. Another case with less uncertainty with respect to the refractive index, size, and shape uncertainties, the reader would get the impression that there are in general no uncertainties related to the forward operator method. And this is not true. The complexity and uncertainties have to be part of the analysis, which does not substantiate at all a rejection of this manuscript.*

➔ *This is equivalent to the consideration concerning P7,l20 above*

P10, l23: One of the best lidar-ratio papers in this field is not mentioned in the article: Groß, Silke und Freudenthaler, Volker und Wiegner, Matthias und Gasteiger,Josef und Geiß, Alexander und Schnell, Franziska (2012) Dual-wavelength linear depolarization ratio of volcanic aerosols: Lidar measurements of the Eyjafjallajökullplume over Maisach, Germany. Atmospheric Environment, 48, Seiten 85-96. DOI:10.1016/j.atmosenv.2011.06.017.

One needs the depolarization ratio to see in which of the layers pure or almost purevolcanic dust was observed. In the lowest layers a mixture of local pollution, sulfateparticle from the volcanic emission of SO2 and volcanic ash prevailed. Nevertheless,the lidar ratio for the volcanic ash was always close to 50 sr, and in the mixtures evenlarger, probably because of the freshly formed quite small sulfate particles. None of the publications on the Eyjafjalla eruptions, showed lidar ratios below 40sr in the volcanic layers. And this must be the guide for all your modeling studies which to my opinionare at all rather speculative. Fact is: The reality did not provide volcanic lidar ratios of10-20sr.

*This reference partially validates the results of our scattering calculations and sensitivity studies*

➔ *We added this reference to the manuscript and implemented its content to the discussion (P20, I 25-31)*

Add to the references. Pappalardo, G., Mona, L., D'Amico, G., Wandinger, U., Adam,M., Amodeo, A., Ansmann, A., Apituley, A., AladosArboledas, L., Balis, D., Boselli, A.,Bravo-Aranda, J. A., Chaikovsky, A., Comeron, A., Cuesta, J., DeTomasi, F., Freudenthaler, V., Gausa, M., Giannakaki, E., Giehl, H., Giunta, A., Grigorov, I., Groß, S.,Haeffelin, M., Hiebsch, A., Iarlori, M., Lange, D., Linné, H., Madonna, F., Mattis, I.,Mamouri, R.-E., McAuliffe, M. A. P., Mitev, V., Molero, F., Navas-Guzman, F., Nicolae,D., Papayannis, A., Perrone, M. R., Pietras, C., Pietruczuk, A., Pisani, G., Preißler, J.,Pujadas, M., Rizi, V., Ruth, A. A., Schmidt, J., Schnell, F., Seifert, P., Serikov, I., Sicard,M., Simeonov, V., Spinelli, N., Stebel, K., Tesche, M., Trickl, T., Wang, X., Wagner, F.,Wiegner, M., and Wilson, K. M.: Four-dimensional distribution of the 2010 Eyjafjallajökull volcanic cloud over Europe observed by EARLINET, Atmos. Chem. Phys., 13,4429-4450, doi:10.5194/acp-13-4429-2013, 2013.

See also. . ..

Wiegner, M., Gasteiger, J., Groß, S., Schnell, F., Freudenthaler, V., Forkel, R., 2011.Characterization of the Eyjafjallajkullash-plume:potential of lidar remote sensing.Physics and Chemistry of the Earth. doi:10.1016/j.pce.2011.01.006.

*If our manuscript was about validating model predictions using Raman lidar measurements, we would add the suggested references. However, the subject of our manuscript is the validation of the COSMO-ART simulation during the Eyafjallajökull using data provided by the ACL network. We agree that the comparison of forward modeled simulation results and EARLINET measurements could be very interesting and consider subjecting this within a subsequent manuscript.*

**Section 3.2**

I miss a Table with technical details of the ceilometer, what, for example, is the pulserep rate? And what does it mean... Assuming that the pulse energy of 8 micro Joulegiven by Flentje is true. Do you know the technical details or not? Are you sure thatthe overlap is completed in 1500m height? Do you know it? Yes or no? A calibrationmeasurement of the ceilometer was not provided... We take one from CALIPSO for1064 nm. All this is really not easy to accept and tolerate by a reviewer trying to teachhis student day by day to do always careful calibrations and high quality observations.

*Our statement concerning the pulse energy is indeed too vague as the technical details are well known and given by for example Flentje et al. (2010, see [http://www.atmos-meas-tech-discuss.net/amt-2010-83/](http://www.atmos-meas-tech-discuss.net/amt-2010-83/)). Information regarding the overlap function can be found in the paper of Heese et al. (2010, see [www.atmos-meas-tech.net/3/1763/2010/](www.atmos-meas-tech.net/3/1763/2010/)).*

*The ACL calibration issue is indeed unfortunate but since the Eyjafjallajökull eruption in 2010, lots of effort was put into calibrated measurements of the ACL systems in the ceilometers network.*

➔ *We have changed the sentence "Assuming that the pulse energy of 8 micro Joulegiven by Flentje is true"*
➔ *We have added an explanation of the ACL calibration in Section 3.2 of our manuscript.*

**Section 3.3.**

P12, l9: COSMO-ART uses six size classes, and class 1 includes all particles with diameters up to 2 micrometers, so all Aitken and accumulation mode particles and the rest of all fine mode particles (up to a diameter of 1 micrometer), and then also a large fraction of the coarse mode (particles with diameters from 1 to 2 micrometers) are altogether in just one single class. Is that justified from the optics point of view? I donot believe. . .

*Indeed, it would make sense to increase the number of size classes for the smallest particle sizes for future model set-ups. From the forward operators' point of view, however, it is restricted to the information it receives from the model.*

➔ *We have added a recommendation for other model set-ups to the results section (P21, l11-12).*

P12, l22: molecular number density of standard air was calculated. What do youmean? Do you use, for simplicity, standard atmospheric conditions? Is that justified? Schumann paper is referenced as ACPD version, this in not acceptable, you need totake the ACP version. . . What size distributions did Schuman et al measure. . .? Should be used as a guide in all the modeling approaches.

*Right! The molecule number density is calculated from the temperature and pressure profiles simulated by COSMO-ART and not a standard atmosphere. We use the equations given by Buchholtz (1995) to calculate the scattering cross sections for standard air.*

*Thank you for the hint on referencing the ACPD paper of Schumann et al. (2011). The location of the final revised paper is www.atmos-chem-phys.net/11/2245/2011/.*

➔ *Changed the description of the molecule number density at P6, l23-l27,*
➔ *Changed the reference list entry of Schumann et al. (2011).*

Section 4 is really not needed if you take a simple demonstration case. . . I will not givecomments, except one: You cannot provide any uncertainty estimation (e.g. Fig 15) aslong as you do not know what the truth is. We know already that neither the spherical nor the spheroidal particles shape models works in the case of irregularly shape dust particles (Gasteiger, 2011, Kemppinene, 2015).

*As the reviewer noted here and as we have written several times in this point by point response letter, there is no simple case which is has no uncertainties concerning the scattering properties and also there is no ideal particle shape which could be used as universal reference scatterer.*

**Section 5 Results**

The Schumann et al size distributions may show approximately the size distribution characteristics. Your size distribution cases 1 and 2 are just speculations. Case 1 (this COSMO modeled size distribution) appears to some extend realistic. Table 3 containsthe modeled lidar ratio, and for the main classes 1,2 and 3 (case 1) the lidar ratios are13, 4.6 and 10 sr, and on average probably close to 10sr and therefore far farawayfrom the real-world values of 50-60sr.

➔ *This concern has become obsolete due to corrected results of the lidar ratio calculations; see the respective discussion of the lidar ratio above.*

As mentioned be careful when using Mediterranean Eyjafjalla dust papers, the dustmaybe already widely mixed with other aerosols. And the paper of Mortier provides some column integrated lidar ratios and not values for well defined isolated lofted dustlayers.

*We thank the reviewer for hinting on these risks.*

All in all, section 5.2 is so confusing and speculative, and simply ignores literature findings. There is also attempt to check the literature for observational hints that wouldcorroborate all there speculative arguments. . . This part of the paper is clearly inacceptable.

*We do not agree with this statement, which is also not substantiated. For instance, it is not explained what is called "confusing and speculative". We think that the only statement, which is confusing, it this statement of the reviewer. If something is called "inacceptable", this needs to be clearly identified in a scientific review. Unfortunately, this information is entirely missing.*

*As this is a new approach, there are only few literature resources available which we already mentioned in the text. As soon as other groups are testing forward operators based on the approach shown in our manuscript, this situation may change.*

**Section 5.3 Qualitative Comparison**

...provides no concluding remarks and any take-home messages (zero!) Again, andvery clear: The shown demonstration case is a disaster for entire article and must beremoved

*We are not happy with these sloppy and polemic comments without any information content. We expect a factual choice of words in a scientific review of ACP. Furthermore, we disagree with the statement of the reviewer concerning the choice of this case.*

*A qualitative comparison is (until today) the state-of-the art method for validating aerosol dispersion models using the available ACL data. As the reviewer points out "very clear", validating the model by such qualitative comparison allows for only few conclusions concerning the validity of the prediction. This gets even worse if the qualitative comparison is performed without applying the backscatter lidar forward operator, for example by comparing the ACL backscatter signal intensity and the model predicted ash number concentration of multiple size classes.*

➔ *As the relevance of the backscatter forward operator was already point out in the abstract and the benefits are rather clear, we have not performed further chances to the manuscript concerning this remark.*

**Section 5.4 Quantitative Comparison**

Because of lack of any harmony between the model results and ceilometer observtions (a factor of 60 difference!), the authors start to play around with the output (attenuated backscatter values) of the forward operator and reduced the output by a factor 10, 20, 30 etc.

*We are not "playing around". This is another wording which disqualifies this reviewer judging this scientific approach in our manuscript. Again, we would appreciate very much an objective approach and wording in a scientific review.*

*"Lack of harmony" is also not an expression with is appropriate to explain our approach. It is just necessary to scale the results because of unknown gain and/or pulse energy of the ceilometers. Please consider that this is the first time the COSMO-ART predicted volcanic ash dispersion was*

*compared to ACL measurements directly, i.e.given in the same physical quantity. There is a lot of work required to get optimum results as we have shown within our manuscript. We regard our manuscript as an important step towards a new generation of lidar forward operators which could become one of the most important techniques for validating atmospheric chemistry models.*

➔ *These considerations are already contained within our manuscript*

Why do you not check other observations (EARLINET lidar observations,even the CALIPSO overflight is available) regarding the truth! So discussion remainedopen. . . at the end.

*We used CALIPSOs CALIOP instrument to calibrate the ACL measurements. The reviewer seems to have missed this information.*

Figure 4 shows particles in diameters, Figs 8,9, etc show particles in radius, Figure 16shows particle size. . ..., please harmonize!

*Figure 4 is a visualization of the model set-up which is defined by the particle diameter while the other figures are related to scattering calculations given in radius. Due to this fact we are convinced that it is better to leave these figures as they are. Due to redundant information of Fig. 16 and Tables 4 and 5, we removed Fig. 16 however.*

➔ *We have removed Fig. 16 as it contains redundant information.*

Figure 17 shows COSMO ART results ..with shape sensitive lidar ratios at all around10sr. All colors are just grey. Nevertheless this figure corroborates best that thisdemonstration case is a disaster. Measured 'real world' Eyjafjalla lidar ratios are afactor of 4 higher. The discrepancy is to my opinion just caused by the shape of thevolcanic particles.

*Another inappropriate wording of the reviewer is the characterization of an intercomparison as a "disaster". Considering the original definition and the displaced application of this word, disqualifies again the quality of this review.*

*This concern has become obsolete due to corrected results; see the discussion of the lidar ratio above. The shown plot was also calculated based on the averaged values of the effective backscatter and extinction cross sections and thus not correct. We therefore removed this figure from the panel.*

➔ *We have removed the plot of the lidar ratio from Fig. 17 as it was calculated based on the effective optical cross sections instead of based on the lidar ratio spectrum.*

Figures 18 and 19 present almost the opposite of what I would like to see, and whatwould be possible to show if a simple European pollution day would be selected. It ishard to believe that the ceilometer output (top panels) have something to do with theCOSMO ART output (bottom panel).

*There is no "simple European pollution day", as we pointed out above. The case study selection and the comparability of forward modeled and measured volcanic ash structures have already been discussed in this point-by-point response.*

---

## Author Comment (AC2) · 12 Nov 2016

**A Backscatter Lidar Forward Operator for Particle-Representing Atmospheric Chemistry Models**

Armin Geisinger1, Andreas Behrendt1, Volker Wulfmeyer1, Jens Strohbach1, Jochen Förstner2, Roland Potthast2, and Ina Mattis3

1Institute of Physics and Meteorology, University of Hohenheim, Germany 2Headquarter of the German Weather Service, Offenbach, Germany 3Observatory Hohenpeissenberg of the German Weather Service, Hohenpeissenberg, Germany

Correspondence to: a.geisinger@uni-hohenheim.de

**Abstract.**

State-of-the-art atmospheric chemistry models are capable of simulating the transport and evolution of particles and trace gases but there is a lack of reliable methods for model validation and data assimilation. Networks of automated ceilometer lidar systems (ACLs) provide a 3D dataset of atmospheric backscatter profiles. However, as the aerosol number concentration

- 5 cannot be obtained from the ACL data alone, a backscatter-lidar forward model is required which allows for a qualitative and quantitative model validation based on ACL data. We developed a new backscatter-lidar forward operator which is based on the distinct calculation of the aerosols' backscatter and extinction properties. The forward operator was adapted to the COSMO-ART ash dispersion simulation of the Eyjafjallajökull eruption in 2010 as - due to its impact on the public sector - such events have become a major motivation for aerosol dispersion modeling. While the particle number concentration is
- 10 provided as model output variable, the scattering properties of each individual particle type has to be determined by extensive scattering calculations. As these scattering calculations require assumptions concerning the particle refractive index and shape, sensitivity studies were performed to estimate the uncertainties related to the assumed particle properties. Therefore, scattering calculations for several types of non-spherical particles required the usage of t-matrix routines. Due to the distinct calculation of the backscatter and extinction properties of the models' volcanic ash size classes, the sensitivity studies could be resolved
- 15 to each size class individually which is not the case for forward models based on a fixed lidar ratio. Finally, the forward modeled lidar profiles were compared to ACL measurements both qualitatively and quantitatively. Therefore, the attenuated backscatter coefficient was chosen as common physical quantity which for calibrated ACL measurements only relies on the ACL laser wavelength. As the ACL measurements were not calibrated automatically, their calibration had to be performed using CALIPSOs/CALIOP measurements. After calibration and comparing model and measurement at the same time and geographic
- 20 location, a slight overestimation of the model predicted volcanic ash number density was observed. By manually reducing the model predicted ash number density, the effect of simple data assimilation methods could be demonstrated. Unfortunately, the uncertainties related to both measurement and forward operator currently limit more precise model validation attempts using the ACL measurements. The forward operator and the detailed analysis of the related uncertainties, however, allowed us to identify the key issues which are mandatory for performing quantitative model validation using a backscatter lidar forward
- 25 operator and ACL measurements. The major issues are (in a nutshell): ACL data quality, the representation of scatterers within

the forward operator, the aerosol representation within the model and motivations for improved measurement networks. We consider this forward operator development as a crucial step for future assimilation of the huge information content delivered by lidar backscatter measurement in chemical weather forecast models. The introduced forward operator offers the flexibility to be adapted and refined and can thus be used on a multitude of model systems and measurement set-ups.

**5 1 Introduction**

In Spring 2010, the volcano Icelandic volcano Eyjafjallajökull erupted several times. The emitted ash was found to be harmful for aircraft and due to uncertain information about spatial distribution and concentration of volcanic ash, the European air space was closed for several days (Sandrini et al., 2014). The high economic costs and impact on public transport lead to efforts of DWD (Deutscher Wetterdienst) to improve at monitoring and predicting aerosol cloud movements in the atmosphere.

10 Therefore, DWD decided to start a dedicated project on backscatter lidar forward operators for validating aerosol dispersion models using available remote sensing measurement data.

[revised manuscript text omitted]
_{\mathrm{mol},\lambda}(z) = N_{\mathrm{mol},\lambda}(z) \,\sigma_{\mathrm{sca,mol},\lambda},\tag{7}$$

15

25

$$\beta_{\mathrm{mol},\lambda}(z) = N_{\mathrm{mol},\lambda}(z) \left(\frac{d\sigma_{\mathrm{sca,mol},\lambda}}{d\Omega}\right)_{\pi}.$$
(8)

The molecule number density  $N_{mol}(z)$  is related to the ideal gas law

$$N_{\rm mol}(z) = \frac{p(z)}{k T(z)},\tag{9}$$

where p is the model prediction of the atmospheric pressure given in Pascal (Pa), T is temperature given in Kelvin (K), and k 20 is the Boltzmann constant which has a value of about  $1.381 \times 10^{-23} \text{ J K}^{-1}$ .

To calculate the scattering cross-section  $\sigma_{\text{sca,mol},\lambda}$  and the scattering phase function  $\phi_{a,\lambda}(\theta)$  of air, empirical equations are available. We used the formulas and look-up tables given by (Buchholtz, 1995). As the empirical equations are only given for wavelengths up to 1000 nm, we simply extrapolated the values to the ACL wavelength in the case study (1064 nm). This method 
[revised manuscript text omitted]

(27)

The emitted photon number per shot  $N_{tr,\lambda}$  was calculated using:

$$N_{\rm tr,\lambda} = \frac{E_{\rm pulse,\lambda}}{E_{p,\lambda}},\tag{28}$$

5 where  $E_{\text{pulse},\lambda}$  is the laser pulse energy.  $E_{p,\lambda}$  is the photon energy, calculated according to:

$$E_{p,\lambda} = \frac{h c}{\lambda},\tag{29}$$

with h as Planck's constant having a value of  $6.62607 \times 10^{-34}$  Js. The pulse energy of the diode-pumped laser is  $8 \mu$ J (Flentje et al., 2010c) resulting in an emitted photon number per pulse of about  $4.28 \times 10^{13}$ . The diameter of the receiving telescope is 100 mm (Flentje et al., 2010c) which results in  $A_{\text{tel}} = 78.54 \text{ cm}^2$ . The vertical resolution  $\Delta z$  is 15 m over the complete profile. The overlap function O(z) was set to 1 which implies that ranges below about 1500 m cannot be used reliably for comparisons with the forward operator.

Unfortunately, the instruments provided no calibrated measurement data at that time. As the true system efficiency  $\eta_{\lambda}$  and the calibration coefficients are not known, we use the symbol  $\eta^*$  as linear calibration factor. From a comparison with the calibrated attenuated backscatter measurements of CALIOP at  $\lambda = 1064 \text{ nm}$  (Fig. 3), we determined a calibration factor of  $\eta^* = 0.003$ :

- First, we used the CALIOP value of the 1064 nm calibrated attenuated backscatter coefficient at 50.15° lat / 4.81° lon in a height of 2 km ASL (about  $5 \times 10^{-6} \text{ m}^{-1} \text{ sr}^{-1}$ ) to estimate the maximum attenuated backscatter coefficient inside the volcanic ash cloud. The second calibration criterion was the forward modeled attenuated backscatter coefficient outside the volcanic ash layer in 3 km ASL which is dominated by molecule scattering and has a value of  $1 \times 10^{-7} \text{ m}^{-1} \text{ sr}^{-1}$ , see Fig. 15. This pragmatic approach is of limited precision and only required for uncalibrated ACL measurements. As most ACL networks
- 20 have been extended by automatic calibration methods after the Eyjafjallajökoll eruption, such a calibration will not be required in future forward operator studies.

[revised manuscript text omitted]

- 15 We modified the double-precision version of the T-matrix procedure to perform scattering calculations of multiple particle sizes automatically. In addition to that, the procedure was extended by the calculation of the backscatter cross section  $\sigma_{\rm bsc}$ according to Mishchenko et al. (2002), Eq. (9.10). Then, as a test, both mie\_single and our modified T-matrix code were set up to calculate the scattering properties of the same spheres and for the same wavelength. The results of both procedures were indeed identical.
- A list of T-matrix options we used for the particle shape sensitivity study is shown in Table 2. The most important particle properties are defined by the variables NP and EPS. NP defines the particle type and has a value of -1 for spheres as well as for ellipsoids. A NP value of -2 is used for cylinders. The variable EPS is an expression for the objects' diameter to length ratio. Consequently, an ellipsoid with EPS=1 is a sphere, prolate objects have an EPS<1 and oblate objects have an EPS>1.
- In Figs 8, 9, and 10, the optical cross sections and the lidar ratio of spheres and several aspherical particles are plotted against the equal-volume radius. The aspherical scatterers are 6 ellipsoids with a diameter-to-length-ratio of 0.50, 0.67, 0.75, 1.25, 1.50 and 2.00 as well as 5 types of cylindric particles with a diameter-to-length-ratio of 0.50, 0.80, 1.00, 1.25 and 2.00. As shown in these plots, the results for a highly asymmetric ellipsoid (EPS: 0.50) is only available up to an equal-volume radius of 3.75 µm. For future research activities in this topic, the quadruple precision version of the t-matrix code should be preferably used.

We found no significant differences of the extinction cross section of spheres and these ellipsoids. By trend, cylindric shaped particles have a higher extinction cross section compared to ellipsoids and the spherical shape has the lowest extinction cross section values over the whole spectrum. Up to a volume-equivalent radius of  $0.7 \,\mu\text{m}$ , the shape effect is not noticeable.

- Regarding the backscatter cross section, however, we find significant differences between the backscatter cross section of 5 spheres and particles with other shapes. Obviously, spheres are affected by interference effects which lead to both fluctuating and oscillating values of the backscatter cross section while the other shapes only show weakly fluctuating values of the backscatter cross section. As observed for the extinction cross section, the shape effect becomes pronounced beginning at an equal-volume radius greater than  $0.7 \,\mu\text{m}$ . Spherical scatterers have a higher value of the backscatter cross section compared to ellipsoids except for one type of ellipsoid (EPS = 1.25). For cylinders, such interference effects are not observable so the
- 10 backscatter cross section of the considered cylinders increases monotonically with the equal-volume radius. As a result, the backscatter cross section of spheres is lower than of cylindric particles with the same size if their equal-volume radius is greater than  $3.75 \,\mu\text{m}$  (for the given wavelength of  $\lambda = 1064 \,\text{nm}$ ).

The particle shape effect on the lidar ratio is only weakly pronounced for small particle sizes (less than  $0.75 \,\mu$ m), too. For larger particles, the lidar ratio of spheres is generally lower than of the other considered shapes which is in agreement to the

- 15 higher backscatter cross section observed before. For the fourth size class (equal-volume radii around 5 µm), the previously observed interference effects of the spheres' backscatter cross section leads to extreme values of the lidar ratio (exceeding a lidar ratio of 200 sr). By trend, however, the lidar ratio of spheres in this size class is not that different than the lidar ratio of other considered shapes. But for the other size classes, the lidar ratio of spheres is the lowest of all observations except for cylinders which have a lidar ratio value at the same order of magnitude as spheres.
- A summary of the particle shape sensitivity study is shown in Figs 11, 12, and 13, giving the relative differences of the effective optical cross sections and average lidar ratios for different particle shapes. The definition of the relative differences follows Eq. (30) and Eq. (31). Again, the reference is a spherical particle. Positive and negative relative differences indicate that the calculated values for spheres are underestimations and overestimations compared to respective non-spherical particles.
- The effective extinction cross section of spheres is lower than the effective extinction cross section of other analyzed asymmetric particles. Regarding the effective backscatter cross section, however, we find relative differences of up to 300% and -80%. While the small aspherical particles have a lower effective backscatter cross section compared to spheres, the effective values of the fourth size class are higher for almost all considered aspherical particles. From this analysis, it can be concluded that due to the assumption of sphericity, the backscatter cross section of size classes 1, 2 and 3 are overestimated by about factor 1.5 to 5 while the backscatter cross section of the fourth size class is underestimated by factor 2. This allows for quantifying
- 30

The relative difference of the average lidar ratio shows that assuming spheres for the scattering calculations results in much lower lidar ratio values of size classes 2 and 3 but in higher lidar ratio values than would be observed for a mixture of particle shapes. For the fourth size class, the opposite has been observed: assuming spheres leads to an higher values of the lidar ratio of

the over- and underestimation of the results for each size class individually which is not possible for forward operators where a fixed lidar ratio is assumed.

large particles than for non spherical particles of the same size. The lidar ratio of the first class, however, is nearby independent on the particle shape.

Of course, the total effect of different particle shapes for a real aerosol mixture has to be estimated with more extensive scattering calculations. For the current state of the forward operator research, the spherical shape has to be used as reference

- 5 even if the real volcanic ash particles are known to be fractal and complex shaped. One of the reasons for doing so is the fact that there is currently no appropriate shape representation scheme for volcanic ash available. For example from the findings of Rocha-Lima et al. (2014), the average aspect ratio of volcanic ash is known but not a representative particle shape. As we have shown, the backscatter cross section of cylinders also differs tremendously from ellipsoids and from spheres etc. so there is currently a lot of indication that using a given aspherical shape as reference would yield in even higher errors than assuming
- 10 a spherical shape.

The above results, however, give us valuable insight into the uncertainties of using Mie calculations for non-spherical particles. These findings are important for interpreting the results of the forward operator.

**5 Results**

**5.1 Scattering Properties of Volcanic Ash Used Within the Forward Operator**

15 A list of the effective extinction cross section and the effective backscatter cross section we determined for atmospheric gas molecules and for the six volcanic ash size classes is shown in Table 3.

**5.2 Output Variables of the Forward Operator**

Using the forward operator allows for plotting each variable of the lidar simulation for analytic purposes (see Fig. 14). These plots of forward-operator output variables are representing the major characteristics of the variables: strong extinction and strong backscattering are usually related. Time and height intervals at which only molecules exist, lead to low values of the extinction coefficient and backscatter coefficient. Due to the decrease of the atmospheric gas number density with height, both extinction and backscatter coefficient decrease with height in an aerosol-free atmosphere. The two-way transmission decreases with height (see Eq. (24)).

In addition, the volcanic ash contribution to the total signal and the total mass density was analyzed for each size class of COSMO-ART within two case studies. The cases were selected to cover extreme situations: Case 1 is the model output from a coordinate inside the volcanic ash layer (Table 4). Case 2 is for a coordinate where the majority of particles are due to size class 4 and 6 (see Table 5).

Regarding case 1, the total backscatter coefficient is dominated by ash size classes 1, 2 and 3 while the signal contribution of classes 4 to 6 is less than 5% in total. The mass contribution is dominated by classes 3 and 4 while classes 2, 5 and 6 are

30 contributing by 10% each to the total mass density and class 1 is very low for the total mass density. Regarding case 2, the total backscatter coefficient depends by about 68% from class 4 and by 30% from class 6. The mass contribution in case 2 is

also dominated by the classes 4 and 6 but, in contrast to the backscatter coefficient, class 6 has a higher contribution to the total mass density than class 4.

General conclusions from this analysis about the relationship between backscattering and mass depending on particle size and wavelength require further investigation. For our application of the forward operator in this study, however, we can conclude

- 5 the following: First, the total signal inside the volcanic ash layer (case 1) is predominately dependent on classes 1, 2 and 3 whose backscatter cross sections are also overestimated by the forward operator due to the assumption of sphericity (see Fig. 12). The real values of the attenuated backscatter coefficient may be by factor 2-3 higher. Second, the larger particles of classes 4, 5 and 6 carry a large portion of the mass but contribute only weakly to the total signal. This may be an important information for the selection of future ACL networks as even the systems operating at 1064 nm have a reduced sensitivity for particles
- 10 within these size classes.

**5.3 Qualitative Comparison**

A qualitative comparison allows for the identification of common and different structures between the measured and simulated lidar profiles. Different ash layer structures can hint, e.g., to errors in the model dynamics, in the source description, or in the sedimentation parametrization. If ash structures are found only in the measured profiles, either the model prediction is wrong

15 or it misses an important aerosol type which is not present in the model. If structures are visible in the forward modeled profiles but missing in the measured profiles, either the ACL signal is too weak because of high extinction in lower heights or the model performed a wrong ash prediction. But it is also possible that the model overestimates the ash concentration so the structures are below the detection limit of the ACL measurements.

[revised manuscript text omitted]

Due to uncertain refractive indices of the volcanic ash and due to the fact that volcanic ash particles show complex shapes, sensitivity studies have been performed to analyze the impact of different particle shapes on the effective extinction and

- 20 backscatter cross section as well the lidar ratio. While the extinction cross section was only weakly sensitive to variable refractive indices and particle shapes, the backscatter cross section was strongly sensitive to both. However, we found that the sensitivities reduce significantly, when averages over size classes are made. We expect that this very interesting result described in this manuscript for the first time will be very helpful for comparisons of modeled with measured backscatter lidar data.
- Using these effective optical cross sections reduced the sensitivity of optical cross sections regarding the refractive index as well as the particle shape. But even after averaging, the relative uncertainty of the effective backscatter cross section exceeds 280% for uncertain refractive indices. This study also indicates the dependency of the forward operator on precise information about the particle's refractive index. Within a particle shape sensitivity study, we were able to resolve the relative uncertainty of each individual size class for the effective backscatter cross section. Assuming that volcanic ash consists of a mixture of
- 30

particle shapes, we analyzed the relative differences between the reference particle shape and 11 particle shapes (6 types of ellipsoids and 5 types of cylinders).

The forward operator matches the lidar ratio values we find in literature (40 sr to values higher than 100 sr for volcanic ash (Kokkalis et al., 2013; Ansmann et al., 2010; Mortier et al., 2013)). On average, the lidar ratio is 61.17 sr which fits well to the literature findings. Comparing the lidar ratio values of the first two size classes with the lidar ratio values reported by Gasteiger

et al. (2011b), a lidar ratio of less than 20 sr seems to be plausible for these particle size to wavelength ratios. The authors found even for irregularly shaped objects a lidar ratio between 5 sr and 20 sr at size parameters between 5 and 15 (equivalent particle diameter at  $\lambda = 1064$  nm would be 1.6 µm and 4.8 µm, respectively). We therefore assume that the calculation results of the backscatter lidar forward operator are valid and allow for both qualitative and quantitative comparison.

- 5 A comparison between ACL measurement and the model predictions used as input for the developed forward operator was shown. Similar structures were observed but some features were referenced to different time and height locations. From our analysis at the ACL station Deuselbach, some ash layer features were predicted quite precisely by the model, for example the time of arrival of the ash plume at about 06:00 UTC with a vertical shift of about 1.5 km. Some other features, such as the intersection with the planetary boundary layer at 17 April 2010, 03:00 UTC, was simulated about 6 hours too early to 16 April
- 10 2010, 18:00 UTC. Fine structures of the ash layer were only observable in the simulation but not in the ACL measurements due to noise.

Due to unknown calibration coefficients of the ACL system, a calibration constant  $\eta^*$  was estimated by comparing the ACL data with calibrated measurements at the same wavelength. Within quantitative comparisons between ACL measurements and the forward operator output, we found that the molecule signal of ACL and forward operator output were of the same

15 order of magnitude which argues that the selected calibration factor was reasonable. Meanwhile, the ACL manufacturers have understood the importance of calibrated backscatter data and implemented technical solutions so that a similar effort as described in this study with data for the year 2010 became obsolete.

A comparison the volcanic ash signal led to the conclusion that the model predicted ash concentration was to be too high as the forward modeled attenuated backscatter coefficient within ash layers was 60 times higher and after attenuation 10 times

- 20 lower than observed by the ACL. If the model-predicted ash concentration is manually reduced by a factor of 20, the forward modeled COSMO-ART predictions and ACL measurements became quantitatively similar. Such a reduction could be part of a simple particle data assimilation system helping to calibrate particle dispersion simulations before in-situ measurements are available. It would of course be beneficial, if even better information on the refractive index and effective particle shape and aspect ratio of volcanic ash particles becomes available in the future.
- Furthermore, we analyzed the contribution of each class to the total backscatter coefficient and to the total mass density for two sample cases. Regarding case 1 inside the volcanic ash layer, the classes 1, 2 and 3 were mostly responsible (94.8 %) for the calculated attenuated backscatter coefficient. As these classes contribute most to the forward modeled signal, the value of the lidar ratio would also be expected to be dominated by their contribution, namely a value between 5.23 sr and 58.83 sr (see Table

3). Raman lidar measurements of the Eyjafjallajökull ash resulted lidar ratio values greater than about  $40 \,\mathrm{sr}$  at wavelengths of

30 355 nmand 532 nm (Groß et al., 2012) which is within this range. Thus, the calculated values of both extinction and backscatter cross section as well as the lidar ratio seem to be plausible.

Assuming that the ACL calibration is valid, the model-predicted ash concentration was about 10 times higher than observable. There are, however, some error sources remaining which are: First, there are only molecules and the six volcanic ash classes represented while background aerosol is missing completely. Second, the ACL calibration is of limited precision.

35 Third, the contribution to the attenuated backscatter coefficient of ash size classes 4, 5 and 6 is relatively low even though these

classes carry a large proportion of the mass. This relationship could rely on the ACL's wavelength which probably limits its sensitivity to particles larger than about  $10 \,\mu\text{m}$  in diameter. Such results strengthen the importance of a joint use of observations and model output in combination with data assimilation in order to get the best state of the atmosphere with respect to aerosol distributions and properties.

- 5 Conclusively, we recommend further investigation in scattering calculations of non-spherical particles to get more realistic optical cross sections for the forward operator. A decrease of uncertainties related to the forward operator can be achieved by refractive index measurements at the exact ACL wavelength. Refractive index measurements are a basic aspect of the forward operator as the optical cross sections can only be calculated if the aerosols' refractive index is known precisely. The model and consequently the forward operator has to represent more aerosol types, especially background aerosols, mineral dust,
- 10 sea salt and soot as missing extinction near ground cause the forward operator to overestimate the signal from layers behind. But also qualitatively, more scatterer size classes are required to also represent the fine fraction and very large particles in the atmosphere. One approach for a better representation of the natural size-spectrum of aerosols is the use of continuous numbersize distributions which are aggregated from multiple distribution functions ("modal" approach). On the one hand, this already includes the size-averaging which is necessary for monodisperse size distributions. But on the other hand, the model delivers
- 15 exact information about the outer margins, i.e. the number-density of the fine and the extreme coarse fraction which is currently not reproduced by model and forward operator in the Eyjafjallajökull case study.

[revised manuscript text omitted]
_{\rm ext}~({\rm m}^2)$ | $\sigma_{\rm bsc}({\rm m^2sr^{-1}})$ | $S_{ m lidar}  ( m sr)$ |
|--------------------------|--------------------------------|--------------------------------------|-------------------------|
| Atmospheric Gas          | $3.125\times10^{-32}$          | $3.680 \times 10^{-33}$              | 8.49                    |
| Ash 1 (1 $\mu m$ )       | $4.324\times10^{-12}$          | $0.328\times10^{-12}$                | 58.83                   |
| Ash 2 $(3\mu\mathrm{m})$ | $17.821 \times 10^{-12}$       | $3.843\times10^{-12}$                | 5.23                    |
| Ash 3 $(5\mu\mathrm{m})$ | $61.672 \times 10^{-12}$       | $6.200 \times 10^{-12}$              | 11.90                   |
| Ash 4 (10 $\mu m$ )      | $177.045 \times 10^{-12}$      | $5.365 \times 10^{-12}$              | 64.16                   |
| Ash 5 (15 $\mu m$ )      | $526.967 \times 10^{-12}$      | $20.442 \times 10^{-12}$             | 47.21                   |
| Ash 6 (30 $\mu m$ )      | $1937.387 \times 10^{-12}$     | $23.781 \times 10^{-12}$             | 179.58                  |

**Table 4.** Point-data extraction of COMSO-ART output; case 1 from 16 April 2010, 18:00 UTC, in a height of 1.9 km ASL. Using the number density  $N_d$  of volcanic ash class d, we calculated the individual backscatter coefficient  $\beta_{\text{par},d,\lambda}$ , the contribution to the total backscatter coefficient  $\sum \beta_{\text{par},d,\lambda}$ , the individual mass density  $\rho_d$ , and the contribution to the total mass density  $\sum \rho_d$ . Ash particles were treated as spherical objects with a volumetric mass density of 2500 kg m-3.

| d | $N_d$    | $\beta_{\mathrm{par},d,\lambda}$  | $rac{eta_{\mathrm{par},d,\lambda}}{\sumeta_{\mathrm{par},d,\lambda}}$ | $ ho_d$              | $\frac{\rho_d}{\sum \rho_d}$ |
|---|----------|-----------------------------------|------------------------------------------------------------------------|----------------------|------------------------------|
| - | $m^{-3}$ | $\mathrm{m}^{-1}\mathrm{sr}^{-1}$ | -                                                                      | ${\rm kgm^{-3}}$     | -                            |
| 1 | 43653522 | $1.4\times 10^{-5}$               | 22.3%                                                                  | $0.57\times 10^{-7}$ | 3.3%                         |
| 2 | 7044794  | $2.7\times10^{-5}$                | 41.9%                                                                  | $2.49\times 10^{-7}$ | 14.2%                        |
| 3 | 3194338  | $2.0\times 10^{-6}$               | 30.7%                                                                  | $5.23\times10^{-7}$  | 29.8%                        |
| 4 | 462402   | $2.5\times10^{-6}$                | 3.8%                                                                   | $6.05\times 10^{-7}$ | 34.5%                        |
| 5 | 37161    | $7.6\times 10^{-7}$               | 1.2%                                                                   | $1.64\times 10^{-7}$ | 9.3%                         |
| 6 | 4474     | $1.1\times 10^{-7}$               | 0.2%                                                                   | $1.58\times 10^{-7}$ | 9.0%                         |

Table 5. The same as Table 4 but for case 2 at 16 April 2010, 09:00 UTC, in a height of 1.5 km ASL.

| d | $N_d$    | $\beta_{\mathrm{par},d,\lambda}$  | $rac{eta_{\mathrm{par},d,\lambda}}{\sumeta_{\mathrm{par},d,\lambda}}$ | $ ho_d$                   | $\frac{\rho_d}{\sum \rho_d}$ |
|---|----------|-----------------------------------|------------------------------------------------------------------------|---------------------------|------------------------------|
| - | $m^{-3}$ | $\mathrm{m}^{-1}\mathrm{sr}^{-1}$ | -                                                                      | ${\rm kgm^{-3}}$          | -                            |
| 1 | 93.0     | $30.7\times10^{-12}$              | 0.2%                                                                   | $

---

## Author Comment (AC3) · 12 Nov 2016

**A Backscatter Lidar Forward Operator for Particle-Representing Atmospheric Chemistry Models**

Armin Geisinger[1], Andreas Behrendt[1], Volker Wulfmeyer[1], Jens Strohbach[1], Jochen Förstner[2], Roland Potthast[2], and Ina Mattis[3]

[1]Institute of Physics and Meteorology, University of Hohenheim, Germany
[2]Headquarter of the German Weather Service, Offenbach, Germany
[3]Observatory Hohenpeissenberg of the German Weather Service, Hohenpeissenberg, Germany

*Correspondence to:* a.geisinger@uni-hohenheim.de

**Abstract.**

State-of-the-art atmospheric chemistry models are capable of simulating the transport and evolution of particles and trace gases but there is a lack of reliable methods for model validation and data assimilation. Networks of automated ceilometer lidar systems (ACLs)  provide a 3D dataset of atmospheric backscatter profiles. However, as the aerosol number concentration cannot be obtained from the ACL data alone,  a backscatter-lidar forward model  is required which allows for a qualitative and quantitative model validation based on ACL data.  We developed a new backscatter-lidar  forward operator which is based on the distinct calculation of the aerosols' backscatter and extinction properties. The forward operator was adapted to the ~~high sensitivity of the optical cross sections to the particle size and shape: A slightly different particle radius may lead to quite a large change of the scattering properties. As most particle size distributions are continuous in reality, the optical cross sections are averaged over certain size-intervals which also reduces the problematic and unrealistic sensitivity significantly. To calculate the attenuated backscatter coefficient, the size-dependent particle number concentration and the scattering properties of each particle type and size have to be simulatedahaveaboutindices and shapesparticle properties as represented by the model system. The strong sensitivity of the scattering characteristics to the particle radius was largely reduced by size-averaging algorithms. We focus on a case study of the eruption of the Islandic volcano Eyjafjallajökull from 20 March 2010 to 24 May 2010. The Consortium for Small-scale Modeling - Aerosols and Reactive Trace gases (COSMO-ART) model of DWD (Deutscher Wetterdienst) and KIT (Karlsruhe Institute of Technology) was used during this event for ash-dispersion simulation over Europe. For the forward model~~ assumed particle properties. Therefore, scattering calculations for several types of non-spherical particles required the

usage of t-matrix routines. Due to the distinct calculation of the backscatter and extinction properties of the models' volcanic ash size classes, the sensitivity studies could be resolved to each size class individually which is not the case for forward models based on a fixed lidar ratio. Finally, the forward modeled lidar profiles were compared to ACL measurements both qualitatively and quantitatively. Therefore, the attenuated backscatter coefficient  was chosen as common physical quantity which - for calibrated ACL measurements - only relies on the ACL laser wavelength. ~~Finally, the forward modeled lidar profiles were compared to ACL measurements. Significant differences between ACL profiles and the output of the forward operator applied to the COSMO-ART data were found but also several identical features have been observed. Comparing the data quantitatively revealed that the model-predicted ash number concentration is slightly too high. We identified the following~~ As the ACL measurements were not calibrated automatically, their calibration had to be performed using CALIPSOs/CALIOP measurements. After calibration and comparing model and measurement at the same time and geographic location, a slight overestimation of the model predicted volcanic ash number density was observed. By manually reducing the model predicted ash number density, the effect of simple data assimilation methods could be demonstrated. Unfortunately, the uncertainties related to both measurement and forward operator currently limit more precise model validation attempts using the ACL measurements. The forward operator and the detailed analysis of the related uncertainties, however, allowed us to identify the key issues which are mandatory for performing quantitative ~~comparisons between forward modeled and measured ACL profiles: First, it is suggested that the ACL systems perform automatic calibration and return the attenuated backscatter coefficient directly. Second, the particles' scattering properties have to be analyzed even more extensively as significant differences of the backscatter efficiency occur depending on the shape and refractive indices. Nevertheless, the results of this study allow for a quantitative estimation of the volcanic ash mass concentration from ACL data as well as the related uncertaintiesthecontents of lidar backscatter data~~ content delivered by lidar backscatter measurement in chemical weather forecast models. The introduced forward operator offers the flexibility to be adapted and refined and can thus be used on a multitude of model systems and measurement set-ups.

**1 Introduction**

In Spring 2010, the volcano Icelandic volcano Eyjafjallajökull erupted several times. The emitted ash was found to be harmful for aircraft and due to uncertain information about spatial distribution and concentration of volcanic ash, the European air space was closed for several days (Sandrini et al., 2014). The high economic costs and impact on public transport lead to efforts of DWD (Deutscher Wetterdienst) to improve at monitoring and predicting aerosol cloud movements in the atmosphere. Therefore, DWD decided to start a dedicated project on backscatter lidar forward operators for validating aerosol dispersion models using available remote sensing measurement data.

[revised manuscript text omitted]
_{\mathrm{mol},\lambda}(z) = N_{\mathrm{mol},\lambda}(z)\ \sigma_{\mathrm{sca},\mathrm{mol},\lambda}, \tag{7}$$

$$\beta_{\mathrm{mol},\lambda}(z) = N_{\mathrm{mol},\lambda}(z)\left(\frac{d\sigma_{\mathrm{sca},\mathrm{mol},\lambda}}{d\Omega}\right)_\pi. \tag{8}$$

The molecule number density $N_{\mathrm{mol}}(z)$ is related to the ideal gas law

$$N_{\mathrm{mol}}(z) = \frac{p(z)}{k\ T(z)}, \tag{9}$$

where $p$ is the model prediction of the atmospheric pressure given in Pascal (Pa), $T$ is temperature given in Kelvin (K), and $k$ is the Boltzmann constant which has a value of about $1.381 \times 10^{-23}\ \mathrm{J K^{-1}}$.

To calculate the scattering cross-section $\sigma_{\mathrm{sca},\mathrm{mol},\lambda}$ and the scattering phase function $\phi_{a,\lambda}(\theta)$ of air, empirical equations are available. We used the formulas and look-up tables given by (Buchholtz, 1995). As the empirical equations are only given for wavelengths up to $1000\,\mathrm{nm}$, we  simply extrapolated the values to the ACL wavelength in the case study ($1064\,\mathrm{nm}$). This method allows us to calculate the scattering properties of air molecules $\sigma_{\mathrm{sca},\mathrm{mol},\lambda}$ and $\left(\frac{d\sigma_{\mathrm{sca},\mathrm{mol},i,\lambda}}{d\Omega}\right)_\pi$ directly from model output for a given ACL laser wavelength $\lambda$.

**2.2.3 Scattering by Particles**

The scattering characteristics of spheres with sizes not much smaller or larger than the wavelength are described by Mie's solution of the Maxwell equations (Mie, 1908; Wiscombe, 1980). Methods like the T-matrix (Mishchenko et al., 2002) or the discrete dipole approximation (DDA, Draine and Flatau (1994)) allow for scattering calculations of non-spherical objects; again, with sizes not much smaller or larger than the wavelength. The T-matrix algorithm is a tool for computing scattering by single and compounded particles (Mishchenko et al., 2002). It is faster than DDA but limited to rotationally symmetric objects such as ellipsoids, cylinders or Chebyshev polynomials. DDA, however, can represent arbitrarily shaped objects at the cost of high computational efforts.

As a rough estimate from our studies, the computational costs increase by about one order of magnitude when using the T-matrix instead of the Mie approach and by another two orders of magnitude when using the DDA instead of the T-matrix approach. Another increase in computational time is resulting from larger scatterers, i. e., an increase of the particle size causes an exponential increase of computing time. We therefore utilize Mie scattering algorithms to perform fast calculations although solid particles are in fact non-spherical. The effect of scattering by non-spherical particles is analyzed in a second step using the T-matrix approach for sensitivity studies. We consider these approaches as sufficient for the sensitivity studies performed in this work.

Mie-scattering related computations were performed using the IDL procedure "mie_single", provided by the Department of Atmospheric, Oceanic and Planetary Physics (AOPP), University of Oxford. Input parameters of the procedure are the real part $m$ and imaginary part $m'$ of the refractive index as well as the so-called size parameter $X_\lambda(R)$:

$$X_\lambda(R) = \frac{2\pi R}{\lambda}, \tag{10}$$

where $R$ is the radius of a single particle. The relevant output parameters are the extinction efficiency $Q_{\text{ext},p,\lambda}(R)$, the scattering efficiency $Q_{\text{sca},p,\lambda}(R)$, and the backscatter efficiency $Q_{\text{bsc},p,\lambda}(R)$. These optical efficiencies are defined as ratio between the optical cross section and the physical cross section:

$$Q_{\text{ext},p,\lambda}(R) = \frac{\sigma_{\text{ext},p,\lambda}(R)}{\pi R^2}, \tag{11}$$

$$Q_{\text{sca},p,\lambda}(R) = \frac{\sigma_{\text{sca},p,\lambda}(R)}{\pi R^2}, \tag{12}$$

$$Q_{\text{bsc},p,\lambda}(R) = \frac{\left(\frac{d\sigma_{\text{sca},p,\lambda}(R)}{d\Omega}\right)_\pi}{\pi R^2}. \tag{13}$$

As a warning, we like to point out that the procedure changed its definition of the backscatter efficiency: The current (2012) release of mie_single returns the so-called radar backscatter efficiency which is $4\pi$ times the backscatter efficiency as we

require it within the forward operator. Furthermore, the procedure expects the imaginary part of the refractive index given as negative number. If positive imaginary part values are used, the procedure runs without showing an error but returns wrong results.

**2.2.4 Discrete Particle Number Size Distributions**

5   A major problem of discrete size distributions is the high sensitivity of the optical cross sections to the particle size: A slightly different particle radius may lead to quite a large change of the scattering properties. We present in the following a suggestion to overcome this fundamental problem. Due to the fact that naturally occurring particle size distributions are not discrete, averaging the optical cross sections over certain size-intervals seems straightforward. We will show that this approach indeed reduces the problematic and unrealistic sensitivity significantly. If the model represents only one type of particle, i. e. with
10  a constant refractive index but with discrete radii $R_d$, we can define the effective extinction cross section and the effective backscatter cross sections with

$$\overline{\sigma_{\text{ext},R_d,m,m',\lambda}} = \frac{1}{R_{d_b} - R_{d_a}} \int\limits_{R_{d_a}}^{R_{d_b}} Q_{\text{ext}}(X_\lambda(R_d),m,m')\,\pi R_d^2\,dR_d, \tag{14}$$

$$\overline{\sigma_{\text{bsc},R_d,m,m',\lambda}} = \frac{1}{R_{d_b} - R_{d_a}} \int\limits_{R_{d_a}}^{R_{d_b}} Q_{\text{bsc}}(X_\lambda(R_d),m,m')\,\pi R_d^2\,dR_d, \tag{15}$$

15  where $R_{d_a}$ and $R_{d_b}$ are size margins for each particle size class $d$. These integrals are then exchanged by sums in the numerical computation routines giving

$$\overline{\sigma_{\text{ext},R_d,m,m',\lambda}} = \frac{1}{n_{\text{samples}}} \sum_{g=1}^{n_{\text{samples}}} Q_{\text{ext}}(X_\lambda(R_{d_g}),m,m')\,\pi R_{d_g}^2, \tag{16}$$

$$\overline{\sigma_{\text{bsc},R_d,m,m',\lambda}} = \frac{1}{n_{\text{samples}}} \sum_{g=1}^{n_{\text{samples}}} Q_{\text{bsc}}(X_\lambda(R_{d_g}),m,m')\,\pi R_{d_g}^2, \tag{17}$$

20  where $n_{\text{samples}}$ is the sampling number and the sampling range $R_{d_b} - R_{d_a}$ is broken down in $g$ subsamples:

$$R_{d_g} = g\,\frac{R_{d_b} - R_{d_a}}{n_{\text{samples}}} + R_{d_a}. \tag{18}$$

This calculation of the effective values is performed for every discrete size class $d$ and - if represented by the model - also for every particle type $k$.

Consequently, the total particle extinction coefficient $\alpha_{\text{par},\lambda}(z)$ and the total particle backscatter coefficient $\beta_{\text{par},\lambda}(z)$ are
25  calculated from:

$$\alpha_{\text{par},\lambda}(z) = \sum_k \sum_d N_{d,k}(z)\,\overline{\sigma_{\text{ext},R_d,m_k,m_k',\lambda}}, \tag{19}$$

$$\beta_{\mathrm{par},\lambda}(z) = \sum_k \sum_d N_{d,k}(z)\, \overline{\sigma_{\mathrm{bsc},R_d,m_k,m'_k,\lambda}}. \tag{20}$$

Here, $N_{d,k}$ is the particle number per volume given by the model, $\overline{\sigma_{\mathrm{ext},R_d,m_k,m'_k,\lambda}}$ and $\overline{\sigma_{\mathrm{bsc},R_d,m_k,m'_k,\lambda}}$ are the effective optical cross sections of particle size class $d$ and particle type class $k$ with the respective real part $m_k$ and imaginary part $m'_k$ of the refractive index.

This simple solution allows for calculating $\alpha_{\mathrm{par},\lambda}(z)$ and $\beta_{\mathrm{par},\lambda}(z)$ by just solving a few multiplications and summations resulting in a minimal demand of computing time.

**2.2.5 Lidar Ratio and Two-Way Transmission**

The forward modeled total extinction coefficient and total backscatter coefficient are the sum of the molecule and the particle extinction and backscatter coefficients:

$$\alpha_\lambda(z) = \alpha_{\mathrm{mol},\lambda}(z) + \alpha_{\mathrm{par},\lambda}(z), \tag{21}$$

$$\beta_\lambda(z) = \beta_{\mathrm{mol},\lambda}(z) + \beta_{\mathrm{par},\lambda}(z), \tag{22}$$

equivalent to Eq. (3) and (4). The lidar ratio $S_{\mathrm{lidar}}(z)$ is calculated by

$$S_{\mathrm{lidar}}(z) = \frac{\alpha_{\mathrm{par},\lambda}(z)}{\beta_{\mathrm{par},\lambda}(z)}. \tag{23}$$

[revised manuscript text omitted]

reads

$$\gamma_\lambda^*(z) = \frac{N_{\mathrm{rec},\lambda}(z,t)\, z^2}{N_{\mathrm{tr},\lambda}\, \eta_\lambda\, A_{\mathrm{tel}}\, O(z)\, \Delta z}. \tag{27}$$

The emitted photon number per shot $N_{\mathrm{tr},\lambda}$ was calculated using:

$$N_{\mathrm{tr},\lambda} = \frac{E_{\mathrm{pulse},\lambda}}{E_{p,\lambda}}, \tag{28}$$

where $E_{\mathrm{pulse},\lambda}$ is the laser pulse energy. $E_{p,\lambda}$ is the photon energy, calculated according to:

$$E_{p,\lambda} = \frac{h\, c}{\lambda}, \tag{29}$$

with $h$ as Planck's constant having a value of $6.62607 \times 10^{-34}\,\mathrm{J\,s}$. The pulse energy of the diode-pumped laser is $8\,\mu\mathrm{J}$ (Flentje et al., 2010c) resulting in an emitted photon number per pulse of about $4.28 \times 10^{13}$. The diameter of the receiving telescope is $100\,\mathrm{mm}$ (Flentje et al., 2010c) which results in $A_{\mathrm{tel}} = 78.54\,\mathrm{cm}^2$. The vertical resolution $\Delta z$ is $15\,\mathrm{m}$ over the complete profile. The overlap function $O(z)$ was set to $1$ which implies that ranges below about $1500\,\mathrm{m}$ cannot be used reliably for comparisons with the forward operator.

Unfortunately, the instruments provided no calibrated measurement data at that time. As the true system efficiency $\eta_\lambda$ and the calibration coefficients are not known, we use the symbol $\eta^*$ as linear calibration factor. From a comparison with the calibrated attenuated backscatter measurements of CALIOP at $\lambda = 1064\,\mathrm{nm}$ (Fig. 3), we determined a calibration factor of $\eta^* = 0.003$: First, we used the CALIOP value of the $1064\,\mathrm{nm}$ calibrated attenuated backscatter coefficient at $50.15°$ lat / $4.81°$ lon in a height of $2\,\mathrm{km}$ ASL (about $5 \times 10^{-6}\,\mathrm{m}^{-1}\,\mathrm{sr}^{-1}$) to  estimate the maximum attenuated backscatter coefficient  inside the volcanic ash cloud. The second calibration criterion was the forward modeled attenuated backscatter coefficient outside the volcanic ash layer in $3\,\mathrm{km}$ ASL which is dominated by molecule scattering and has a value of  $1 \times 10^{-7}\,\mathrm{m}^{-1}\,\mathrm{sr}^{-1}$, see Fig. 15. This pragmatic approach is of  limited precision and only required for uncalibrated ACL measurements. As most ACL networks have been extended by automatic calibration methods after the Eyjafjallajökoll eruption,  such a calibration will not be required in future forward operator studies.

[revised manuscript text omitted]

The refractive index measurements by Schumann et al. (2011) were not performed for the wavelength of the ACL systems as explained in section 3.4. Therefore, we have to assume the refractive index we use as reference as well as an interval of uncertainty. Schumann et al. (2011) take a refractive index of 1.59 - 0.004i for their medium "M" case study and therefore, we also used this value as reference for our study. Our uncertainty intervals of real and imaginary parts were chosen according to the range of measured values at $630\,nm$ and $2000\,nm$, namely a real part range of 1.54 to 1.64 and an imaginary part range of -0.006 to -0.002. To get an estimate of the overall refractive index sensitivity for such particles, we decided to extend the range of analyzed refractive indices to real parts between 1.49 and 1.69 using increments of 0.001 and to imaginary parts between -0.011 and -0.001 using increments of 0.00005. Consequently, the total element number of one LUT is $4.0 \times 10^7$. These look-up tables were the base for the refractive index sensitivity study.

**4.2 Sensitivity to the Complex Index of Refraction**

Fig. 5 and Fig. 6 show $\sigma_{\mathrm{ext}}$ and $\sigma_{\mathrm{bsc}}$ as well as $\overline{\sigma_{\mathrm{ext}}}$ and $\overline{\sigma_{\mathrm{bsc}}}$ against the real and imaginary parts of the complex index of refraction. While the extinction cross section $\sigma_{\mathrm{ext}}$ is more sensitive to the real part than to the imaginary part of the refractive

index, the backscatter cross section $\sigma_{\text{bsc}}$ is strongly sensitive to both. These sensitivities are strongly reduced for the effective extinction cross section $\overline{\sigma_{\text{ext}}}$ and the effective backscatter cross section $\overline{\sigma_{\text{bsc}}}$.

An overview of the refractive index sensitivity of the effective optical cross sections is given by Fig. 7 which shows the relative errors

$$\quad \sigma_{\text{ext,err},p}(m,m') = \frac{\overline{\sigma_{\text{ext},p}(m,m')} - \overline{\sigma_{\text{ext},p}(m^*,m'^*)}}{\overline{\sigma_{\text{ext},p}(m^*,m'^*)}} \cdot 100\%, \tag{30}$$

and

$$\sigma_{\text{bsc,err},p}(m,m') = \frac{\overline{\sigma_{\text{bsc},p}(m,m')} - \overline{\sigma_{\text{bsc},p}(m^*,m'^*)}}{\overline{\sigma_{\text{bsc},p}(m^*,m'^*)}} \cdot 100\%. \tag{31}$$

The relative error is the error of the optical cross sections if we assume that the refractive index we defined as reference ($m^*$ and $m'^*$) to be true but the real particles have a refractive index of $m$ and $m'$. We can conclude from this analysis that the
10   maximum relative error for the given range of refractive indices is less than 10 % for the extinction cross section but ranges up to 230% for the backscatter cross section.

**4.3  T-matrix Particle Shape Sensitivity Study**

T-matrix calculations within this study are based on the FORTRAN code for randomly oriented particles written and provided by Mishchenko and Travis (1998). A detailed description of the method can be found in the work of Mishchenko et al. (2002).
15   We modified the double-precision version of the T-matrix procedure to perform scattering calculations of multiple particle sizes automatically. In addition to that, the procedure was extended by the calculation of the backscatter cross section $\sigma_{\text{bsc}}$ according to Mishchenko et al. (2002), Eq. (9.10). Then, as a test, both mie_single and our modified T-matrix code were set up to calculate the scattering properties of the same spheres and for the same wavelength. The results of both procedures were indeed identical.
20   A list of T-matrix options we used for the particle shape sensitivity study is shown in Table 2. The most important particle properties are defined by the variables NP and EPS. NP defines the particle type and has a value of -1 for spheres as well as for ellipsoids. A NP value of -2 is used for cylinders. The variable EPS is an expression for the objects' diameter to length ratio. Consequently, an ellipsoid with EPS=1 is a sphere, prolate objects have an EPS<1 and oblate objects have an EPS>1.

In Figs 8, 9, and 10, the optical cross sections and the lidar ratio of spheres and several aspherical
25   particles are plotted against the equal-volume radius. The  aspherical scatterers are  6 ellipsoids with a diameter-to-length-ratio of  0.50, 0.67, 0.75, 1.25, 1.50 and 2.00 as well as 5 types of cylindric particles with a diameter-to-length-ratio of 0.50, 0.80, 1.00, 1.25 and 2.00. As shown in these plots, the results for a highly asymmetric ellipsoid (EPS: 0.50) is only available up to an equal-volume radius of 3.75 µm. For future research activities in this topic, the quadruple precision version of the t-matrix code should be preferably used.
30   We found no significant differences of the extinction cross section of spheres and these ellipsoids.  By trend, cylindric shaped particles have a higher extinction cross section compared to ellipsoids and the

spherical shape has the lowest extinction cross section values over the whole spectrum. Up to a volume-equivalent radius of $0.7\,\mu m$, the shape effect is not noticeable.

Regarding the backscatter cross section, however, we find significant differences between the backscatter cross section of spheres and particles with other shapes. Obviously, spheres are affected by interference effects which lead to both fluctuating and oscillating values of the backscatter cross section while the other shapes only show weakly fluctuating values of the backscatter cross section. As observed for the extinction cross section, the shape effect becomes pronounced beginning at an equal-volume radius greater than $0.7\,\mu m$. Spherical scatterers have a higher value of the backscatter cross section compared to ellipsoids except for one type of ellipsoid (EPS = 1.25). For cylinders, such interference effects are not observable so the backscatter cross section of the considered cylinders increases monotonically with the equal-volume radius. As a result, the backscatter cross section of spheres is lower than of cylindric particles with the same size if their equal-volume radius is greater than $3.75\,\mu m$ (for the given wavelength of $\lambda = 1064\,nm$).

The particle shape effect on the lidar ratio is only weakly pronounced for small particle sizes (less than $0.75\,\mu m$), too. For larger particles, the lidar ratio of spheres is generally lower than of the other considered shapes which is in agreement to the higher backscatter cross section observed before. For the fourth size class (equal-volume radii around $5\,\mu m$), the previously observed interference effects of the spheres' backscatter cross section leads to extreme values of the lidar ratio (exceeding a lidar ratio of $200\,sr$). By trend, however, the lidar ratio of spheres in this size class is not that different than the lidar ratio of other considered shapes. But for the other size classes, the lidar ratio of spheres is the lowest of all observations except for cylinders which have a lidar ratio value at the same order of magnitude as spheres.

A summary of the particle shape sensitivity study is shown in Figs 11, 12, and 13, giving the relative differences of the effective optical cross sections and average lidar ratios for different particle shapes. The definition of the relative differences follows Eq. (30) and Eq. (31). Again, the reference is a spherical particle. Positive and negative relative differences indicate that the calculated values for spheres are underestimations and overestimations compared to respective non-spherical particles.

The effective extinction cross section of spheres is lower than the effective extinction cross section of other analyzed asymmetric particles. Regarding the effective backscatter cross section, however, we find relative differences of up to 300% and −80%. While the small aspherical particles have a lower effective backscatter cross section compared to spheres, the effective values of the fourth size class are higher for almost all considered

5  aspherical particles.

 From this analysis, it can be concluded that due to the assumption of sphericity, the backscatter cross section of size classes 1, 2 and 3 are overestimated by about factor 1.5 to 5 while the backscatter cross section of the fourth size

10  class is underestimated by factor 2. This allows for quantifying the over- and underestimation of the results for each size class individually which is not possible for forward operators where a fixed lidar ratio is assumed.

The relative difference of the average lidar ratio shows that assuming spheres for the scattering calculations results in much lower lidar ratio values of size classes 2 and 3 but in higher lidar ratio values than would be observed for a mixture of particle shapes. For the ~~effective backscatter cross section, the mean difference is between 110% for large scatterers and −60 % for

15  medium and small scatterers.~~ fourth size class, the opposite has been observed: assuming spheres leads to an higher values of the lidar ratio of large particles than for non spherical particles of the same size. The lidar ratio of the first class, however, is nearby independent on the particle shape.

Of course, the total effect of different particle shapes for a real aerosol mixture has to be estimated with more extensive scattering calculations. For the current state of the forward operator research, the spherical shape has to be used as reference

20  even if the real volcanic ash particles are known to be fractal and complex shaped. One of the reasons for doing so is the fact that there is currently no appropriate shape representation scheme for volcanic ash available. For example from the findings of Rocha-Lima et al. (2014), the average aspect ratio of volcanic ash is known but not a representative particle shape. As we have shown, the backscatter cross section of cylinders also differs tremendously from ellipsoids and from spheres etc. so there is currently a lot of indication that using a given aspherical shape as reference would yield in even higher errors than assuming

25  a spherical shape.

The above results, however, give us valuable insight into the uncertainties of using Mie calculations for non-spherical particles.  These findings are important for interpreting the results of the forward operator. ~~The assumption of spherical volcanic ash particles results in an overestimation of the backscatter cross section of size

30  classes 1, 2, and 3 by factor 2-3 which results in an equivalent underestimation of the respective lidar ratio values.~~

**5   Results**

**5.1   Scattering Properties of Volcanic Ash Used Within the Forward Operator**

A list of the effective extinction cross section and the effective backscatter cross section we determined for atmospheric gas molecules and for the six volcanic ash size classes is shown in Table 3.

**5.2   Output Variables of the Forward Operator**

Using the forward operator allows for plotting each variable of the lidar simulation for analytic purposes (see Fig. 14). These plots of forward-operator output variables are representing the major characteristics of the variables: strong extinction and strong backscattering are usually related. Time and height intervals at which only molecules exist, lead to low values of the extinction coefficient and backscatter coefficient. Due to the decrease of the atmospheric gas number density with height, both extinction and backscatter coefficient decrease with height in an aerosol-free atmosphere. The two-way transmission decreases with height (see Eq. (24)).

~~The forward operator yields a lidar ratio for volcanic ash of between $15\,\mathrm{sr}$ and $80\,\mathrm{sr}$ which differs to lidar ratio values we find in literature ($40\,\mathrm{sr}$ to $100\,\mathrm{sr}$ for volcanic ash (Kokkalis et al., 2013; Ansmann et al., 2010; Mortier et al., 2013)). This difference may be a consequence of assuming spherical particles. But also for spherical particles we would expect the given lidar ratio as the extinction cross section increases exponentially with the particle size while the backscatter cross section is influenced by interference effects (see. Fig. ?? and Fig. ??). This is also represented by the effective optical cross sections which are used by the forward operator (see. Table. 3). Comparing these values with the lidar ratio findings by Gasteiger et al. (2011b), a lidar ratio of less than $20\,\mathrm{sr}^{-1}$ seems to be plausible. The authors found even for irregularly shaped objects a lidar ratio between $5\,\mathrm{sr}^{-1}$ and $20\,\mathrm{sr}^{-1}$ for size parameters between 5 and 15 (equivalent particle diameter at $\lambda = 1064\,\mathrm{nm}$ would be $1.6\,\mu\mathrm{m}$ and $4.8\,\mu\mathrm{m}$, respectively). Due to the different refractive indices and particle shapes, this comparison is no validation for our findings but shows that the result of our analysis is close to the reference calculations.~~

 In addition, the volcanic ash contribution to the total signal and the total mass density was analyzed for each size class of COSMO-ART  within two case studies. The cases were selected to cover  extreme situations: Case 1 is the model output from a coordinate inside the volcanic ash layer (Table 4). Case 2 is for  a coordinate where the majority of particles are due to size class 4 and 6 (see Table 5).

Regarding case 1, the total backscatter coefficient is dominated by ash size classes 1, 2 and 3 while the signal contribution of classes 4 to 6 is less than $5\,\%$ in total. The mass contribution is dominated by classes 3 and 4 while classes 2, 5 and 6 are contributing by $10\,\%$ each to the total mass density and class 1 is  very low for the total mass density. Regarding case 2, the total backscatter coefficient depends by about $68\,\%$ from class 4 and by $30\,\%$ from class 6.  The mass contribution in case 2 is also dominated by the classes 4 and 6 but, in contrast to the backscatter coefficient, class 6 has a higher contribution to the total mass density than class 4.

General conclusions from this analysis about the relationship between backscattering and mass depending on particle size and wavelength require further investigation. For our application of the forward operator in this study, however, we can conclude the following: First, the total signal inside the volcanic ash layer (case 1) is predominantly dependent on classes 1, 2 and 3 whose backscatter cross sections are also overestimated by the forward operator due to the assumption of sphericity (see Fig. 12). The real values of the attenuated backscatter coefficient may be by factor 2-3 higher. Second, the larger particles of classes 4, 5 and 6 carry a large portion of the mass but contribute only weakly to the total signal. This may be an important information for the selection of future ACL networks as even the systems operating at 1064 nm have a reduced sensitivity for particles within these size classes.

**5.3 Qualitative Comparison**

[revised manuscript text omitted]

Due to uncertain refractive indices of the volcanic ash and due to the fact that volcanic ash particles  show complex shapes, sensitivity studies have been performed to analyze the  impact of different particle shapes on the effective extinction and backscatter cross section as well the lidar ratio. While the extinction cross section was only weakly sensitive to variable refractive indices and particle shapes, the backscatter cross section was strongly sensitive to both  . However, we found that the sensitivities reduce significantly, when averages over size classes are made. We expect that this very interesting result - described in this manuscript for the first time - will be very helpful for comparisons of modeled with measured backscatter lidar data.

Using these effective optical cross sections reduced the sensitivity of optical cross sections  regarding the refractive index as well as the particle shape. But even after averaging, the relative uncertainty of the effective backscatter cross section exceeds 280% for uncertain refractive indices. This study also indicates the dependency of the forward operator on precise information about the particle's refractive index. Within a particle shape sensitivity study, we  were able to resolve the relative uncertainty of  each individual size class for the effective backscatter cross section . Assuming that volcanic ash consists of a mixture of particle shapes, we  analyzed the relative differences  between the reference particle shape and 11

particle shapes (6 types of ellipsoids and  5 types of cylinders). ~~The resulting relative difference of the effective backscatter cross sections was quantified to be lower than 110%. The respective relative difference of the effective extinction cross section for mixed particle shapes is less than 6%. Tests using the t-matrix code also for a variety of refractive index values resulted in the same uncertainties regarding the backscatter cross section as observed for spheres. Increasing the aspect ratio of considered particle shapes, however, resulted in even higher relative differences of the backscatter cross section compared to spheres. Unfortunately, the used t-matrix routine was unable to calculate the scattering properties of these particles for particle sizes larger than 3.75 µm (ellipsoids) or 1.0 µm in radius, respectively. An improved sensitivity study approach would demonstrate the sensitivity for different refractive indices, particle shapes and asymmetry factors within one analysis. The sensitivities could then be used by the forward operator to calculate a weighted (either weighted by number or mass concentration) uncertainty estimate~~

The forward operator matches the lidar ratio values we find in literature (40 sr to values higher than 100 sr for volcanic ash (Kokkalis et al., 2013; Ansmann et al., 2010; Mortier et al., 2013)). On average, the lidar ratio is 61.17 sr which fits well to the literature findings. Comparing the lidar ratio values of the first two size classes with the lidar ratio values reported by Gasteiger et al. (2011b), a lidar ratio of less than 20 sr seems to be plausible for these particle size to wavelength ratios. The authors found even for irregularly shaped objects a lidar ratio between 5 sr and 20 sr at size parameters between 5 and 15 (equivalent particle diameter at $\lambda = 1064\,\text{nm}$ would be 1.6 µm and 4.8 µm, respectively). We therefore assume that the calculation results of the backscatter lidar forward operator are valid and allow for both qualitative and quantitative comparison.

A comparison between ACL measurement and the model predictions used as input for the developed forward operator was shown. Similar structures were observed but some features were referenced to different time and height locations. From our analysis at the ACL station Deuselbach, some ash layer features were predicted quite precisely by the model, for example the time of arrival of the ash plume at about 06:00 UTC with a vertical shift of about 1.5 km. Some other features, such as the intersection with the planetary boundary layer at 17 April 2010, 03:00 UTC, was simulated about 6 hours too early to 16 April 2010, 18:00 UTC. Fine structures of the ash layer were only observable in the simulation but not in the ACL measurements due to noise.

Due to unknown calibration coefficients of the ACL system, a calibration constant $\eta^*$ was estimated by comparing the ACL data with calibrated measurements at the same wavelength. Within quantitative comparisons between ACL measurements and the forward operator output, we found that the molecule signal of ACL and forward operator output were of the same order of magnitude which argues that the selected calibration factor was  reasonable. Meanwhile, the ACL manufacturers have understood the importance of  calibrated backscatter data and implemented technical solutions so that a similar effort as described in this study with data for the year 2010 became obsolete.

A comparison the volcanic ash signal led to the conclusion that the model predicted ash concentration was to be too high as the forward modeled attenuated backscatter coefficient within ash layers was 60 times higher and after attenuation 10

times lower than observed by the ACL. If the model-predicted ash concentration is manually reduced by a factor of 20, the forward modeled COSMO-ART predictions and ACL measurements became quantitatively similar. Such a reduction could be part of a simple particle data assimilation system helping to calibrate particle dispersion simulations before in-situ measurements are available.
5  It would of course be beneficial, if even better information on the refractive index and  effective particle shape and aspect ratio of volcanic ash particles becomes available in the future.

Furthermore, we analyzed the contribution of each class to the total backscatter coefficient and to the total mass density  for two sample cases. Regarding case 1 inside the volcanic ash layer, the classes 1, 2 and 3 were mostly responsible
10 (94.8 %) for the calculated attenuated backscatter coefficient. As these classes contribute most to the forward modeled signal, the value of the lidar ratio would also be expected to be dominated by their contribution, namely a value between 5.23 sr and 58.83 sr (see Table 3). Raman lidar measurements of the Eyjafjallajökull ash resulted lidar ratio values greater than about
15  40 sr at wavelengths of 355 nm 532 nm ~~but not for the ACLs wavelength (1064 nm). Second, the forward modeled lidar ratio values of classes 1, 2 and 3 are known to be underestimated by factor 2-3 due to the assumption of sphericity (see section 4). Third, the volcanic ash size classes defined by the model are maybe too coarse. Size class 1 represents ash particles with a diameter of 1 µm so lower margin of the size-averaging algorithm of this class was set to its half, namely 0.5 µm in diameter. This also implies that the majority of the fine fraction is not represented by the model and also not by the forward operator. It~~
20 ~~should be mentioned here that comparisons between measured and forward modeled lidar ratios require further investigation such as the calculated of the weighted lidar ratio where the contribution to the total lidar ratio of each class depends on the respective contribution to the backscattered signal or on the respective number concentration. Lidar ratio simulations of other scientists, however are exactly in the same scope as the forward modeled lidar ratio values of our study. Gasteiger et al. (2011b), for example found lidar ratio values of less than 20 sr$^{-1}$ for absorbing and non-absorbing dust particles with a radius close to~~
25  (Groß et al., 2012) which is within this range. Thus, the calculated values of both extinction and backscatter cross section
30  as well as the lidar ratio seem to be plausible.

[revised manuscript text omitted]

The same as Fig. 11 but for the backscatter cross sections. The mean error is the average relative error of all considered shapes on this plot.

[Figure]

Geoinformation ® Bundesamt für Kartographie und Geodäsie (www.bkg.bund.de)

**Figure 2.** The German ACL network in 2010. Each dot represents an ACL station and the dot color is an indicator for the ash layer visibility within the measurement. Red color: Near-ground fog or water clouds cover the ash cloud, orange: ash clouds are almost (yellow: partially) covered by fog or clouds, green: clean air situation with a full view on the ash layers.

[Figure]

**Figure 3.** Attenuated backscatter coefficient measurement from CALIOP which was used to calibrate the ACL measurement during the Eyjafjallajökull eruption phase. The volcanic ash cloud is visible around $50.15°$ lat and $4.81°$ lon. Image obtained from http://www-calipso. larc.nasa.gov/

[Figure]

**Figure 4.** Sketch of the particle size distribution represented by COSMO-ART within the case study (red dots). The red lines with bars indicate the averaging margins we defined for the calculation of effective optical cross sections.

[Figure]

**Figure 5.** Sensitivity of $\sigma_{\text{ext}}$ to the real and imaginary part of the refractive index for a single particle radius $R_p$ (left) and after calculating the effective extinction cross section $\overline{\sigma_{\text{ext}}}$ (right). The green shaded area is the relevant range of real part $m$ and imaginary part $m'$ as explained in section 4.1.

[Figure]

**Figure 6.** The same as Fig. 5 but for the backscatter cross section $\sigma_{\mathrm{bsc}}$ and the effective backscatter cross section $\overline{\sigma_{\mathrm{bsc}}}$.

[Figure]

**Figure 7.** Relative errors of the effective extinction cross section (top row) and of the effective backscatter cross section (bottom row) if the assumed reference refractive index (red dot) is not equal to the true refractive index. Plots in the left row show the error for variable imaginary parts; plots in the right row for variable real parts of the refractive index. Uncertainties of the imaginary part may lead to a maximum error of 0.5% for the effective extinction cross section and of 230% for the backscatter cross section. Uncertain real parts also may lead to errors of 7% for the extinction cross section as well as of 225% for the backscatter cross section in the worst case.

[Figure]

**Figure 8.** Extinction cross section spectrum for  the reference particle ( sphere, dark grey line), six types of ellipsoids (EPS=1, solid lines), and  5 types of cylinders (EPS=-2, dashed lines)  against the particles' equal-volume radius $R_p$. The  dotted lines indicate the size-margins of each class.

[Figure]

**Figure 9.** The same as Fig.  8 but for the backscatter cross section.

[Figure]

**Figure 10.** The same as Fig.  8 but for the lidar ratio.

[Figure]

**Figure 11.**  Relative errors of the effective extinction cross section if spherical particles are assumed but the real particles would be of elliptical (NP: -1) or cylindrical shape (NP: -2). Negative (positive) values indicate that spherical particles have larger (smaller) effective extinction cross section. Except for a strongly asymmetric cylinder, the  effective extinction cross section of spherical particles are mostly smaller (up to 12 %).

[Figure]

**Figure 12.** The same as Fig.  11 but for  the effective backscatter cross section.

[Figure]

**Figure 13.** The same as Fig. 11 and 12 but for the effective lidar ratio.

[Figure]

**Figure 14.**  Time-height cross section of  total extinction coefficient, backscatter coefficient, and  two-way transmission, calculated by the  forward model based on COSMO-ART output at the station Deuselbach (West Germany). The forward model represents clean air molecules and  volcanic ash particles (no clouds, rain, fog, background aerosol or other scattering objects).

[Figure]

**Figure 15.** Attenuated backscatter coefficient of ceilometer (top) and forward model (bottom) at the station Deuselbach in Germany from the 16th of April 2010, 00:00 UTC to the 17th of April 2010, 24:00 UTC.

[Figure]

**Figure 16.** The same as Fig. 15 with a decreased volcanic ash number density. For this purpose, the ash number density predicted by COSMO-ART was reduced by factor 20 before applying the forward operator.